# Three-dimensional liquid metal-based neuro-interfaces for human hippocampal organoids

Yan Wu[1,2], Jinhao Cheng[1], Jie Qi[1], Chen Hang[1], Ruihua Dong[1], Boon Chuan Low [2,3], Hanry Yu[2,4] & Xingyu Jiang [1]✉

Human hippocampal organoids (hHOs) derived from human induced pluripotent stem cells (hiPSCs) have emerged as promising models for investigating neurodegenerative disorders, such as schizophrenia and Alzheimer's disease. However, obtaining the electrical information of these free-floating organoids in a noninvasive manner remains a challenge using commercial multi-electrode arrays (MEAs). The three-dimensional (3D) MEAs developed recently acquired only a few neural signals due to limited channel numbers. Here, we report a hippocampal cyborg organoid (cyb-organoid) platform coupling a liquid metal-polymer conductor (MPC)-based mesh neuro-interface with hHOs. The mesh MPC (mMPC) integrates 128-channel multielectrode arrays distributed on a small surface area (~2*2 mm). Stretchability (up to 500%) and flexibility of the mMPC enable its attachment to hHOs. Furthermore, we show that under Wnt3a and SHH activator induction, hHOs produce HOPX⁺ and PAX6⁺ progenitors and ZBTB20⁺PROX1⁺ dentate gyrus (DG) granule neurons. The transcriptomic signatures of hHOs reveal high similarity to the developing human hippocampus. We successfully detect neural activities from hHOs via the mMPC from this cyb-organoid. Compared with traditional planar devices, our non-invasive coupling offers an adaptor for recording neural signals from 3D models.

The hippocampus is crucial for memory formation, social cognition, spatial orientation, and learning[1,2]. Neurodegeneration and impairments in the hippocampus can lead to neurological and psychiatric disorders, such as Alzheimer's disease, schizophrenia, and depression[3]. These functions are facilitated by newborn neurons, maturation of granule cells, and synaptic plasticity in the hippocampal DG[4]. However, due to ethical and technical constraints, non-invasive measurement of hippocampal neural activity presents a significant challenge. Recent developments in hippocampal modeling using hiPSCs and 3D neural organoids offer potential solutions[5–7]. Specifically, extrinsic guided factors applied during

organoid derivation mimic in vivo developmental signaling pathways, leading to region-specific neural organoid formation[8]. Notably, the remarkable capability of self-patterning and fusion between different organoids allows the generation of specific tissue structures and functional cellular connections[9,10]. Researchers have generated hippocampal spheroids from patients-derived hiPSCs to investigate schizophrenia[5] and AD[6]. A comparison of single-cell transcriptome analysis between hippocampal spheroids and the human hippocampus revealed the presence of subgroups of glial cells in long-term cultured organoids, similar to those found in adult and aging humans[11]. These findings demonstrate that hHOs can

[1]Department of Biomedical Engineering, Southern University of Science and Technology, Shenzhen, China. [2]Mechanobiology Institute, National University of Singapore, Singapore, Singapore. [3]Department of Biological Sciences, National University of Singapore, Singapore, Singapore. [4]Department of Physiology, National University of Singapore, Singapore, Singapore. ✉e-mail: jiang@sustech.edu.cn

replicate some aspects of the human hippocampus under physiological or pathological conditions.

The lack of suitable devices hinders the characterization of neural activity from hHOs despite their function in transmitting information via huge neural activities. Currently, characterization for all neural organoids primarily focuses on morphogenesis, including composition and structure, using established biotechnologies such as immunofluorescent staining and single-cell sequencing. The electrical information in neural systems in vitro is widely detected through a planar MEA[12,13] and calcium imaging[14–16]. Electrodes in commercial MEAs distribute over a small surface area at the bottom of a single well. Thus, these devices cannot match hHOs in the suspension culture system. Implantable electrodes designed for animals are also unsuitable for hHOs due to a mismatch in shape and size[13,17,18]. Single or arrays of needle electrodes may potentially be inserted into neural organoids to detect signals, like vertically arrayed 3D MEAs[19]. Calcium imaging can allow live imaging, which provides information about electrical activity on a larger scale but is dependent on imaging capabilities. The size and 3D nature of neural organoids pose a challenge for acquiring calcium imaging data. Thus, fabricating stretchable and flexible MEAs specifically tailored for neural organoids has become essential. Although some 3D MEAs have been developed recently for sensing and manipulating neural activity in neural organoids[20–23], there is still plenty of room for improvement regarding adhesion, non-invasion, and effectiveness. With an insulating layer made from photocurable polymer SU-8 and a conductive layer of solid metals, common limitations of several existing 3D MEAs are (1) a few channels (3–25 channels), which cannot completely detect signals from the surface of organoids, and (2) poor flexibility and stretchability, which affect the autonomous growth of organoids.

Existing 3D MEAs specifically formulated for neural organoids were only applied to cerebral[20–22] or cortical organoids[23] and have yet to be employed on hHOs, as there exist more established protocols for generating the cerebral and cortical organoids. Generating hHOs is challenging because more accurate manipulation of signaling factors is required to induce the differentiation process. Human hippocampal primordium develops from dorsomedial telencephalon (DMT), which is regulated by BMPs, Wnts, and SHH[24,25]. Feedback pathways among BMPs and transcription factors, such as LHX2 and FGFs, pattern the telencephalon and induce and organize hippocampal fields[26,27]. Previous in vivo[24,25] and in vitro models[7,28] proved the importance of Wnt3a and SHH signals in hippocampal development. Researchers have identified the feasible timing to add BMP4 and CHIR (a chemical activator of Wnts) in the medium to generate DMT organoids. These DMT organoids gave rise to cortical hem (CH), choroid plexus (ChP), and hippocampal primordium[29]. The CH and ChP release Wnts and BMPs to regulate hippocampus growth[24]. In particular, several works to grow hHOs in vitro have opted for long-term use of the Wnt pathway (Wnt3a, 14 days[7]/Wnt3a, 20 days[5]/ CHIR, 90 days[6]) to induce hippocampal fate. However, the relationship between the Wnt&SHH pathways, ChP in DMT organoids, and hippocampal differentiation has yet to be demonstrated.

The hippocampus seems to hold significant importance concerning neurological disorders. Sourcing electrical information from hHOs holds key potential in laying a cogent foundation for future diagnosis and treatment of neurological diseases. In this article, we report a method to fabricate the flexible and stretchable mMPC composed of gallium indium (GaIn) alloy and elastic polymers (thermoplastic polyurethane, TPU, and polyurethane, PU), aiming to detect signals of hHOs (Fig. 1a). Unlike traditional MEAs with the insulating layer made from photocurable polymer SU-8, our mMPC provided (1) free deformation, including fold, bend, and twist at any angle; (2) stretchability up to 500%, without losing electrical performance under strain. A conductive polymer, poly (3,4-ethylenedioxythiophene)

(PEDOT), was deposited on the electrode surface to improve stability. These features allowed the mMPC to couple with hHOs.

Furthermore, we modulated the DMT induction method into a more hippocampal side, generating a more hippocampal region and a smaller ChP region in the organoid. Under the Wnt3a and SHH activator induction, we generated hHOs with many PAX6[+], HOPX[+] hippocampal progenitors and PROX1[+] granule neurons. After integrating the mMPC with the hHOs to form the hippocampal cyb-organoid platform, 128 channels were distributed on the upper and lower hemisphere surfaces of the hHO. We recorded up to 85 channels of neural signaling, glutamine-mediated higher spike rates, synchronization, and oscillatory network activity from the cyb-organoid. This performance implied that the mMPC enabled higher-throughput detection of neural signals, which is crucial for understanding neural circuits. Overall, the new mMPC exhibited considerable potential for neural signal detection from neural organoids, offering unique opportunities for studying signal transmission in neural tissues.

## Results
### Design and fabrication of 128-channel mMPC
Compared with the conventional electrodes utilized in various neural recordings and wearable devices, the fabrication of 3D mesh MEA poses significant challenges, namely, (1) the reduction of extra substrate and insulating material and (2) the precise encapsulation of the conductor, except for electrodes. In response to these challenges, we developed stretchable electronics by designing three poly(dimethylsiloxane) (PDMS) patterns for the conductor, the bottom substrate, and the top insulating layers, respectively, using the soft lithography process (Supplementary Fig. 1a–d). The conductor layer, which included electrodes and circuits, comprised MPC ink, where nano- and micro-meter GaIn particles were dispersed in a solvent. The MPC ink was not affected by the larger surface tension of pure GaIn and was easily manipulated to print on other substrates (Fig. 1a). Preparation and mechanism of the MPC ink have been described in our previous work[30–32]. To ensure precise exposure of the electrodes while encapsulating the circuits, we designed the PDMS patterns for the substrate and insulating layer differently. This difference was the addition of pillars in the top insulating layer, which matched the location of the electrodes (Supplementary Fig. 1e–h). These pillars prevented the insulating material from covering MPC electrodes.

The complete fabrication process is depicted in Fig. 1b. In brief, the process began with scraping the MPC ink into PDMS slab #1, which contains microchannels, to create the conductor layer (Step i). Following this step, we aligned PDMS slab #2, which also contains microchannels, on the surface of PDMS slab #1 and immersed them in the TPU solution. The TPU solution was allowed to completely fill the microchannels of PDMS slab #2 to produce mesh TPU, which served as the substrate after the solvent had evaporated (Steps ii and iii). These operations were then repeated to generate mesh PU as the insulation (Steps iv and v). Pillars in PDMS slab #3 for the PU layer effectively prevented the PU solution from covering these dots, thus enabling the proper exposure of electrodes (Step v). The mMPC was obtained after removing all PDMS slabs. In the end, the final device has no PDMS, only containing three components: TPU, PU, and liquid metal. To facilitate the easy connection of the mMPC with a signal acquisition instrument, two ends of the mMPC were bonded to the flexible printed circuit through an anisotropic conductive adhesive (Step vii). Notably, this fabrication approach is highly versatile and can produce other shapes of MPC electronics, depending on the design of PDMS patterns.

### Morphology of mMPC
Considering the inevitable impacts of devices on the growth of hHOs, the balance between miniaturization and multi-channels should be considered. To achieve this balance, we employed a design strategy that involved reducing the width of a single MPC circuit to 15 μm while

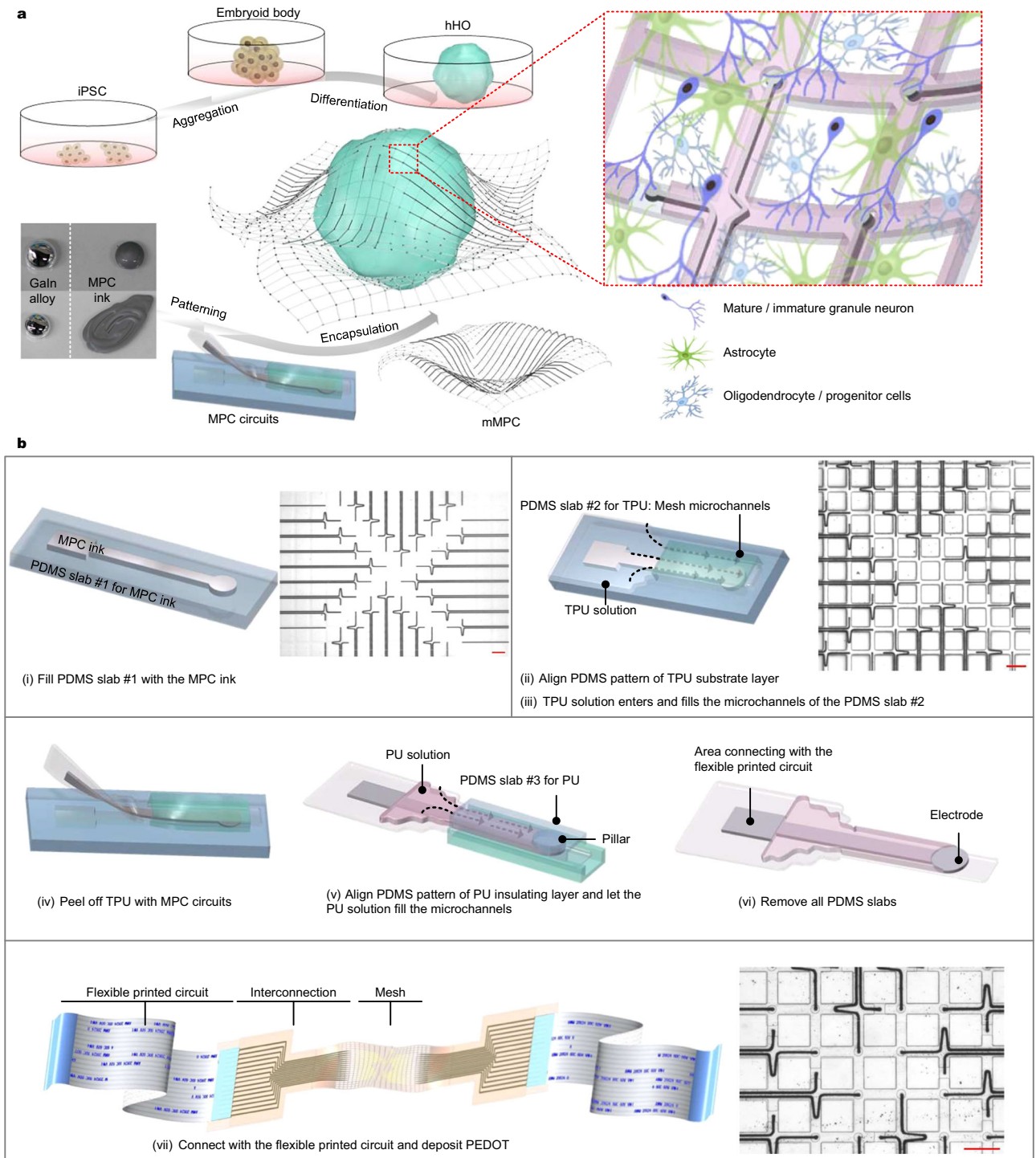

**Fig. 1 | Overview of coupling the mMPC with the hHO and design of the mMPC. a** Schematic diagram of coupling the mMPC with the hHO. **b** Fabrication process of the mMPC. Scale bar: (i) 500 μm; (ii) and (iii) 200 μm; and (vii) 200 μm. hHO human hippocampal organoid, iPSC induced pluripotent stem cell, MPC liquid metal-polymer conductor, mMPC mesh MPC, TPU thermoplastic polyurethane, PU polyurethane.

integrating two such circuits in a 50 μm-wide polymer structure to increase the channel number (Fig. 2a). A total of 64 electrodes, each with a diameter of 20 μm (Fig. 2b and c), distributed in four directions across a ~2*2 mm area, and located at different points within the mesh structure. Furthermore, our 128-channel mMPC was comprised of two layers of 64-electrode mesh, which enclosed the top and bottom hemispheres of the hHO, respectively. The two-layer design not only increased channels but also provided a stable holder for free-floating organoids. The mesh, featuring 150 μm gaps and 30 μm thickness (both

15 μm thickness in TPU and PU layer, Fig. 2b), weighed only ~500 μg. Given all these parameters, our extremely light, soft mMPC was conformal to the object surface (Fig. 2d).

## Mechanical and electrical characterization of mMPC
Conformal attachment is a key issue to consider when applying electronics to neural organoids. The highly flexible and stretchable nature of the mMPC allowed for attachment to the surface of objects (Fig. 2d). The mechanical properties of the mMPC were attributed to the

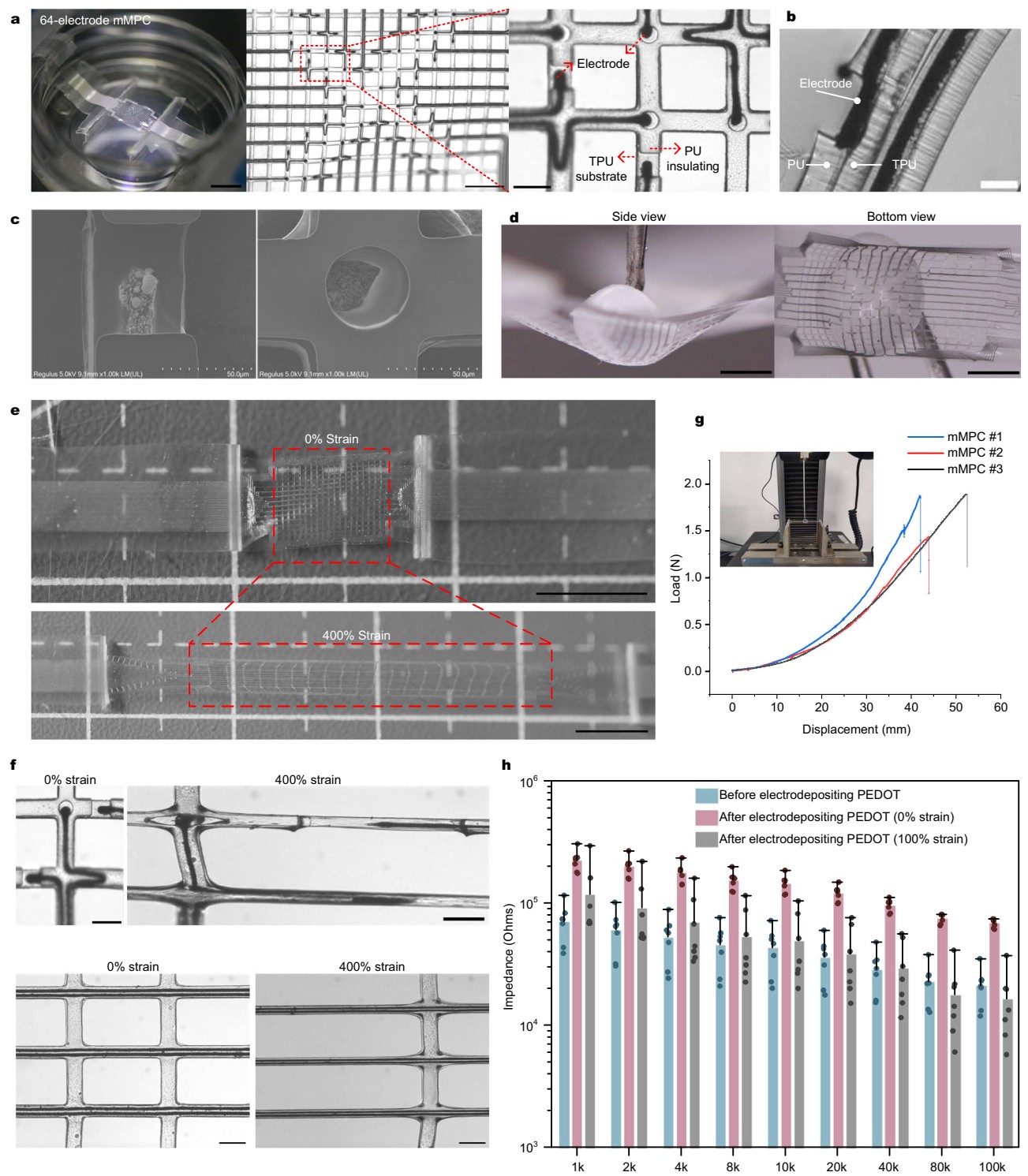

**Fig. 2 | Characterization for the mMPC. a** (Left) the single 64-electrode mMPC. Scale bar: 5 mm. (Middle) the distribution of 64 electrodes in the mesh structure. Scale bar: 500 μm. (Right) the structure of electrodes and encapsulation layers. Scale bar: 100 μm. **b** Optical image of the sandwich structure. Scale bar: 20 μm. **c** SEM images of electrodes. **d** Conformal attachment of the stretchable mMPC to the surface of a TPU ball. Scale bar: 1.5 mm. **e** The mMPC under 400% strain. Scale bar: 5 mm. **f** Two electrodes, circuits of the mMPC before and after 400% strain. Scale bar: 100 μm. **g** The bending test of 3 mMPCs. **h** Impedance of the mMPC across frequency before and after depositing PEDOT on electrodes and under 100% strain ($n = 7$, 7 independent electrodes randomly selected from 2 mMPCs; data are presented as median with maxima). mMPC mesh liquid metal-polymer conductor, TPU thermoplastic polyurethane, PU polyurethane. Images in **c** are representative of $n = 4$ independent measurement.

encapsulation materials, namely TPU and PU, which exhibit excellent elasticity, high resistance, and good biocompatibility, making them broadly used in the medical industry for biomedical devices[33]. The fluidity of GaIn alloy, combined with the properties of TPU and PU, enabled the mMPC to perform ideally in both flexibility and stretchability, allowing for free folding, bending, and twisting (Supplementary Movies 1 and 2).

The extension test demonstrated that the mMPC could stretch up to five times its original length (400% strain, Fig. 2e), while the strain of another mesh MPC with a width of 30 μm was up to 500% (Supplementary Fig. 2a). Remarkably, the GaIn alloy remained entirely enclosed within the sandwich TPU–PU structure without any leakage or fracture, both for the exposed electrodes and the fully encapsulated circuits (Fig. 2f and Supplementary Fig. 2b). While we concerned that the TPU and PU layers might separate under strain, exposing the conductor layer, these two layers remained tightly bonded with no split between them. Notably, the conductivity was maintained at ~4000 S/cm when elongating the mesh by 400% (Supplementary Fig. 2f), underscoring the mMPC's superior mechanical and conductive properties. The bending test also performed great stretchability and easy deformation of the mMPC, and displacement of 500 μm requires only 0.005 N force (Fig. 2g, an average bending stiffness of $1.50 \times 10^{-3}$ N m²).

Electrical characterization revealed that electrodes on the mMPC had relatively low impedance, which decreased as frequency increased (Fig. 2h and Supplementary Fig. 2c). Impedance measurements were taken across 7 electrodes in 2 mMPCs, ranging from 0.1 Hz to 100 kHz and revealed an initial mean impedance of 70.69–28.78 kΩ at 1–40 kHz (Fig. 2h). To improve the stability of the electrodes, we coated them with PEDOT on MPC electrodes using electrodeposition, as GaIn alloy at the electrodes might be degraded after culturing the hHO on the mMPC. PEDOT is a conductive polymer commonly used for preparing flexible electrodes due to its biocompatibility, conductivity, and stability. However, the conductivity of polymers is lower than that of metals. The current-time curve of the electrodeposition process showed an initial rapid increase in resistance (performing at a decrease in current value, Supplementary Fig. 2d), followed by a gradual decline as more PEDOT covered the electrode surface. The impedance of PEDOT-coated electrodes only increased to 95.83 kΩ at 40 kHz (Supplementary Fig. 2e). Furthermore, the impedance decreased slightly under 100% strain compared to that without any strain (Fig. 2h). The result can be attributed to the stretching force breaking the oxidation on the surface of GaIn particles, which allowed more liquid metal into the circuit to reduce the resistance. The average root mean square (RMS) noise of 32 electrodes at a single mMPC, was only 6.88 μV (Supplementary Fig. 2g), which was over 7 times lower than the range of neural spike (from ~50 μV to hundreds of microvolts). In summary, our mMPC presented excellent flexibility, stretchability, conductivity, and electrical stability, all crucial properties for a neuro-interface to acquire signals from neural organoids.

## The formation of hHOs

Researchers have proved that some hippocampal neurons, accompanied by the ChP and CH regions, could be derived from DMT organoids (Supplementary Fig. 3a, b)[29]. However, in an in vitro environment, the expansion of ChP tissues in DMT organoid tissues might affect the development of hippocampal tissue. Specifically, it has been observed that the TTR⁺ ChP tissue became more abundant while the neural tissue became smaller after long-term culture of the DMT organoids (Supplementary Fig. 3c–e). This result might be attributed to the enriched expression of BMPs in the ChP, which induced ChP epithelial fate and promoted its proliferation[34–36]. To generate hHOs from DMT tissue, alternative growth factors were considered to inhibit the ChP expansion.

Wnt3a and purmorphamine (the SHH signaling activator) were incorporated into the culture protocol to generate hHOs under the

hypothesis that Wnt3a and SHH weakened the expansion of ChP in DMT organoids and promoted the hippocampal fate[25,37]. Hence, some modifications were made to the culture protocol, including reducing the duration of BMP4 exposure from 4 days to 2 days and supplementing the culture medium with Wnt3a and purmorphamine after DMT induction (Fig. 3a). Fortunately, the resulting hHOs maintained their spherical structures without loose ChP tissues throughout the growth period (Fig. 3b). Key biomarkers of the development process are summarized in Fig. 3c[38–40]. The expression of FOXG1, LEF1, and PAX6 in day-30 hHOs indicated successful induction into the DMT stage (Fig. 3d). Furthermore, the hippocampus marker ZBTB20 expressed in the hHO and NESTIN⁺ neural stem cell and beta III-tubulin⁺ cell suggested neuronal differentiation within the hHO (Fig. 3d).

In the resulting hHOs, two important hippocampal progenitors, HOPX⁺ and PAX6⁺ progenitors[39], and ZBTB20⁺PROX1⁺ granule neurons were observed (Fig. 3e and Supplementary Fig. 4a). PAX6⁺ progenitors were even more early in the hHO (Fig. 3d). PROX1 induces neural progenitors to the granule cell fate, which is essential for developing the DG and adult hippocampal neurogenesis[41,42]. Furthermore, mature neuron markers TAU and MAP2 were expressed throughout the hHO (Supplementary Fig. 4b). Other hippocampal markers, SEMA5A (for DG) and SULF2 (for CA3 and DG) were also expressed in the hHOs (Supplementary Fig. 4c). These results suggested the presence of DG granule neurons within the hHOs. Cells dissociated from hHOs also expressed these markers (Supplementary Fig. 4d). Additionally, GFAP⁺ astrocytes and OLIG2⁺ oligodendrocyte progenitor cells were present (Supplementary Fig. 4e). These cells have been reported to play roles in maintaining the growth environment of neurons and support the signal transmission between nerves[43–45].

Immunostaining showed increased PROX1 expression in the more mature hHO (Fig. 3f). Along with the hHO growing, the ZBTB20 and PROX1 mRNA expression gradually increased (Fig. 3g). Wnt3a mRNA expression did not, however, consistently increase (Fig. 3g). Compared with day-45 DMT organoids, we found a significant increase in PROX1 mRNA expression and a significant decrease in TTR mRNA expression in the day-45 hHO (Fig. 3h). Taken together, these results demonstrated successful induction of organoids into a hippocampal fate through supplementing Wnt3a and SHH signals.

## Transcriptome analyses of hHOs

To further characterize cell identities, we performed droplet-based single-cell RNA-sequencing (scRNA-seq) to analyze the transcriptome of 7511 cells obtained from 4 organoids at day 70, and 6781 cells from 10 organoids at day 81, using 10x Genomics Chromium. The two samples were dissociated differently, and the specific process can be found in the methods section. By analyzing differentially expressed genes for each cluster, we further segregated cells into eight major groups, including excitatory neurons (ExN), inhibitory neurons (InN), immature neurons (ImmN), neural progenitor cells (NPCs), astrocytes, oligodendrocyte progenitor cells (OPCs) and oligodendrocytes, ChP cells and others (Fig. 4a, b and Supplementary Fig. 8a). Two samples of hHOs had similar cell types (Supplementary Fig. 5), but the hHO at day 81 was missing the population of cells co-expressing oligodendrocyte markers (Fig. 4a and Supplementary Fig. 8a, b). These identified groups closely resemble the cell types found in the human developmental hippocampus up to 22 gestational weeks (GW22)[39].

The ventral telencephalon markers NKX2-1, GSX1, GSX2, and LHX6, were not expressed in the cell clusters, suggesting that hHO was fully oriented toward dorsal telencephalic development during this period (Supplementary Fig. 6). Although the hHOs did not display an obvious presence of ChP epithelial cells, a small subset of cells expressed ChP markers (Supplementary Fig. 7a). We further examined the expression of TTR in cryosections of hHOs and observed TTR⁺ cells surrounding some cavities within the hHOs (Supplementary Fig. 7b). The result suggested that although Wnt3a and SHH inhibited the

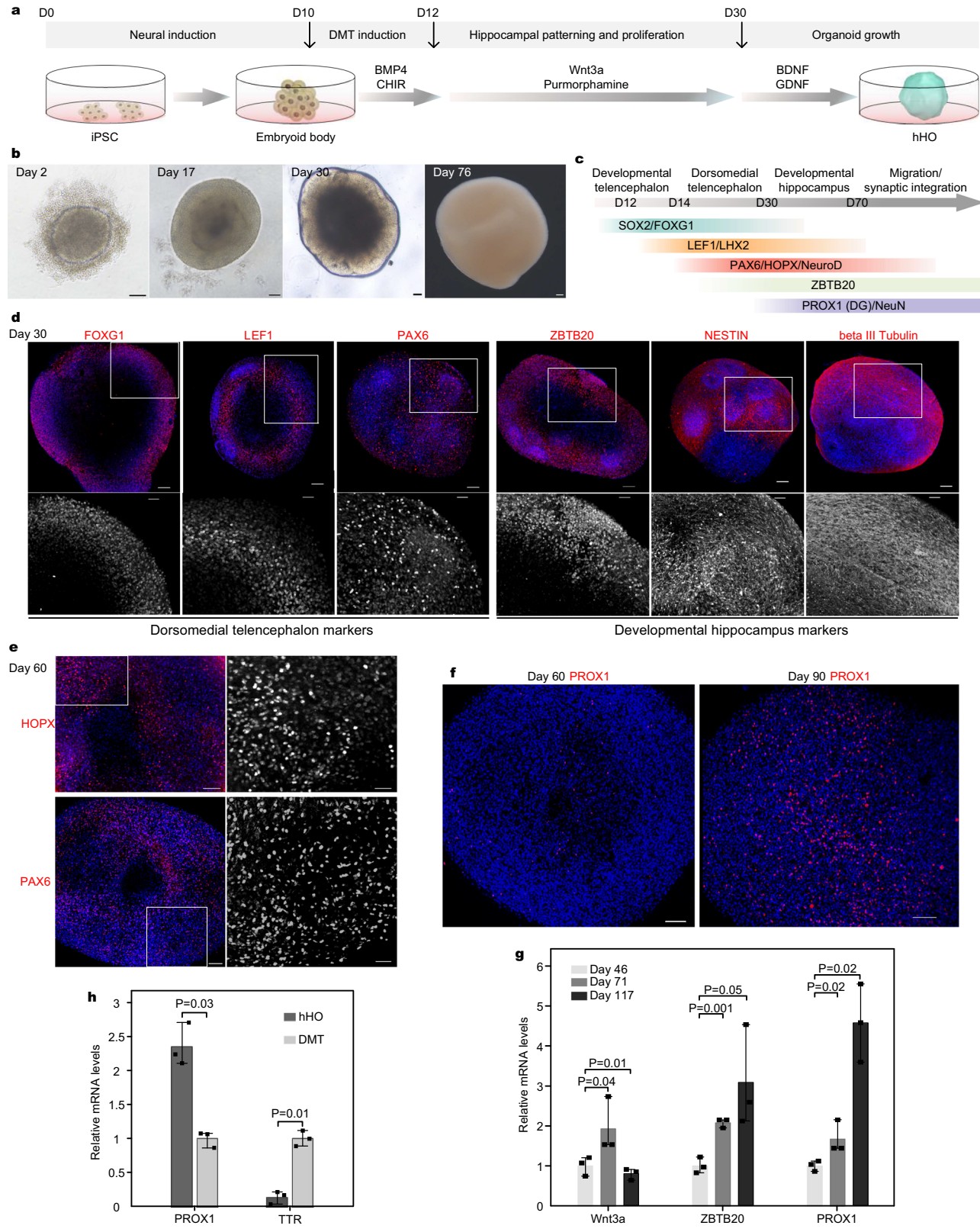

growth of ChP, it did not eliminate its presence. This might explain why a few hHOs developed cavities after being culture longer (Supplementary Fig. 7c), a phenomenon also observed in ChP organoids[46]. To confirm the hippocampal identity, we observed ZBTB20 expression in almost all cells (Fig. 4c). The ZBTB20+SOX2+ cells were clustered as immature cells (progenitors) and were further divided into HOPX+ and PAX6+ progenitor subgroups (Fig. 4c). Notably, some ZBTB20+PROX1+

cells were located in the cluster of excitatory neurons, suggesting the presence of PROX1+ granule cells in the hHOs (Fig. 4c). These results were consistent with the earlier immunofluorescence staining results in Fig. 3.

We integrated our two samples with a published scRNA-seq dataset of the developing human hippocampus[39] and compared cell types and their distributions (Fig. 4d and Supplementary Fig. 8c).

**Fig. 3 | Generation of hHOs. a** The overall strategy to generate hHOs. **b** The typical morphology of hHOs at different ages. Scale bar: 100 μm. **c** The significant markers in the developmental process of the hHO. **d** Staining images of day-30 hHOs. Upper: Lager-scale image of the whole hHO. Scale bar: 100 μm. Lower: Zoom-in greyscale view of the white box in the upper figures. Scale bar: 40 μm. **e** The hippocampal PAX6⁺ and HOPX⁺ progenitors in day-60 hHOs. Scale bar: (left) 100 μm and (right, zoom-in greyscale view of the white box in the left figures) 40 μm. **f** The hippocampal PROX1⁺ cells in day-60 hHO and day-90 hHO. Scale bar: 150 μm. **g** qPCR for

genes expressed in day-117 hHOs, day-71 hHOs versus day-46 hHOs (*n* = 3, three independent measurements using three independent samples, one-tailed *t*-test). **h**, qPCR for genes expressed in day-45 hHOs versus day-45 DMT organoids (*n* = 3, 3 independent measurements using three samples from different organoids, one-tailed *t*-test). Images in **d**–**f** are representative of *n* = 3 independent experiments. Exact sample size and values for **h** and **g** are provided in Source Data. Source data are provided as Source data files. hHO human hippocampal organoid, DMT dorsomedial telencephalon.

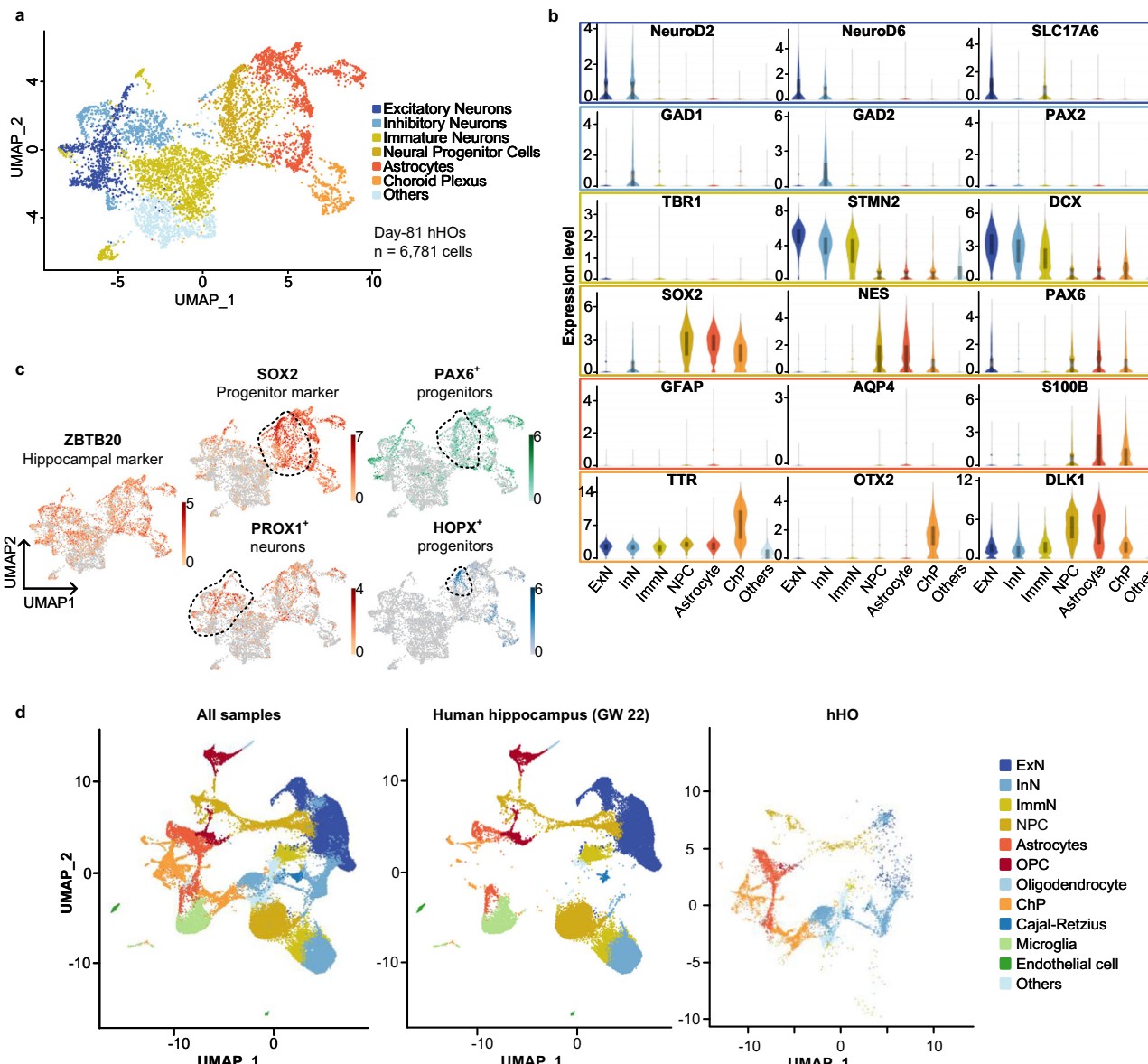

**Fig. 4 | Transcriptomic signature of hHOs. a** UMAP visualization of 7 scRNA-seq clusters from day-81 hHOs. The cluster of oligodendrocytes and oligodendrocyte progenitor cells was presented in day-70 hHOs dataset in Supplementary Fig. 8a–c. **b** Violin plots of expression levels of markers in six clusters. **c** Feature plots of hippocampal ZBTB20⁺ cells, SOX2⁺HOPX⁺ progenitors, SOX2⁺PAX6⁺ progenitors

and PROX1⁺ neurons. **d** UMAP visualization of the integrated dataset of the human hippocampus in GW22 and day-81 hHOs. (Left) All samples dataset. (Middle) The human hippocampus. (Right) Day-81 hHOs. The distribution of day-70 hHOs was present in Supplementary Fig. 8c. hHO human hippocampal organoid, GW gestational weeks.

Both had a similar distribution of cell clusters, but the sample in vivo had some unique ImmN and InN, in addition to separate cell populations of endothelial cells and microglia. The absence of endothelial and microglial cells, differentiated from mesodermal cells, was reasonable. This result also suggested that the in vitro differentiation of the hHO needed to be more finely regulated and even co-

cultured with other cell types to more closely resemble the in vivo hippocampus. Cell clusters of day-81 hHOs were missing oligodendrocytes, consistent with Fig. 4a, b. We identified a subgroup of cells as 'others' that we could not define based on their differentially expressed genes. The 'other' group mainly presented in hHOs samples, close to the neuron clusters. The single-cell trajectory analysis

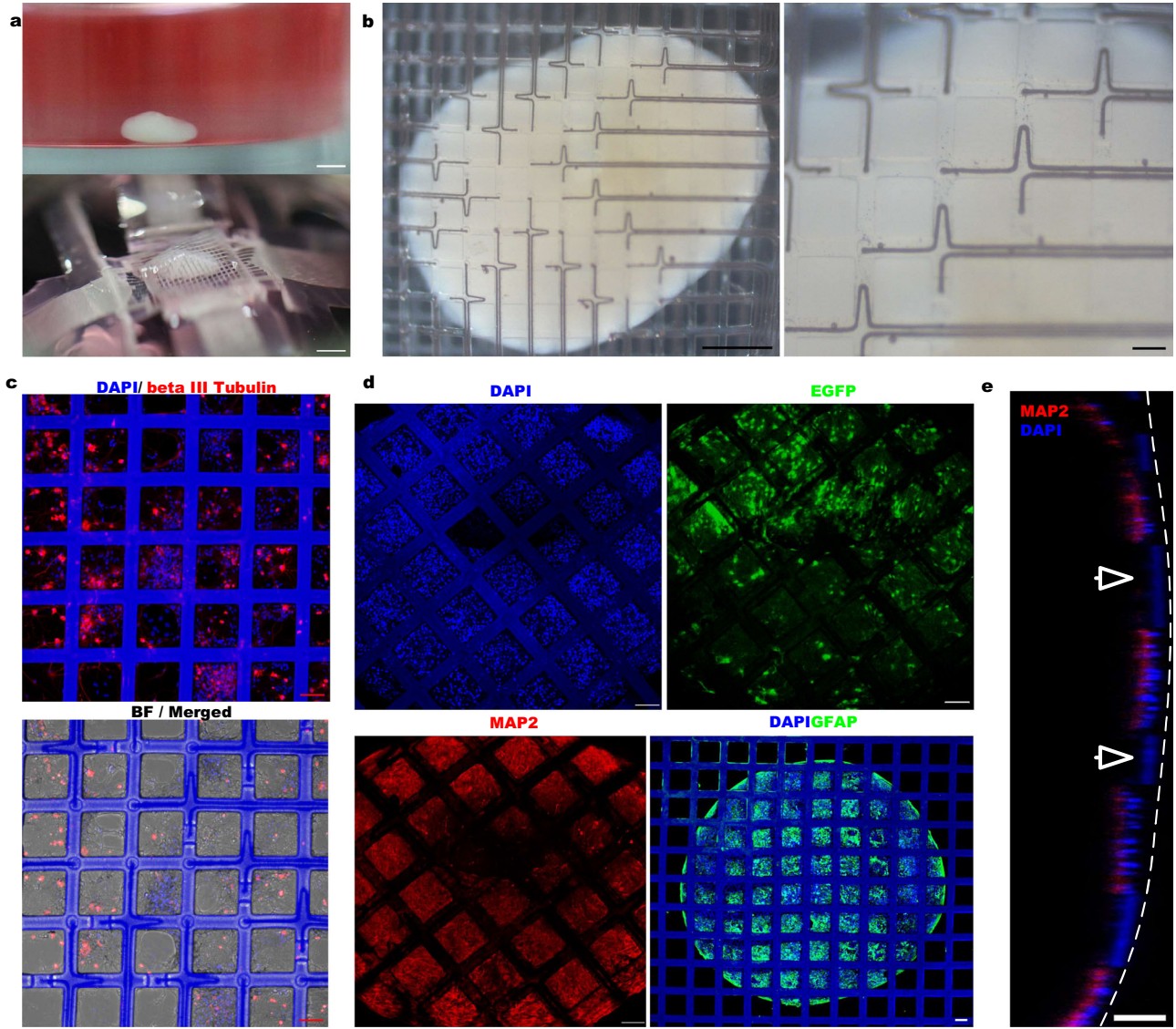

**Fig. 5 | The hippocampal cyb-organoid platform. a** Integration of the 128-channel mMPC and the hHO. (Upper) The hHO was stayed in a low-attachment dish. (Lower) The hHO was enveloped in two-layer mMPC. Scale bar: 1 mm. Notably, there was no significant deformation when the hHO was inserted between two layers of mMPC. **b** The bottom view captured by a microscope. Scale bar: (Left) 500 μm, (right) 100 μm. **c** Confocal images of cells from suckling mice hippocampus culturing in the mMPC for 30 days. Scale bar: 100 μm. **d** Confocal images of MAP2⁺ neurons, GFAP⁺ astrocytes and EGFP⁺ cells in the cyb-organoid platform. Scale bar: 100 μm. **e** The cross-sectional view of the attachment between the mMPC and the hHO. The arrowhead points to the mMPC. Scale bar: 100 μm. Images in **c**−**e** are representative immunofluorescence staining cross-independent experiments: **c** (n = 4), **d** (n = 3), and **e** (n = 3).

also indicated that the identified cells in the 'others' subgroup were located at a separate termination of neuron branches that began from the NPCs (Supplementary Fig. 8d). They did not occur in the fetal hippocampus. The imperfect in vitro culture might lead to other neuronal cells in the hHO that did not belong to the hippocampus. But, more experiments will be required to confirm the causal relationship. Combining the immunofluorescence and scRNA-seq datasets, our results indicated the successful generation of hHOs.

## The hippocampal cyb-organoid platform

To enable the integration of the mMPC with the hHO and facilitate the connection with the Plexon instrument, we assembled the mMPC and culture dishes. To prevent the two-layer meshes from severely squeezing on the soft hHO, the bottom mMPC was designed as a 'bowl' to hold the hHO when we assembled this device (Supplementary Fig. 9a). We placed a TPU ball of ~3 mm when assembling the device to

reserve the space the hHO may need (Supplementary Fig. 9a, b). After sterilization, the cultured medium was added to the dish, and the TPU ball was removed from the side; the hHO was inserted into the space between the top and bottom mMPCs. The oblique view and the side view showed that the top mMPC varies with the undulation of the hHO surface without causing significant compression on the hHO and the hHO assembly (Fig. 5a, b and Supplementary Fig. 9b). More details about the process can be found in Supplementary Movie 3. Comparisons before and after the attachment of the top mMPC showed that the top mMPC could slightly squeeze the hHO, resulting in an area enlargement of <5% in the bottom view and a reduction of <5% in the side view (Supplementary Movie 4 and Supplementary Fig. 10a–c). Micro-computed tomography (micro-CT) images showed the hHO changes in height within 1% compared to its suspended state (Supplementary Fig. 10d and Supplementary Movie 5). This squeeze did not severely affect the morphology of the hHO or cause damage to the hHO.

Because of the concern about the potential risk of leakage of liquid metal on the electrode locations and possible toxicity, we cultured cells from suckling mice hippocampus on the mMPC to verify its long-term biocompatibility. After placing the mMPC in a Petri dish, cells could proliferate and grow stably on the mMPC for more than 30 days (Supplementary Fig. 9c). The cell density was not significantly different from that of the control group without the mMPC (Supplementary Fig. 9d). A large number of axonal structures were present at electrode spots (Fig. 5c), indicating that the mMPC possessed excellent biocompatibility.

The mMPC offered distinct advantages over the 2D MEA. Specifically, the two-layer mMPCs supported hHOs directly, while fixing hHOs on the 2D MEA was challenging. Additionally, we observed that cells from the hHO gradually migrated in the 2D MEA, leading to a contribution to the signal from these migrated cells rather than the neural network of hHOs (Supplementary Fig. 11a). In contrast, without coating poly-L-lysine (PLL) -laminin, the mMPC allowed for the clear boundary of the hHO to be maintained for 30 days without significant migration of cells away from the hHO (Supplementary Fig. 11b). Cell migration also occurred in the PLL-laminin-coated mMPC, leading to fusion locally between the hHO and the mesh structure (Supplementary Fig. 11b).

That transparent mMPC facilitated direct imaging by confocal microscopy. Figure 5 shows confocal microscope images of hHOs enveloped inside the mMPC. The attachment of the mMPC to the surface of the hHO, spanning most of its surface without any noticeable tissue damage, was highlighted in Supplementary Movie 4 and the cross-sectional view in Fig. 5e and Supplementary Fig. 11e. Fluorescence immunostaining of the hHO which was removed from the mMPC revealed no obvious tissue or cellular damage on the hHO surface (Supplementary Fig. 11d). In addition, the deposition of PEDOT was critical for long-term integration. In the control group, electrodes without PEDOT coating displayed gradual degradation of GaIn alloy within 2 weeks, resulting in transparent electrodes in bright-field images, whereas PEDOT-coated electrodes maintained their original appearance (Supplementary Fig. 11c).

## Electrical activity recording through the mMPC

Spontaneous spikes from hHOs were successfully recorded across the mMPC (Fig. 6a, b). The uniform spike waveform with an amplitude of ~50 to 150 μV and an average duration of ~800 μs, was similar to the waveform of extracellular action potential (Fig. 6c). Furthermore, the same electrode could capture multiple signals, and 2–4 spike types were displayed after sorting (Fig. 6d).

A similar signal was observed using the 2D MEA (Supplementary Fig. 12a). After placing hHOs on the 2D MEA for ~3 days, cells migrated away from the hHOs and formed a monolayer on the MEA surface (Supplementary Fig. 11a), and some neural signals were detected from this area. The higher spike rates were concentrated on these electrodes covered by the hHOs (Supplementary Fig. 12b, c). These results suggested that neurons in tissues might maintain higher nerve activity than monolayer neurons. Even after removing the data of the extended nerves which were not integrated into the hHO, we compared the spike rates detected by the electrodes covered by hHO in the 2D MEA with the mMPC, showing more active neural activities in the mMPC (Fig. 6e). The comparison highlighted the necessity to develop 3D MEAs.

In a series of experimental tests, we detected signals from the single hHO or hHO-fused assemblies, and the number of active channels was listed in Supplementary Table 1. The number of active channels was mainly affected by the hHO size. For signal acquisition of an individual hHO, we detected signals in a maximum of 54 channels, whereas when we used fused assemblies of 2 hHOs, the maximum number of active channels increased to 85. Simultaneous signals always appeared in pairs in two channels. We found that the time

interval of these paired signals was <1 ms. They always appeared one after the other (Fig. 6f). This observation suggested that neurons on the hHO formed functional synapses capable of transmitting signals. In addition, neural activity was detected on the hHO (Fig. 6g). As potential evidence of neural network maturation, the synchrony was proposed to reflect a balance between ExN and InN and coordinate neuronal communication in the hHO[47]. We further confirmed the capability of mMPCs by examining the signal of hHOs response to 10 mM glutamine, a precursor of the neurotransmitter glutamate. The spike raster before and after the supplement of glutamine showed a significant glutamine-induced increase in spikes (Fig. 6h). We found no significant difference in spike amplitude (only a slight increase in the average spike waveform, +6.23 μV, Supplementary Fig. 13a). Electrophysiological activity was distributed throughout the 128 channels and was able to be visually represented in 3D plots based on the location of the electrodes (Fig. 6i and Supplementary Movie 6). These visualizations have the potential for future spatial and temporal mapping of electrical activities across neural organoids.

Concerned that pressure from the top mMPC could affect the electrical activity of the hHO, we compared the neural signals before and after cutting the top mMPC. A comparison of the average spike rates of multiple active channels over 2 min showed no significant difference, with only a few channels showing slightly higher spike rates (Supplementary Fig. 14a). However, as in the absence of top mMPC, we placed the hHO at the bottom mMPC directly and found that only a few electrodes were able to detect the signal when the hHO was sunk onto the bottom mMPC by its gravity alone (Supplementary Fig. 14b). When the top mMPC was used simultaneously, more electrodes at bottom mMPC detected the signal. Pressure from the top mMPC can improve the contact between the hHO and the mMPC, allowing more channels to work. Following one month of integration, we measured the electrode impedance after removing the hHO from the mMPC. At 40 kHz, impedance increased from 95.83 to 196.03 kΩ compared with the initial stage (Supplementary Fig. 13b). This increase in impedance might have resulted from the degradation of some GaIn alloy from the electrodes, leading to a reduction in the conductivity.

## Discussion

Our protocol to generate hHOs revealed that the Wnt3a and SHH signals promoted hippocampal fate while inhibiting the ChP expansion in vitro. These hHOs contained a significant number of hippocampal progenitors and DG granule neurons, with a high correlation of transcriptomic signature to the human in vivo hippocampus. Our fabrication process of the mMPC is easy-to-operate and low equipment-dependent using low-cost materials. The mMPC offered multichannel recording, enabling the acquisition of as many nerve signals as possible from neural organoids. Our mMPC detected neural spike, synchronization, and oscillation activities in the hHO and the hHO assembly. These results indicated that hHOs grow a complex neural network.

There are still some limitations of the current cyb-organoid. The free-floating status of all neural organoids poses great challenges for designing electronics and integrating electronics with organoids. Even if the hHOs can maintain its morphology in the mMPC for a long time, this fixation of hHO in the same position must be detrimental to its growth compared to the culture condition in a bioreactor. The decreasing medium used when recording signals also affects organoid growth. The medium diffusion in the system is not as good as the flow medium, but this problem may be improved if a flowing medium system can be provided for the cyb-organoid.

Although the sandwich structure of our cyb-organoid greatly increased the channel number, this configuration might result in a slight pressure on the hHO. In order to make the top mMPC attach to the hHO, we would reduce the culture medium to only submerge a little past the surface of the hHO. The comparison of the signal

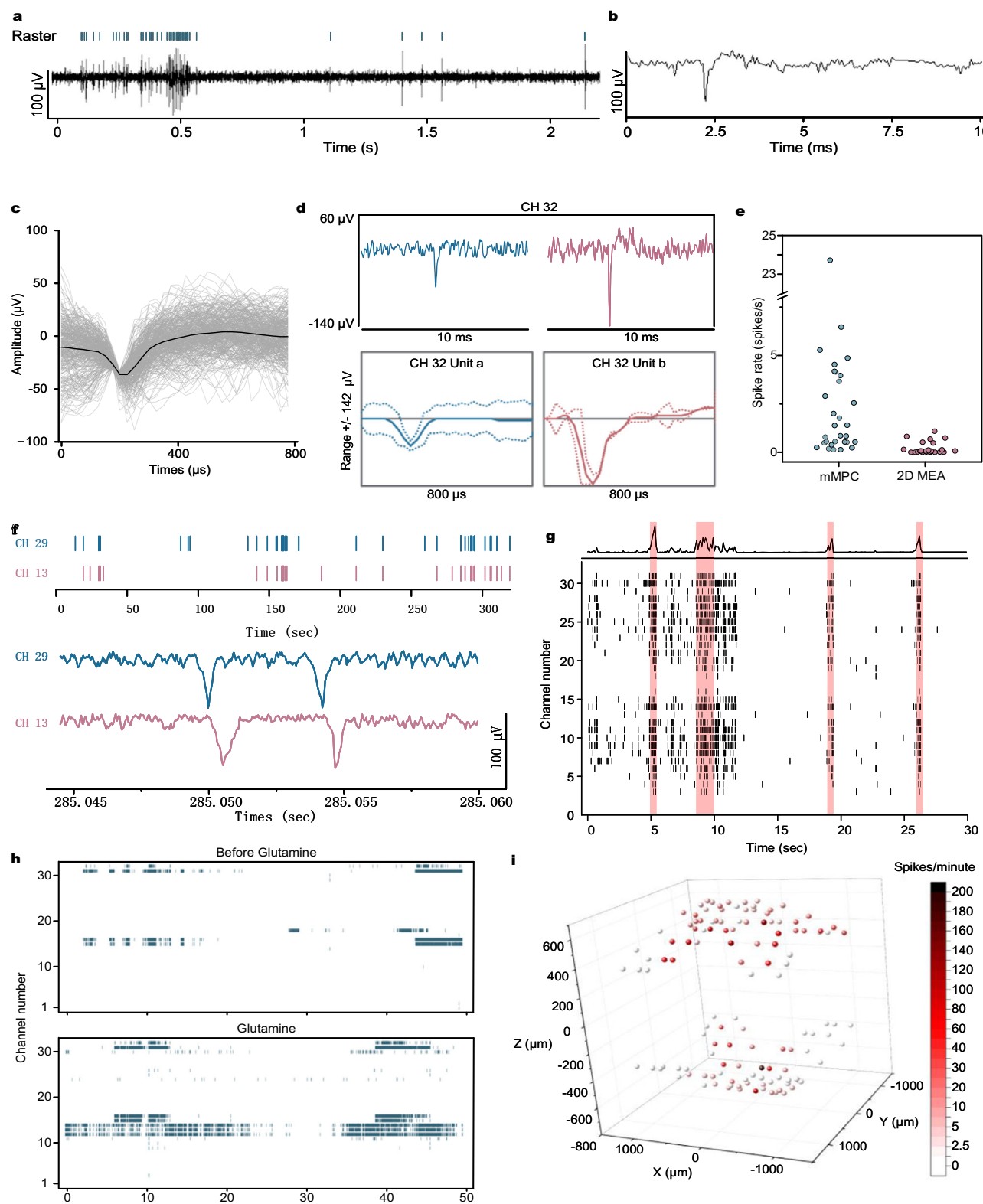

**Fig. 6 | Electrical activity recording. a** (Upper) Raster plots of neural spikes, and (lower) the continuous signal in a channel of the mMPC. **b** Continuous recording for a spike. **c** The uniform spike waveforms and the average waveform in a channel of the mMPC. **d** Two different spikes appeared in the same channel (upper) and the sorting results (lower). **e** The comparison of spike rates detected by the mMPC and the 2D MEA (these channels covered by the hHO). **f** The synchronous signals between two channels. (Upper) Raster plots of neural spikes in these two channels and (lower) continuous recording for two spikes. The time interval was <1 ms. **g** The oscillatory network activity of the hHO. (Upper) the population activity histogram and (lower) the raster plot (black) identified bursts of spiking detected on 32 channels and coordinated bursts (red) of activity across channels. **h** Raster plots before and after Glutamine supplement. **i** 3D visualization plot showing average spike numbers throughout 128 channels. mMPC mesh liquid metal-polymer conductor, MEA multi-electrode array.

recordings with and without the top mMPC illustrated that this squeezing improved the contact between the hHO and the electrodes. At the same time, however, it was obvious that the signals detected in this case were not exactly the spontaneous activity of the hHO in a free state but rather the neural activity under a slight squeeze. This limitation may cause some bias in our understanding of neural activity. In addition, this configuration leaves a distance between the two layers of mMPC and does not cover the complete surface of the organoid. In future work, we will focus on the problem by changing device geometry and using softer materials.

Compared with mature commercial systems, such as MaxWell Biosystem, HyperCAM Alpha, and Axion system, which offer an integrated platform with high-density electrodes in multiwell, real-time high-resolution scanning, and controlled environment, there is still much room for developing the mMPC. The ability of our Plexon instrument to detect only 32 channels at a time limits the mMPC to record the complete neural activity of the hHOs. In addition, obtaining long-term neural activity from neural organoids requires a stable and healthy culture environment. However, there are many challenges in upgrading a system, including instrumentation, stimulation modules, temperature regulation, and gas regulation. Adding other modules also introduces noise to the signal acquisition. These will be explored in the future. More importantly, we relied more on the mMPC's softness to allow for non-invasive signal detection, although the mMPC has great stretchability. We are concerned that a stretched device would damage the tissue for a long time. Thus, our next plan is to find a suitable material with plastic deformation properties to replace the TPU and PU we are currently using, and we hope that the next device will be able to grow with the brain tissue without damaging the brain tissue. In this way, we will be able to monitor the growth and maturation of brain tissue in situ for a long time.

The mMPC is the first 3D MEA system for detecting neural signals in hHOs. However, the mMPC is not limited to the hHO, and other neural organoids and heart organoids can be detected. Our results suggested that the electrical activity of neurons in tissue blocks was more active than that in monolayer neurons, probably due to the more suitable environment for neurons provided by tissue with multicellular types. As more region-specific brain tissues generate in neural organoids, corresponding mMPCs can be configured accordingly, and their flexible design will enable them to be adapted to other shapes. Future research will focus on remodeling the classical unidirectional projections of the hippocampus, such as the olfactory cortex projecting to the DG and the DG projecting to the CA3. Assembling neural organoids to form these specific projections and studying the signal transmission on the mMPC system will provide a powerful platform for understanding neural circuits in the hippocampus.

## Methods

### Fabrication of the mMPC

The graphic patterns were created in AutoCAD software, and subsequently, photomasks were produced based on these designs in Shenzhen Qingyi Photomask. The preparation of PDMS patterns was completed through the soft lithography technique. This process involved several crucial steps, starting with the fabrication of molds through photolithography. Specifically, the process involved spin-coating (Spin coater, AC-200SE, China) a thin layer of SU-8 negative photoresist (SU-8 2015) on a clean silicon wafer at a speed of 3000 rpm for 30 s. The thin layer was then cured at 65 and 95 °C for 3 min each (Hot plate, KW-4AH-350, America), followed by exposure to 365 nm UV at a dosage of 140 mJ/cm² (MASK ALIGNER MA6/BA6, SUSS, Germany). The thin layer was cured at 65 and 95 °C for 3 min each and developed using SU-8 developer for 3 min. The developed mold was cured at 150 °C for 30 min. Finally, the PDMS mixture (10:1) was poured on the

silicon mold and cured at 80 °C for 1 h to produce a PDMS slab. The PDMS patterns were obtained by peeling the slab from the molds.

To prepare the MPC ink, 6 g GaIn alloy (75 wt% Ga and 25 wt% In) was sonicated in 1 mL n-Decyl alcohol for 1 minute at a power of 300 W. The GaIn alloy underwent a transfer process to produce micro- or nano-particles. The resulting MPC ink was then scraped into the microchannels of PDMS slab #1 to create MPC circuits and electrodes. Any residual ink on the PDMS surface was carefully removed using tape. Subsequently, PDMS slab #1 was incubated at 80 °C for 5 min to vaporize n-Decyl alcohol. PDMS slab #1 was then immersed in a TPU solution (3 wt%) dissolved in Dimethylformamide (DMF) and vacuumed for 30 minutes. After removing PDMS slab #1 from the solution, it was allowed to drip dry completely, ensuring that the TPU solution only remained in the gaps between the GaIn particles. DMF was allowed to evaporate after PDMS slab #1 was cured at 60 °C for 20 min. This process resulted in embedding the GaIn particles in a TPU matrix, and the MPC circuits became conductive when tensile stress from TPU broke the oxide layer of the GaIn particles.

PDMS slab #2 (for the TPU substrate layer) was aligned with PDMS slab #1 under a stereo microscope. Subsequently, the TPU solution was carefully dropped on one side of PDMS slab #2, and the TPU solution (3 wt%) flowed into the mesh microchannels inside PDMS slab #2 due to capillary force without applying pressure. The lower the TPU solution concentration, the faster the inflow. A higher concentration of TPU could also be used, but it needs vacuuming to help the TPU solution flow into the mesh microchannels quickly. After the complete evaporation of DMF, the TPU film, along with PDMS slab #2, was peeled off PDMS slab #1. This process was followed by the alignment of PDMS slab #3 (for the PU insulating layer), the dropping of PU solution (5 wt %, dissolved in ethanol), and the subsequent evaporation of ethanol to complete the TPU-PU encapsulation. Upon the gentle removal of all PDMS slabs, the mesh structure was released, and a flexible printed circuit (FFC, 32p) was connected to MPC circuits using an anisotropic conductive tape.

### Electrodeposition of PEDOT:PSS

The electrodeposition of PEDOT was carried out in potentiostatic mode using an electrochemical workstation (Metrohm autolab, M204). The PEDOT electrolyte was prepared by dissolving 100 mM sodium sulfate, 15 mM EDOT, and 0.315 mM PSS in deionized water. Before the deposition process, the mMPC was horizontally placed on a PDMS or glass sheet, treated with oxygen plasma, and soaked in PBS under vacuum for several minutes. This step ensured proper contact between the electrolyte and the MPC electrodes. The mMPC was then attached to the cathode, and a platinum electrode was attached to the anode of the electrochemical workstation. After replacing PBS with the electrolyte, PEDOT was deposited at a constant voltage of 1.2 V for 5 min. The mMPC was subsequently cleaned with PBS after electrodeposition.

### Mechanical characterization

The mechanical properties of the mMPC were evaluated on a padded board with a scale to facilitate real-time calculation of tensile degrees. To begin the test, the mMPC was placed flat on a PDMS slab, and the tape was attached to one end. The other end of the mMPC was then taped and stretched to 400–500% before being secured to the PDMS slab. Finally, the assembly was placed under a microscope for observation. Videos of the stretchability were recorded using a camera. The mMPC was repeatedly subjected to stretching and twisting by manually pulling both ends of the mMPC. The bending test was performed at Shiyanjia Lab (www.shiyanjia.com) using a three-point bending method in an electronic universal testing machine. The bending stiffness was calculated using the following Eq. (1) (We assumed the device was a completely thin sheet to replace the mesh shape for calculations

and that the cross-section was rectangular):

$$\text{Bending stiffness}: B.S. = EI = \frac{FL^3}{4bh^3\delta} \cdot \frac{bh^3}{12} \tag{1}$$

$F$ is the average force applied to the mMPC when the mMPC fractures (1.73 N); $E$ is the modulus of elasticity, and $I$ is the area moment of inertia; $L$ is the distance between the two support points (124 mm); $b$ is the width of the section (4.54 mm); and $h$ is the thickness of the section (0.03 mm); $\delta$ is the deflection, we used the average maximum deflection (45.93 mm). The bending stiffness is calculated as $1.50 \times 10^{-3}$ N · m$^2$.

## Electric characterization

The electrical properties of the mMPC were characterized by measuring the resistance and impedance. To determine the conductivity of the mMPC at 400% strain, an ohmmeter was used to measure the resistance before and after stretching (Supplementary Fig. 2f). The conductivity was calculated using the following Eq. (2), taking into account the length ($L$: 1 cm), width ($W$: 5 μm), thickness ($H$: less than the initial thickness of 15 μm), and measured resistance ($R$: 283.8 ohms, Supplementary Fig. 2f):

$$\text{Conductivity } \sigma = \frac{L}{RWH} \tag{2}$$

The calculated conductivity value was above 4000 S/cm, indicating that the mMPC has excellent conductivity even under significant strain.

The impedance measurements were carried out on the electrochemical workstation (Metrohm autolab, M204) using the same preparation process as that for PEDOT deposition. The impedance measurements were taken in 1x PBS with a constant voltage of 0.01 V in the frequency range of 0.01 Hz–100 kHz.

## Culture of hiPSCs

HiPSCs were used in accordance with the local ethical regulations. Details of the hiPSC lines used in the study are provided in the Key resources table in the Supplementary Information file. The lines were authenticated through STR profiling. HiPSCs were cultured under feeder-free conditions at Matrigel-coated six-well plates using a mTeSR medium. The culture medium was changed daily, and hiPSCs were passaged when they reached 80–90% confluency (5–7 days). To detach the hiPSC colonies, they were washed with DPBS and incubated with ReLeSR reagent at room temperature for 1 min. After aspiration of ReLeSR, hiPSCs were incubated in a humidified CO$_2$ incubator at 37 °C for 5–7 min. The small hiPSC clusters were obtained by detaching the colonies with wide-bore tips and seeded on fresh Matrigel-coated six-well plates.

## Generation of hHOs

The hiPSCs colonies were dissociated using Accutase when they reached 80–90% confluency. Briefly, the hiPSCs were washed with DPBS, and then they were incubated with Accutase at 37 °C in a humidified CO$_2$ incubator for 10–15 min. The reaction was then neutralized by adding DMEM/F-12 medium. The resulting single-cell suspension was collected and centrifuged at 200×$g$ for 4 min. After removal of the supernatant, the hiPSCs were resuspended, and 15,000 cells were transferred to each well of a low-attachment U-bottom 96-well plate with neural induction medium (150 μL per well; Embryonic body formation).

The neural induction medium contained DMEM/F12 medium with 10% KnockOut Serum Replacement, 1X GlutaMax, 1X MEM-NEAAs, 1X Penicillin–Streptomycin, 50 μM 2-mercaptoethanol, 10 μM SB 421542, 2 μM XAV-939, 100 nM LDN-193189, and 1 μM cyclopamina,

supplemented with 5% FBS and 50 μM Rock inhibitor on day 0. On day 2, half of the medium was replaced with 150 μL fresh neural induction medium with 10 μM Rock inhibitor, but without FBS. From day 4 to day 10, the neural induction medium was changed every other day.

From day 10 to day 20, embryonic bodies grew in differentiation medium 1 consisting DMEM/F-12:Neurobasal (1:1), 1% B27 (without Vitamin A), 0.5% N2 supplement, 1X GlutaMax, 1X MEM-NEAAs, 1X Penicillin–Streptomycin, 50 μM 2-mercaptoethanol. 15 ng/mL BMP4 and 3 μM CHIR 99021 were added to the differentiation medium 1 from day 10 to day 12 (DMT induction) but 20 ng/mL Wnt3a from day 12 to day 20 (Hippocampal primordium induction). The culture was switched to differentiation medium 2 from day 20 to day 30, consisting of the same formula as differentiation medium 1 except for replacing B27 (without Vitamin A) with B27. Wnt3a at 20 ng/mL and 1 μM purmorphamine were added to the differentiation medium 2. Organoids were maintained with the media changes every other day. At the maturation stage, hHOs were transferred to low-attachment 6-well or 24-well plates with the differentiation medium 2, supplemented with 10 ng/mL BDNF and 10 ng/mL GDNF. Half of the medium was changed twice per week.

## Single-cell RNA sequencing

**Sample preparation of day-70 hHOs.** For minimizing information loss and capturing more cells from hHOs, four hHOs were dissected into smaller tissues and replated on a PLL-laminin-coated flask. Briefly, hHOs were incubated with papain enzyme at 37 °C for 20–30 min, and then papain enzyme was gently removed. The hHOs were dissected into fragments using tweezers on the differentiation medium 2 and transferred into PLL-laminin-coated flasks. The fragments were cultured for 3–4 days to allow for cell migration. The single-cell suspension of ~200,000 cells was obtained by dissociating cells via papain enzyme on the flasks, and the resulting single-cell suspension had a viability of ~4.8%.

***Sample preparation of day-81 hHOs.*** Ten hHOs were dissociated using the Pierce Primary Neuron Isolation Kit (Thermo Fisher). Briefly, 2–3 hHOs were placed into microcentrifuge tubes and washed with ice-cold HBSS. Then, HBSS was removed gently, and per 0.2 mL papain enzyme was added for 2–3 hHOs. We incubated them at 37 °C water bath for 20–30 min, with manual shaking. Then papain enzyme was gently removed and washed hHOs with HBSS. The differentiation medium 2 was added, and the hHOs were broken up by pipetting up and down 15–20 times. The pre-sorting of live cells was completed using magnetic-activated cell sorting. The single-cell suspension of ~67,800 cells was obtained, and the resulting single-cell suspension had a viability of ~78.02%.

The scRNA-seq and data analysis were performed at Genergy Biological Technology Limited Co. (Shanghai, China). The 3′ Gene Expression Library was prepared with the quality-qualified cDNA based on Chromium Next GEM Single-Cell 3′ GEM protocol. The quality of the scRNA-seq library was assessed according to the standard protocol after fragmentation, adaptor ligation, and sample index PCR. Cluster cell identity was determined manually using the highly differentially expressed and known marker genes. Loupe produced visualization via UMAP and violin plots.

## Integration of the mMPC and hHOs

The mMPC was fixed in a 100 mm dish with a 35 mm confocal dish in which the glass bottom was removed, leaving the end of flexible printed circuits outside the 100 mm dish. When assembling the two-layer mMPC, a TPU ball was used to pre-simulate the replacement of the hHO and place it between the two layers. All relevant materials, including tapes, dishes, and devices, were sterilized by overnight UV light illumination. Before coculturing with the hHO, the mMPC was soaked in PBS to promote electrode and medium attachment. After

removing the TPU ball from the side between the top and bottom meshes, the hHO was inserted into the two-layer mMPC, replacing PBS with 2.5 mL differentiation medium 2 (supplement with BDNF and GNDF) in the 35 mm dish. When the medium was enough to submerge the mMPC completely, the upper mMPC would float up to create a gap between the two layers. The hHO was then aspirated using a wide bore tip and gently placed in the gap. In daily cultures, to maintain the hHO in a healthier state, a sufficient amount of medium would be provided so that the top mMPC would not squeeze the hHO. With enough medium, the hHO could be maintained with the mMPC system long-term in a 37 °C incubator. When detecting neural signal, a small amount of medium was carefully removed until the upper layer adhered to the hHO.

### Detection of electrophysiological signal
The neural signal in the 2D MEA was acquired by the Maestro Pro commercial system (Axion Biosystems, Inc.). The hHO was seeded on 6-well CytoView MEA plates containing 64 electrodes coated with PLL-laminin. The hHO was seeded in advance for 2 weeks for adhesion before recording neural activities. During this period, hHOs were cultured in the differentiation medium 2 supplemented with BDNF and GDNF. Before recording spontaneous spikes, the Axion instrument was allowed to stabilize the environment for 30 min without any stimulation. The AxIS Navigator performed spike waveform and network bursts. Data were analyzed through the companion software NeuralMetric Tool.

Signal acquisition by the mMPC was performed on the OmniPlex Neural Recording Data Acquisition System (Plexon Inc.). Our mMPC was connected to the head stage of the system via an FFC adapter. The Plexon system recorded data from 32 electrodes simultaneously, with a sampling rate of 40 kHz. The threshold for spike detection was set above 5 standard deviations from the average noise level, with a bandpass filter between 200 and 6000 Hz. The window length to isolate spikes was 800 μs. The neural spike sorting, analysis, and result presentation were done in Plexon companion softwares, Offline Sorter, and NeuroExplorer.

### Isolation and culture of cells from hHOs
To isolate neurons from hHOs, the neuronal isolation enzyme (papain) was utilized. In brief, two or three hHOs were collected into a 1.5 mL tube containing 1 mL of ice-cold HBSS. After removing the HBSS, 200 μL of neuronal isolation enzyme was added and incubated at 37 °C in an incubator for 30 min. The enzyme was then gently removed, and the hHOs were washed with pre-warmed HBSS. Subsequently, the hHOs were dissected into fragments on differentiation medium 2, and the fragments were transferred into PLL-laminin-coated six-well plates to allow for cell migration. The culture medium was replaced every three days, and after a few days, monolayer cells were formed, which could be passaged or dissociated using the papain enzyme.

### SEM imaging
The scanning electron microscope was applied to characterize the mMPC (Hitachi-SU8220, Japan; ZEISS Merlin, Germany). The mMPC was laid flat on the conductive tapes and sputter-coated gold on the surface for 30 s.

### Immunofluorescent staining of the organoid, cryosections and monolayer cells
The hHO was collected in 1.5 mL tubes using wide-bore tips and fixed in 4% paraformaldehyde for 1 h at room temperature. After washing with PBS, the hHO was permeabilized in 0.1% Triton X-100 in PBS on a shaker for 1 h at room temperature. Subsequently, the hHO was blocked in a blocking buffer consisting of 3% bovine serum albumin and 0.1% Triton X-100 in PBS for 1 h at room temperature. Primary antibodies were diluted in the blocking buffer and incubated with the

hHO overnight at 4 °C. Following PBST washing, secondary antibodies were incubated with the hHO overnight at 4 °C.

To prepare cryosections, the hHO was incubated with 30% sucrose after fixation and washing, then stored at 4 °C until complete infiltration. The hHO was embedded in O.C.T. in plastic cryomolds and stored at −80 °C until making cryosections. Cryosections were made at 25 μm using a cryostat (Leica CM1950), melted on glass slides, and outlined with a hydrophobic pen.

The staining for cryosections and monolayer cells was similar to the process above but decreased the time, 15 min for fixation, 30 min for permeabilization, and 1 h for blocking. All the samples were washed with PBS after completing the secondary antibody staining and stored in PBS before imaging.

### Acquisition and processing of images
Fluorescence images were obtained by confocal microscopy (Nikon Confocal A1R with FLIM; ZEISS LSM 980, Germany) and analyzed by Imars viewer. Optical photographs were obtained with the camera and microscopes. We processed images with ImageJ.

### Quantitative real-time PCR
qPCR (5–10 organoids per sample) was performed using Taq Pro Universal SYBR qPCR Master Mix (Vazyme Biotech Co., Ltd.). The cycle threshold (CT) values of samples were collected for statistical analysis. The data were normalized to the GAPDH expression. Primers were listed in Supplementary Information. Data processing and hypothesis testing were completed in Origin.

### Micro-CT scanning
Micro-CT images were scanned in Skyscan 1276 (Bruker Corporation). The hHO was inserted in the mMPCs assembled in a 35-mm glass bottom dish. The micro-CT scanning was performed after the top mMPC attached the hHO. The data was analyzed by CT-Analyser and CTvox.

### Statistics and reproducibility
The specific sample size and data values for all performed analyses are reported in the Source Data file if not stated in the figure legends. Each batch is defined as an independent experiment. Each sample from different organoids is defined as an independent experiment.

### Reporting summary
Further information on research design is available in the Nature Portfolio Reporting Summary linked to this article.

## Data availability
The processed 10X Genomics datasets generated in this study have been deposited in the Gene Expression Omnibus (GEO) database under accession code GSE264363. All other data generated in this study are provided in the Supplementary information files or in the Source Data file, or from the corresponding author upon request. Source data are provided with this paper.

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

## Acknowledgements

This work was supported by the National Key R&D Program of China (2021YFF1200800), the National Natural Science Foundation of China (22234004), Shenzhen Technology Program (KQTD20190929172743294 and JCYJ20200109141231365), Shenzhen Key Laboratory of Smart Healthcare Engineering (ZDSYS20200811144003009), Guangdong Provincial Key Laboratory of Advanced Biomaterials (2022B1212010003), Guangdong Innovative and Entrepreneurial Research Team Program

(2019ZT08Y191), Guangdong Major Talent Introduction Project (2019CX01Y196) and Tencent Foundation through the XPLORER PRIZE for financial support. The authors also acknowledge the assistance of Southern University of Science and Technology (SUSTech) Core Research Facilities, Genergy Biotechnology (Shanghai) Co. Ltd. for the assistance with scRNA-seq and the National University of Singapore. Yan Wu is a scholar under the Collaborative Ph.D. Program with SUSTech.

## Author contributions

The experimental design, hHO culture, mMPC fabrication, and data analysis were all undertaken by Yan Wu. The design of the mMPC fabrication was partially contributed by Jinhao Cheng, whereas Jie Qi was responsible for the microstructure characterization of the mMPC. Signal acquisition was designed by Chen Hang and Ruihua Dong. The work was supervised by Boon Chuan Low, Hanry Yu, and Xingyu Jiang. All the authors discussed and commented on the manuscript.

## Competing interests

The authors declare no competing interests.
