## [Peer Review File · Nature Communications]

Three-dimensional liquid metal-based neuro-interfaces for human hippocampal organoidsREVIEWER COMMENTS

Reviewer #1 (Remarks to the Author):

This paper showed the development of 3D-MEA device that enable to analyze neural activity of organoids in 3D. The idea to detect hippocampal organoid's activities by 3D-MEA is interesting, however, there are some concerns as follows;

Major Points

1. The scientific significance by 3D-MEA detection is not enough in current data. By 2D-MEA, researchers can apply thousands of channels and can analyze detailed neural network activity, but this paper's channel number is 128 and there are no data that showed network activities such as synchronization and propagation of spike etc. Although the stretchability and flexibility with stable electrical stability is novel and useful, to surpass previous achievements by other researches, the significant data that only 3D-MEA-with-organoid can approach is needed. The content of Fig6 in current version is general assessment of neural activities, and this paper need more data and analysis to prove what the new device only can contribute to the neural organoid research field.

2. The assessment of hippocampal organoid is not enough in current version. The immunostaining of Figs3/5 and Supple Figs4-5 are not clear and it is difficult to evaluate nuclear staining pattern of FOXG1/LEF1/PAX6/HOPX/ZBTB20/PROX1, and NESTIN and MAP2 staining pattern is also not clear. SULF2 and SEMA5A staining at Supple fig5 looks vague. Besides, the generation of DG neurons, CA3 neurons, and dorsomedial telencephalic region that produce DG/CA3 neurons have already achieved by refs 5, 31, and 26 respectively. The authors claimed their organoid increase number of hippocampal progenitor and DG neuron by addition of Wnt3a and SHH, but there are no comparison with previous protocols and no quantification data. Thus, to make scientific significance regarding generation of hippocampal organoids, there need more novelty with clear data sets.

Also, the authors inadequately described previous achievements. The authors put refs 26/31 in result section but these papers with ref5 brought scientific advantage in stem cell research field regarding generation of hippocampal neuron/tissue. The authors can describe these achievements in introduction and can make it clear what have been done and not done. This will make the authors achievement more clear.

3. Page2 lines 56-61: The authors claimed 3D-MEA surpass 2D-MEA regarding higher spike rates, but 3D-MEA detect only surface of organoids whereas 2D-MEA detect neurites that elongate from organoid. Fig3d suggests the surface cells of organoid are PAX6 positive progenitor cells, but this point is not described in article. The interpretation of neural activity detected by 3D-MEA in more detail is needed.

4. In Fig4, authors treat ZBTB20 as hippocampal marker, but ZBTB20 also express in ventral telencephalon and astrocyte. It is good to show these region are not included in hippocampal cell cluster.

Minor points:

1. Page2 lines 56-61: The authors describe only MEA but calcium imaging is another strong approach for neural activities in organoid research field. The authors can compare MEA and Ca imaging in brief.
2. Supplementary Fig7e: There is no information about what red and blue color indicate.

Reviewer #2 (Remarks to the Author):

In this manuscript, Wu et al. use liquid metal and stretchable polymers to create a conformal 3D metal-polymer conductor (MPC) meshwork as a neural interface to study human hippocampal organoids. The MPC incorporates 128 microelectrode channels with stretchability of up to 500%. The authors further demonstrate successful differentiation of human induced pluripotent stem cells into human hippocampal organoids (hHO) through Wnt3a and SHH activator induction, which is supported by the immunostaining data for expression of hippocampal markers and single cell RNA-seq transcriptomics. They finally use the MPC platform to detect neural signal from the hHO through multiple electrode channels. The paper demonstrates impressive engineering in creating the advanced flexible conductive meshwork along with interesting biological strategy for directing the lineage commitment and differentiation of iPSCs into hHOs. While the manuscript is overall novel and strong, there are notable concerns that need to be addressed before publication.

- 1- A central claim of the manuscript is the flexibility and conformality of the MPC for noninvasive recording from the hHO. While this is correct, a notable concern with the sandwich approach shown in Fig. 1c is the significant deformation of the organoid when confined between the two meshes. The deformation looks dramatic when compared to the more round and spherical profile seen in Fig. 2a where the hHO is placed on a 64-channel single MPC mesh. The sandwich strategy is clearly effective for creating close contact between the organoid and the electrodes as well as increasing the channel numbers but given the obvious squeeze seen in Fig1c., the biological recordings from such conditions could be very questionable. The authors need to address this issue by providing data on the extent of deformation caused by this approach as well as potential viability/functionality effects that might have been caused through the deformation.
- 2- Data in Fig. 2 demonstrates the stretchability of the MPC meshwork. Nonetheless, given that the organoid system used in this study is grown externally and then inserted into the device, parameters such as bending stiffness and flexural rigidity have higher relevance and are required for quantitative characterization of the non-invasiveness aspect of the device. Furthermore, I am curious whether authors think the stretchability of the current device could be harnessed for minimally invasive growth in place strategies? If so, it may be beneficial to discuss the potential strategies in the discussion section.
- 3- Another strength of the manuscript is the advanced number of electrodes (128) integrated into the MPC mesh structure that deserves further emphasis and representation in the paper. The only place where this is currently shown is the text and Fig. 6f that demonstrates recordings from only a handful of electrodes before glutamine treatment and a moderate number afterwards. Nonetheless, authors report around 40% recording from the channels (54 out of 128) in Fig. s8, which looks inconsistent with the data represented in Fig. 6f and weakens the claim for functionality of the electrode array. Is the 40%

yield a typical number or the highest number ever recorded? Corresponding statistics should be provided. Also, how do authors interpret / explain the lack of recording from the other 74 channels?

4- The data in Fig. 6e suggests higher spike rates from the MPC mesh compared to the 2D MEA. The authors interpret this as an advantage of examining 3D neural tissues compared to the monolayer neuron cultures, which tends to be likely. Nonetheless, how can we be sure that this is not due to the mechanical assault of squeezing the organoid between the meshes or its downstream effect? Perhaps this should be reexamined and demonstrated with a 64-channel mesh where the original organoid shape is retained?

5- It is argued that the mesh-type 3D MEA is applied to the hHO for the first-time. Was the first demonstration possible because of technological advancement of the mesh-type? Or do other existing 3D MEAs have capability to be applied to hHO or not? Or were the existing devices not just applied to hHO? Clarification on this would be helpful for understanding the state-of-the-art technologies.

6- The authors mention the liquid metal electrodes may degrade during the 1-month recording period. Will the degraded metal ions be toxic to the organoids and further affect their activity?

7- How could the TPU solution fill the mesh-shaped microchannel in PDMS slab #2? By capillary force or pressurization?

Reviewer #3 (Remarks to the Author):

Wu and co-authors present a device that can detect electrophysiological activity from human hippocampal organoids. The work presents two fundamental aspects: the development of a device that goes beyond the state of the art for recording the electrophysiological activity of a floating system such as the organoids and an improvement in the protocol for the human hippocampal organoid. However, I find that the work is poorly organized and does not highlight what could have been the real strong point: the device.

In particular:

In paragraph lines 56-72, the authors highlight the state of the art of the devices available and developed over the years to record the electrophysiological activity from cellular preparations in vitro, highlighting their limitations. However, they do not mention the work of Sosia 2020, which presents a flexible device with 256 channels able to record the basal activity and stimulate at the same time. Moreover, it allows recording the third dimension. On the other hand, the proposed device "envelops" the whole organoid but does not acquire information related to signal transmission within the structure. How can the presented device be considered better?

The authors point out that their device is able to record from a larger area than other MEA devices, but the reported percentage of active electrodes is about 50%. How can the authors ensure that their device is still a viable alternative?

The device is built using the properties of PDMS; however, the literature shows how this material can influence the effects of a chemical compound due to its absorption and adsorption processes (Toapke et al., 2006).

The focus (also considering the title reported and chosen by the authors) should be the device. However,

all manufacturing and testing processes have been reported in the Supplementary.

Looking at the images shown of the organoids inside the device, the first appear quite deformed. How can the authors prove that this substantial modification of the geometry of the cell preparation does not alter its functioning?

In general, for all the analyzes reported, it is not clear to me what the number of organoids tested is, especially those coupled with the developed device. What is the success rate?

In general, it would be worth investigating all the aspects related to improving the proposed device compared to those on the market, especially referring to high-density commercial systems that allow you to record from thousands of electrodes simultaneously (MaxWell and 3Brain, for example).

Minor revision:

Pay attention to the abbreviations, always writing them the same, for example, wnt3a or WNT3a?

Reviewer #4 (Remarks to the Author):

Wu et al. have developed a mesh multi electrodes array that can form a 3D neuro-interface with human hippocampal organoids (hHOs). The electrodes are a liquid metal-polymer conductor (MPC)-based system that are flexible and stretchable which enables the mesh MPC (mMPC) to couple with hHOs in a conformal manner. They also found a culture condition for generating hHOs by incorporating Wnt3a and purmorphamine (the SHH signalling activator) that suppresses the expansion of choroid plexus (ChP) during dorsomedial telencephalon (DMT) induction where the hippocampal primordial develops from. They have identified the cell types that compose of the organoids using single-cell RNA sequencing (scRNAseq) and immunofluorescence imaging. Overall, the mMPC system shows a great potential to measure the electrical activity of brain organoids in 3D, but there are some important points that would be important to be addressed for this manuscript to be published.

1. scRNAseq

Wu et al. have performed scRNAseq to identify the cell types of the organoids, but I have a major concern about the cells that were used for sequencing. According to the methods section for scRNAseq, the hHOs were dissected into smaller tissues and replaced on PLL-laminin-coated flask to minimise the information loss and capturing more cells and even grow further in this condition for 3-4 days. To my knowledge, to identify the cell type of the organoids, the cells should be dissociated and collected directly from the organoid as a 3D multicellular tissue for sequencing. As soon as the cells from the organoid attach to the flask and be cultured in 2D, the different microenvironment between the cells, such as the different cell-cell interaction, mechanical cues, nutrient or oxygen delivery that the cell sense between 2D and 3D, the cells might differentiate into different cell types than the original 3D hHOs that the current scRNAseq data might not faithfully represent the actual cell types of the hHOs when it was originally 3D. For this reason, the researchers in the organoid field pool as many organoids as possible to have enough cell numbers for scRNAseq. Thus, scRNAseq of the cells that are directly dissociated and

collected from the hHOs would be essential to accurately identify the cell types. Also, to present the correlation between hHOs and in vivo developing hippocampus, it would be much more informative to analyse the integrated data of the hHOs scRNAseq data and in vivo developing hippocampus scRNAseq data as a reference. I believe, in this way, it would show much more informative than the heat map on Fig. 4e. Besides, it might help to identify the cluster of “Others” which contains a substantial number of the cells.

2. Advantages of mMPC being a 3D MEA system

Throughout the manuscript, the three-dimensionality of mMPC system was emphasized. It is exciting to have developed a system where you can “sandwich” and easily fix the position of the brain organoid and can get the electrical signals of the broad range of the organoid. However, there were no data that has shown its advantages in the aspect of 3D. For example, have you observed/could you capture how electrical signal propagate from one location to the other in 3D? If yes, it would be very powerful to show to strengthen the importance of 3D. Also, when comparing mMPC to 2D MEA, one aspect that was mentioned was the cell migration on 2D MEA. According to the methods section, it is stated that the MEA plate was coated with PLL-laminin which would have led the cell migration. To compare the 2D and 3D system in the aspect of cell migration equally, I think the coating condition should be the same, either both non-coated or both coated to compare more properly.

3. In Fig. 6g, 1 month of usage means actual 1 month of culturing the organoid on mMPC?

4. In Fig. 3c, it would be more informative if you could specify the time point of the development of the organoid as a continuation of Fig. 3a.

Reviewer 1

This paper showed the development of 3D-MEA device that enable to analyze neural activity of organoids in 3D. The idea to detect hippocampal organoid's activities by 3D-MEA is interesting, however, there are some concerns as follows;

Major Points

- 1. The scientific significance by 3D-MEA detection is not enough in current data. By 2D-MEA, researchers can apply thousands of channels and can analyze detailed neural network activity, but this paper's channel number is 128 and there are no data that showed network activities such as synchronization and propagation of spike etc. Although the stretchability and flexibility with stable electrical stability is novel and useful, to surpass previous achievements by other researches, the significant data that only 3D-MEA-with-organoid can approach is needed. The content of Fig6 in current version is general assessment of neural activities, and this paper need more data and analysis to prove what the new device only can contribute to the neural organoid research field.**

Author response: Thank you for pointing this out. As suggested by the reviewer, we detected more neural signals from hHOs cultured for more than one hundred days by our mMPC. During the signal acquisition, we observed the neural signals from two channels that always appeared in pairs. Moreover, these paired signals always occurred one after the other, and the time interval was stable on millisecond time scales. These data suggested the existence of signal transmission between the neurons of those two channels. Our hHOs, more than one hundred days old, have grown functional synapses between neurons, and our mMPC was able to detect this synchrony. This data we have added is in Fig. 6f.

In addition, we found oscillatory network activity, i.e., coordinated bursts of spikes across electrodes, in the hippocampal cyb-organoid. These indicated that the mesh MPC could characterize the functional network on the surface of the hHO with excitatory and inhibitory neurons. We add this data in Fig. 6g.

Fig. 6 | Electrical activity recording. **a**, (Upper) Raster plots of neural spikes, and (lower) the continuous signal in a channel of the mMPC. **b**, Continuous recording for a spike. **c**, The uniform spike waveforms and the average waveform in a channel of the mMPC. **d**, Two different spikes appeared in the same channel (upper) and the sorting results (lower). **e**, The comparison of spike rates detected by the mMPC and the 2D MEA (these channels covered by the hHO). **f**, The synchronous signals between two channels. (Upper) Raster plots of neural spikes in these two channels, and (lower) continuous recording for two spikes. The time interval was less than 1 ms. **g**, The oscillatory network activity of the hHO. (Upper) the population activity histogram and (lower) the raster plot (black) identified bursts of spiking detected on 32 channels and coordinated bursts (red) of activity across channels. **h**, Raster plots before and after

Glutamine supplement. *i*, The impedance of the mMPC before and after integration with the hHO for 1 month.

However, we must admit that the current detection of hHO signaling by mesh MPC still needs to be improved. The main limitation is that the instrumentation we used (Plexon system) has a transmission capacity of only 32 channels. Even though our mesh MPC has 128 channels, acquiring signals from all channels simultaneously is impossible, thus we have to carry out the measurements in 4 separate experiments. That also limited our access to understanding the complete surface neural network of the hHO. However, the fact that 32 channels could detect neural activities has demonstrated the potential of the mMPC. In the future, we will strive to upgrade our detection equipment to utilize mMPC's ability in organoid signal detection fully. This limitation has been added in the discussion section.

The revised text reads as follows on (Results section, Page 15 lines 428-441):

“[In a series of experimental tests, we detected signals from the single hHO or hHOs-fused assemblies, and the number of active channels was listed in Supplementary Table 1. The number of active channels was mainly affected by the hHO size. For signal acquisition of an individual hHO, we detected signals in a maximum of 54 channels, whereas when we used fused assemblies of 2 hHOs, the maximum number of active channels increased to 85. Simultaneous signals always appeared in pairs in two channels. We found that the time interval of these paired signals was less than 1 ms. They always appeared one after the other (Fig. 6f). This observation suggested that neurons on the hHO formed functional synapses capable of transmitting signals and were detected by the mMPC. In addition, neural activity was detected on the hHO (Fig. 6g). As potential evidence of neural network maturation, the synchrony was proposed to reflect a balance between ExN and InN, and coordinate neuronal communication in the hHO⁴². These results highlight the potential of the hippocampal cyb-organoid applied to understand the development and disorders in the hippocampus.]”

(Discussion section, Pages 17-18 lines 506-515):

“[Compared with mature commercial systems, such as MaxWell Biosystem, HyperCAM Alpha, and Axion system, which offer an integrated platform with high-density electrodes in multiwell, real-time high-resolution scanning, and controlled environment, there is still much room for developing neural activity detection of hHOs by the mMPC. The ability of our Plexon instrument to detect only 32 channels of data at a time limits the mMPC's ability to record the complete neural activity of the hHO. In addition, obtaining long-term neuronal activity from 3D organoids requires a stable and healthy culture environment. However, there are many challenges in upgrading a system, including instrumentation, stimulation modules, temperature regulation, and gas regulation, to name a few. Adding other modules also introduces noise to the signal acquisition. These will be explored in the future.]”

2. **(2.1) The assessment of hippocampal organoid is not enough in current version. The immunostaining of Figs3/5 and Supple Figs4-5 are not clear and it is difficult to evaluate nuclear staining pattern of FOXG1 / LEF1 / PAX6 / HOPX / ZBTB20 / PROX1, and NESTIN and MAP2 staining pattern is also not clear. SULF2 and SEMA5A staining at Supple fig5 looks vagu.**

Author response: Thank you for this suggestion. We had previously used more cryosections, mainly to save on sample consumption. This time, we used whole organoids for immunostaining. To improve the quality of images to allow for better interpretation of the results, we added separate magnified images of each marker to demonstrate its nuclear location. The staining of the neurofilaments was also changed to a magnified image to show their structures clearly, and the staining of SULF2 and SEMA5A was redone to show their nuclear and cytoplasmic locations this time. When characterizing the hHO in the mMPC, the mMPC is particularly susceptible to staining with DAPI and secondary antibodies, making imaging more difficult. To highlight the structure of the neurons on the surface of the hHO, and the relative positions between the cells and the mMPC, we used hHOs that expressed green fluorescent protein, and some of these cells that emitted bright green fluorescence demonstrated that the neurons had not been damaged after the hHOs had been sandwiched in between mMPCs. Those changes can be found in Fig. 3/5 and Supplementary Fig. 4.

Fig. 3 | Generation of hHOs. a, The overall strategy to generate hHOs. **b**, The typical morphology of hHOs at different ages. Scale bar: 100 μ m. **c**, The significant markers in the developmental process of the hHO. **d**, Staining images of day-30 hHOs. Upper: Lager-scale image of the whole hHO. Scale bar:

100 μm . Lower: Zoom-in greyscale view. Scale bar: 40 μm . **e**, The hippocampal PAX6⁺ and HOPX⁺ progenitors in day-60 hHOs. Scale bar: (left) 100 μm and (right, zoom-in greyscale view) 40 μm . **f**, The hippocampal PROX1⁺ cells in day-60 hHO and day-85 hHO. Scale bar: 150 μm . **g**, qPCR for genes expressed in day-117 hHOs, day-71 hHOs versus day-46 hHOs (* P <0.05; ** P <0.01, n =3, unpaired t -test). **h**, qPCR for genes expressed in day-45 hHOs versus day-45 DMT organoids (* P <0.05, n =3, unpaired t -test).

Fig. 5 | The hippocampal cyb-organoid platform. **a**, Integration of the 128-channel mMPC and the hHO. (Upper) The hHO was stayed in a low-attachment dish. (Lower) The hHO was enveloped in two-layer mMPC. Scale bar: 1 mm. Notably, there was no significant deformation when the hHO was inserted between two layers of mMPC. **b**, The bottom view captured by a microscope. Scale bar: (Left) 500 μm , (right) 100 μm . **c**, Confocal images of mouse hippocampal cells culturing in the mMPC for 30 days. Scale bar: 100 μm . **d**, Confocal images of MAP2⁺ neurons, GFAP⁺ astrocytes and EGFP⁺ cells in the cyb-organoid platform. Scale bar: 100 μm . **e**, The cross-sectional view of the conformal attachment between the mMPC and the hHO. The arrowhead points to the mMPC. Scale bar: 100 μm .

Supplementary Fig. 4 | HHOs. **a**, ZBTB20⁺PROX1⁺ granule neurons in the day-85 hHO. Scale bar: 100 μ m. **b**, Neuronal marker TAU and MAP2 were expressed in hHOs. Scale bar: 100 μ m. **c**, The hippocampal markers SULF2 and SEMA5A, were expressed in hHOs. Scale bar: 40 μ m. **d**, The expression of hippocampal markers ZBTB20 and PROX1, and neuron markers NEUN and MAP2 in the cells dissociated from hHOs. Scale bar: 100 μ m. **e**, The expression of astrocyte marker GFAP and oligodendrocyte progenitor cell marker OLIG2 in hHOs. Scale bar: 100 μ m.

(2.2) Besides, the generation of DG neurons, CA3 neurons, and dorsomedial telencephalic region that produce DG/CA3 neurons have already achieved by refs 5, 31, and 26 respectively. The authors claimed their organoid increase number of hippocampal progenitor and DG neuron by addition of Wnt3a and SHH, but there are no comparison with previous protocols and no quantification data. Thus, to make scientific significance regarding generation of hippocampal organoids, there need more novelty with clear data sets.

Author response: Thank you for pointing this out. Our generation protocol was based on the dorsomedial telencephalon (DMT) organoid (ref. 26). Ref. 26 demonstrated that hippocampal cells could be differentiated in the DMT organoid, but these cells were obtained after the researcher cut off the choroid plexus (ChP) from the DMT organoid, dissociated and cultured the remaining tissue alone. We were puzzled as to why the hHO could not be obtained by directly culturing the DMT organoid for a long period, as in this condition, the ChP was able to release growth factors continuously to regulate hHO development as occurs in vivo development. Thus, we conjectured that the presence of ChP may have hindered hHO development, which may be why it was cut off in ref. 26. The activation of the Wnt pathway in ref. 5/6/7 to differentiate hHO suggests the importance of Wnt in hHO generation, and the roles of Wnt3a and SHH for hippocampal development have been proved in vivo. These studies remind us that Wnt3a and SHH may have the ability to inhibit ChP and improve hHO growth in the DMT organoid. Our results indicate that our protocol has yielded more hippocampal progenitors and DG neurons than the DMT organoids in ref. 26. Because ref. 5/31 (have been changed to ref. 5/6 in the revised manuscript) have used completely different protocols, we have not compared ours with ref. 5/31. We would more like to explore the relationship between ChP and hHO growth in DMT organoids and the roles played by Wnt3a and SHH, not claiming that our protocol is better than all previous studies.

The scientific significance of our protocol lies in the fact that we have made comparisons on the mRNA level with the newly available publication on the pattern of mRNA in developing human hippocampus. Combining the immunofluorescence and scRNA-seq datasets, our results indicate the successful generation of hHOs.

Thanks to the reviewers for pointing out that we did not quantify this metric. We used qPCR to compare the key hippocampal marker PROX1 and the ChP marker TTR, and the results suggested that the expression of TTR was higher in DMT than in our hHO, but PROX1 was higher in hHO than in DMT. This data is added in Fig. 3h. We also compared the expression of growth factor Wnt3a, hippocampal marker ZBTB20\PROX1 in hHOs at different stages. The endogenous Wnt3a, which promotes hippocampal differentiation, increases during the growth period but decreases during maturation, whereas the hippocampal marker ZBTB20\PROX1 has been increasing. This data is added in Fig. 3g and corresponds to the staining results in Fig. 3f. Together, these results demonstrate the importance of Wnt3a and SHH for hippocampal cell differentiation.

Fig. 3 | Generation of hHOs. a, The overall strategy to generate hHOs. **b**, The typical morphology of hHOs at different ages. Scale bar: 100 μ m. **c**, The significant markers in the developmental process of the hHO. **d**, Staining images of day-30 hHOs. Upper: Lager-scale image of the whole hHO. Scale bar:

100 μm . Lower: Zoom-in greyscale view. Scale bar: 40 μm . **e**, The hippocampal PAX6⁺ and HOPX⁺ progenitors in day-60 hHOs. Scale bar: (left) 100 μm and (right, zoom-in greyscale view) 40 μm . **f**, The hippocampal PROX1⁺ cells in day-60 hHO and day-85 hHO. Scale bar: 150 μm . **g**, qPCR for genes expressed in day-117 hHOs, day-71 hHOs versus day-46 hHOs (* $P < 0.05$; ** $P < 0.01$, $n = 3$, unpaired t-test). **h**, qPCR for genes expressed in day-45 hHOs versus day-45 DMT organoids (* $P < 0.05$, $n = 3$, unpaired t-test).

The revised text reads as follows on (Results section, Page 8 lines 272-279):

“[Compared with day-45 DMT organoids, we found a significant increase in PROX1 mRNA expression and a significant decrease in TTR mRNA expression in the day-45 hHO (Fig. 3h). Along with the hHO growing, the ZBTB20 and PROX1 mRNA expression gradually increased (Fig. 3g). Immunostaining also showed increased PROX1 expression in the more mature hHO (Fig. 3f). Notably, Wnt3a mRNA expression increased and then decreased (Fig. 3g), demonstrating that the endogenous growth factor Wnt3a, which facilitates hippocampal differentiation, was synthesized in greater abundance during the growth period, whereas during the maturation phase, Wnt3a was no longer so required.]”

(Methods section, Page 24 lines 795-799):

*“[**Quantitative real-time PCR** qPCR (5-10 organoids per sample) was performed, using Taq Pro Universal SYBR qPCR Master Mix (Vazyme Biotech Co., Ltd.). The cycle threshold (CT) values of samples were collected for statistical analysis. The data were normalized to the GAPDH expression. Primers were listed in Supplementary information.]”*

To make the scientific significance more explicit, we have clarified the statement in (Introduction section, Page 2 lines 108-111):

“[Furthermore, we demonstrated the mutual antagonism of ChP and hippocampal differentiation in the DMT organoid. Under the Wnt3a and SHH activator induction, we generated hHOs with many PAX6⁺, HOPX⁺ hippocampal progenitors, and PROX1⁺ granule neurons.]”

(Results section, Page 11 lines 344-345):

“[Combining the immunofluorescence and scRNA-seq datasets, our results indicated the successful generation of hHOs.]”

(2.3) Also, the authors inadequately described previous achievements. The authors put refs 26/31 in result section but these papers with ref5 brought scientific advantage in stem cell research field regarding generation of hippocampal neuron/tissue. The authors can describe these achievements in introduction and can make it clear what have been done and not done. This will make the authors achievement more clear.

Author response: We think this is an excellent suggestion. In the results section, we

deleted the texts about past developments in inducing hippocampal organoids and added them to the introduction. We focused on the description of hippocampal cells differentiating out of DMT tissues (ref. 26) and past protocols reflecting the importance of the Wnt pathway for differentiated hippocampal cells (ref. 5/6/7). However, these protocols have yet to demonstrate the relationship between the Wnt&SHH pathway, ChP in DMT tissue, and hippocampal differentiation. The present exploration of hippocampal organogenesis will describe this issue.

The revised text reads as follows on (Introduction section, Pages 2-3 lines 79-111):

*“[Existing 3D MEAs specifically formulated for brain organoids were only applied to cerebral^{20–22} or cortical organoids²³ and have yet to be employed on hHOs, as there exist more established protocols for generating the cerebral organoid and the cortical organoid. Generating hHOs is challenging because more accurate manipulation of signaling factors is required to induce the differentiation process. Human hippocampal primordium develops from dorsomedial telencephalon (DMT), which is regulated by BMPs, Wnts, and SHH^{24,25} (Supplementary Fig. 3a). Researchers have identified the feasible timing to add BMP4 and CHIR (a chemical activator of Wnts) in the medium to generate DMT organoids. These DMT organoids gave rise to cortical hem (CH), choroid plexus (ChP), and hippocampal primordium²⁶. The CH and ChP release Wnts and BMPs to regulate hippocampus growth²⁴. Previous *in vivo*^{24,25} and *in vitro* models^{7,27} have proved the importance of Wnt3a and SHH signals in hippocampal development. In particular, several works to grow hHOs *in vitro* have opted for long-term use of the Wnt pathway (Wnt3a, 14 days⁷/Wnt3a, 20 days⁵/CHIR, 90 days⁶) to induce hippocampal fate. However, the relationship between the Wnt&SHH pathways, ChP in DMT organoids, and hippocampal differentiation has yet to be demonstrated.*”

The hippocampus seems to hold significant importance concerning neurological disorders. Sourcing electrical information from hHOs holds key potential in laying a cogent foundation for future diagnosis and treatment of neurological diseases. In this article, we report a method to fabricate the flexible and stretchable mMPC composed of gallium indium (GaIn) alloy and elastic polymers (thermoplastic polyurethane, TPU, and polyurethane, PU), aiming to detect signals of hHOs (Fig. 1a). Unlike traditional MEAs with the insulating layer made from photocurable polymer SU-8, our mMPC provided (1) free deformation, including fold, bend, and twist at any angle; (2) stretchability up to 500%, without losing electrical performance under strain. A conductive polymer, poly (3,4-ethylenedioxythiophene) (PEDOT), was deposited on the electrode surface to improve stability. These features allowed the mMPC to couple with hHOs in a conformal manner, with a wrapping coverage of over 75% of the organoid surface, which is larger than all current 3D MEAs (<50%). Furthermore, we demonstrated the mutual antagonism of ChP and hippocampal differentiation in the DMT organoid. Under the Wnt3a and SHH activator induction, we generated hHOs with many PAX6⁺, HOPX⁺ hippocampal progenitors, and PROX1⁺ granule neurons.]”

3. **(3.1) Page2 lines 56-61: The authors claimed 3D-MEA surpass 2D-MEA regarding higher spike rates, but 3D-MEA detect only surface of organoids whereas 2D-MEA detect neurites that elongate from organoid.**

Author response: Thank you for pointing this out. The reviewer is correct, and we have excluded from the dataset of spike rates detected by those channels that measured from neurons that have elongated from organoids. The excluded spike rates are all below 0.15 spike/sec.

The revised text reads as follows on (Results section, Pages 14-15 lines 417-426):

“[A similar signal was observed using the 2D MEA (Supplementary Fig. 11a). After placing hHOs on the 2D MEA for ~3 days, cells migrated away from the hHOs and formed a monolayer on the MEA surface (Supplementary Fig. 10a), and some neural signals were detected from this area. The higher spike rates were concentrated on these electrodes covered by the hHO (Supplementary Fig. 11b-c). These results suggested that neurons in tissues might maintain higher nerve activity than monolayer neurons. Even after removing the data of the extended nerves which were not integrated into the hHO, we compared the spike rates detected by the electrodes covered by hHO in the 2D MEA with the mMPC, showing more active neural activities in the mMPC (Fig. 6e). The comparison highlighted the necessity to develop 3D MEAs.]”

- (3.2) Fig3d suggests the surface cells of organoid are PAX6 positive progenitor cells, but this point is not described in article.**

Author response: Thank you for pointing this out. We have added the suggested content to the manuscript on (Results section, Page 8 lines 258-260):

“[In the resulting hHOs, two important hippocampal progenitors, HOPX⁺ and PAX6⁺ progenitors³⁷, and ZBTB20⁺PROX1⁺ granule neurons were observed (Fig. 3f and Supplementary Fig. 4a). PAX6⁺ progenitors were even more early in the hHO (Fig. 3d).]”

- (3.3) The interpretation of neural activity detected by 3D-MEA in more detail is needed.**

Author response: Thank you for this suggestion. We have added the detection of synchrony, oscillatory activities in results.

The revised text reads as follows on (Results section, Page 15 lines 428-446):

“[In a series of experimental tests, we detected signals from the single hHO or hHOs-fused assemblies, and the number of active channels was listed in Supplementary Table 1. The number of active channels was mainly affected by the hHO size. For signal acquisition of an individual hHO, we detected signals in a maximum of 54 channels, whereas when we used fused assemblies of 2 hHOs, the maximum number of active channels increased to 85. Simultaneous signals always appeared in pairs in two channels. We found that the time

interval of these paired signals was less than 1 ms. They always appeared one after the other (Fig. 6f). This observation suggested that neurons on the hHO formed functional synapses capable of transmitting signals and were detected by the mMPC. In addition, neural activity was detected on the hHO (Fig. 6g). As potential evidence of neural network maturation, the synchrony was proposed to reflect a balance between ExN and InN, and coordinate neuronal communication in the hHO⁴². These results highlight the potential of the hippocampal cyb-organoid applied to understand the development and disorders in the hippocampus.

Concerned that pressure from the top mMPC could affect the electrical activity of the hHO, we compared the neural signals detected with and without the top mMPC using the same cyb-organoid in both cases. A comparison of the average spike rates of multiple active channels over two minutes showed no significant difference (Supplementary Fig. 12a).]

Fig. 6 | Electrical activity recording. **a**, (Upper) Raster plots of neural spikes, and (lower) the continuous signal in a channel of the mMPC. **b**, Continuous recording for a spike. **c**, The uniform spike waveforms and the average waveform in a channel of the mMPC. **d**, Two different spikes appeared in the same channel (upper) and the sorting results (lower). **e**, The comparison of spike rates detected by the mMPC and the 2D MEA (these channels covered by the hHO). **f**, The synchronous signals between two channels. (Upper) Raster plots of neural spikes in these two channels, and (lower) continuous recording for two spikes. The time interval was less than 1 ms. **g**, The oscillatory network activity of the hHO. (Upper) the population activity histogram and (lower) the raster plot (black) identified bursts of spiking detected on 32 channels and coordinated bursts (red) of activity across channels. **h**, Raster plots before and after

Glutamine supplement. *i*, The impedance of the mMPC before and after integration with the hHO for 1 month.

We also added the limitation of 3D-MEA in detecting neural activities in discussion. The revised text reads as follows on (Discussion section, Pages 17-18 lines 506-522):

“[Compared with mature commercial systems, such as MaxWell Biosystem, HyperCAM Alpha, and Axion system, which offer an integrated platform with high-density electrodes in multiwell, real-time high-resolution scanning, and controlled environment, there is still much room for developing neural activity detection of hHOs by the mMPC. The ability of our Plexon instrument to detect only 32 channels of data at a time limits the mMPC's ability to record the complete neural activity of the hHO. In addition, obtaining long-term neuronal activity from 3D organoids requires a stable and healthy culture environment. However, there are many challenges in upgrading a system, including instrumentation, stimulation modules, temperature regulation, and gas regulation, to name a few. Adding other modules also introduces noise to the signal acquisition. These will be explored in the future. More importantly, we relied more on the mMPC's softness to allow for non-invasive signal detection, although the mMPC has great stretchability. We are concerned that a stretched device would damage the tissue for a long time. Thus, our next plan is to find a suitable material with plastic deformation properties to replace the TPU and PU we are currently using, and we hope that the next device will be able to co-grow with the brain tissue without damaging the brain tissue. In this way, we will be able to monitor the growth and maturation of brain tissue in situ for a long time.]”

- 4. In Fig4, authors treat ZBTB20 as hippocampal marker, but ZBTB20 also express in ventral telencephalon and astrocyte. It is good to show these region are not included in hippocampal cell cluster.**

Author response: This is an excellent suggestion. To confirm this, we separately examined the distribution of key dorsal and ventral telencephalon markers in all clusters. The key markers of ventral telencephalon, such as NKX2-1\GSX1\GX2\LHX6, were largely not expressed. This also occurred in the single-cell RNA sequencing of 81-day hHOs. That result suggested no cell clusters of the ventral telencephalon in the hHO above 70 days. We have added this comparative result in Supplementary Fig. 5a.

Supplementary Fig. 5 | Ventral telencephalon and ChP in hHOs. a, UMAP visualization of the dorsal/ventral telencephalon markers. b, UMAP visualization of the ChP cluster and neural cluster. c, The expression of TTR in two cryosections of the day-85 hHO. d, Cavities appeared in day-132 hHOs.

The revised text reads as follows on (Results section, Page 10 lines 303-305):

“[The ventral telencephalon markers *NKX2-1*, *GSX1*, *GSX2*, and *LHX6* were not expressed in the cell clusters, suggesting that hHO was fully oriented toward dorsal telencephalic development during this period (Supplementary Fig. 5a).]”

Minor points:

- Page 2 lines 56-61: The authors describe only MEA but calcium imaging is another strong approach for neural activities in organoid research field. The authors can compare MEA and Ca imaging in brief.**

Author response: Thank you for this suggestion. We have added the suggested content to the manuscript on (Introduction section, Page 2 lines 60-69)

“[The electrical information in neural systems in vitro is widely detected through a planar MEA^{12,13} and calcium imaging¹⁴⁻¹⁶. Electrodes in commercial MEAs distribute over a small surface area at the bottom of a single well. Thus, these devices cannot match hHOs in the suspension culture system. Implantable electrodes designed for animals are also unsuitable for hHOs due to a mismatch in shape and size^{13,17,18}. Single or arrays of needle

electrodes may potentially be inserted into brain organoids to detect signals, like vertically-arrayed 3D MEAs¹⁹. Calcium imaging can allow live imaging, which provides information about electrical activity on a larger scale but is dependent on imaging capabilities. The size and 3D nature of brain organoids pose a challenge for acquiring calcium imaging data.]”

2. Supplementary Fig7e: There is no information about what red and blue color indicate.

Author response: Thank you for pointing this out. The blue and red channels are DAPI and MAP2, respectively. We have added this information in Supplementary Fig. 10e.

Supplementary Fig. 10 | Long-term change of hHOs in MEA and noninvasive assessment. a, (Left) The hHO on a commercial 2D MEA after culturing for 20 days. Scale bar: 500 μ m. **(Right)** zoom-in view of the red box in the left figure showing the cell migration on the 2D MEA. Scale bar: 100 μ m. **b, (Left)** bright-field images of hHOs on the normal mMPC after cocultured for 30 days and on the PLL-Laminin-coated mMPC for 20 days. Scale bar: 100 μ m. **c, The mMPC electrodes with and without PEDOT coating**

after culturing the hHO for 2 weeks. The red arrows point to the electrodes. Scale bar: 50 μm . **d**, Non-invasive assessment. The morphology of the same hHO in dish (Left), in the mMPC for 4 days (Middle), after fixation and immunostaining (Right). Scale bar: 100 μm in the staining image and 2 mm in others. **e**, The cross-sectional view showed the attachment boundary between the mMPC and the hHO. Scale bar: 100 μm . (Right) The estimated areas of the hHO and that covered by the mMPC.

Reviewer 2

In this manuscript, Wu et al. use liquid metal and stretchable polymers to create a conformal 3D metal-polymer conductor (MPC) meshwork as a neural interface to study human hippocampal organoids. The MPC incorporates 128 microelectrode channels with stretchability of up to 500%. The authors further demonstrate successful differentiation of human induced pluripotent stem cells into human hippocampal organoids (hHO) through Wnt3a and SHH activator induction, which is supported by the immunostaining data for expression of hippocampal markers and single cell RNA-seq transcriptomics. They finally use the MPC platform to detect neural signal from the hHO through multiple electrode channels. The paper demonstrates impressive engineering in creating the advanced flexible conductive meshwork along with interesting biological strategy for directing the lineage commitment and differentiation of iPSCs into hHOs. While the manuscript is overall novel and strong, there are notable concerns that need to be addressed before publication.

1. (1.1) A central claim of the manuscript is the flexibility and conformality of the MPC for noninvasive recording from the hHO. While this is correct, a notable concern with the sandwich approach shown in Fig. 1c is the significant deformation of the organoid when confined between the two meshes. The deformation looks dramatic when compared to the more round and spherical profile seen in Fig. 2 where the hHO is placed on a 64-channel single MPC mesh.

Author response: Thank you for pointing this out. First, we are using in Fig. 2 not an organoid but a TPU ball. In order to show the conformality, we first used the non-deformable polymer ball to deform the mMPC so that it could better fit the TPU ball.

Of course, we also consider the scenario where the two meshes will squeeze the hHO, especially the top mMPC, and we must admit that the pressure is present and unavoidable during the use. However, to minimize the pressure from the top mesh, we have improved the assembly of the cyb-organoid as much as possible. We think the pressure on the top mesh may have two major sources. If the mesh is stretched, it will exert a force on the hHO. Second, suppose the culture medium is gradually reduced during the changing medium or evaporated after being placed in the incubator. In that case, the top mesh will move downward due to a lower liquid level, which may result in compression.

To address these issues, our improvements include:

- (1) When assembling the two-layer mMPCs, we placed a TPU ball between the two layers of mMPCs to form a reserved space such that they will accommodate the spherical shape of the hHO. When we placed the hHO in the middle of the mMPCs, there was sufficient space to accommodate the hHO. Thus, the top mMPC will not be stretched so that little external force is applied to the hHO. We added this picture in Supplementary Fig. 9a-b.

Supplementary Fig. 9 | Assembling of 128-channel mMPC and biocompatibility of mMPC. **a**, The integrated device coupling the mMPC with culture dishes. (Left): After removing the glass bottom, a confocal 35-mm glass-bottom dish was covered above the circuits of the mMPC, with the mesh part remaining inside the dishes. PDMS simultaneously fixed them in a large 100-mm dish. The end of the flexible printed circuit was left outside the dish to connect to the Plexon instrument. (Right): Assembly of single- and double-layer mMPCs with the 35-mm dish. The TPU ball in the center was to reserve the space the hHO may need. **b**, (Left) The two-layer mMPCs with hHO's culture medium. After remove the TPU ball, there existed some space between these two layers. (Middle and right) The top and the side view after placing one hHO into the two-layer mMPC. Scale bar: 5 mm. When the culture solution was just submerged over the hHO, the top mMPC covered the surface of the hHO without significant squeezing. **c**, Biocompatibility. The mouse hippocampal cells grew in the mMPC for 30 days. Scale bar: 100 μ m.

- (2) In order for the top mMPC to fit the hHO better, after placing the hHO, we removed a certain amount of culture medium until the top mMPC slowly moved down to the surface of the hHO, at which time the culture medium in the dish was generally a little bit above the upper surface of the hHO. The top view of the cyb-organoid tended to give the impression that the hHO was being squeezed, so we replaced the top view via the picture with a certain tilt angle. We also added a picture of the organoid when it was in the petri dish for comparison. This change can be found in Fig. 5a. Our mMPC can wrap around the hHO but maintains its 3D morphology. Also, in Supplementary Fig. 9b, we added a side view. Although the petri dish and culture medium affected capturing the side view, we can still see that the top mMPC is above the hHO, wrapping around the hHO as a curved surface instead of directly squeezing the hHO. This approach greatly reduces the pressure that may be caused by the mesh MPC. We also describe this assembly in detail in the method.

Fig. 5 | The hippocampal cyb-organoid platform. **a**, Integration of the 128-channel mMPC and the hHO. (Upper) The hHO was stayed in a low-attachment dish. (Lower) The hHO was enveloped in two-layer mMPC. Scale bar: 1 mm. Notably, there was no significant deformation when the hHO was inserted between two layers of mMPC. **b**, The bottom view captured by a microscope. Scale bar: (Left) 500 μm , (right) 100 μm . **c**, Confocal images of mouse hippocampal cells culturing in the mMPC for 30 days. Scale bar: 100 μm . **d**, Confocal images of MAP2⁺ neurons, GFAP⁺ astrocytes and EGFP⁺ cells in the cyb-

organoid platform. Scale bar: 100 μm . e, The cross-sectional view of the conformal attachment between the mMPC and the hHO. The arrowhead points to the mMPC. Scale bar: 100 μm .

- (3) To minimize the effect of decreasing culture medium during the changing medium, we changed the medium daily, placed fresh culture medium into the dish first, and then removed one-third of the medium. This approach is described in the method.

The revised text reads as follows on (Results section, Pages 12-13 lines 358-367):

“[To enable the integration of the mMPC with the hHO and facilitate the connection with the Plexon instrument, we assembled the mMPC and culture dishes (Supplementary Fig. 9a). To prevent the two-layer meshes from severely squeezing on the soft hHO, we placed a TPU ball of ~3 mm when assembling the device to reserve the space the hHO may need (Supplementary Fig. 9a-b). After sterilization, the cultured medium was added to the dish, and the TPU ball was removed from the side; the hHO was inserted into the space between the top and bottom mMPCs. The oblique view and the side view showed that the top mMPC varies with the undulation of the hHO surface without causing significant compression on the hHO (Fig. 5a and Supplementary Fig. 9b), indicating the conformal adhesion of the mMPC to the surface of the hHO.]”

(Methods section, Pages 22-23 lines 719-733):

“[The mMPC was fixed with a 35 mm confocal dish which had its glass bottom removed, in a 100 mm dish, leaving the end of flexible printed circuits outside the 100 mm dish (Supplementary Fig. 9a). When assembling the two-layer mMPC, use a TPU ball to pre-simulate the replacement of the hHO and place it between the two layers. All relevant materials, including tapes, dishes, and devices, were sterilized by overnight UV light illumination. Before coculturing with the hHO, the mMPC was soaked in PBS to promote electrode and medium attachment. After removing the TPU ball from the side between the top and bottom meshes, the hHO was inserted into the two-layer mMPC, replacing PBS with 2.5 mL differentiation medium 2 (supplement with BDNF and GDNF) in the 35 mm dish. When the medium was enough to submerge the mMPC completely, the upper mMPC would float up to create a gap between the two layers. The hHO was then aspirated using a wide bore tip and gently placed in the gap. Finally, a small amount of medium was carefully removed until the upper layer adhered to the hHO. The hHO was maintained with the mMPC system in a 37 °C incubator, and the medium was changed daily, only one-third each time.]”

(1.2) The sandwich strategy is clearly effective for creating close contact between the organoid and the electrodes as well as increasing the channel numbers but given the obvious squeeze seen in Fig1c., the biological recordings form such conditions could be very questionable. The authors need to address this issue by providing data on the extent of deformation caused by this approach as well as potential viability/functionality effects that might have been caused through the deformation.

Author response: Thank you for this suggestion. We compared the neural signals detected with and without the top mMPC using the same cyb-organoid in both cases. First, we detected neural signals through the bottom mMPC in the case where the hHO was sandwiched between two layers of mMPC. We then carefully cut the top mMPC and gently lifted it off the hHO surface. After removal of the top mMPC, we went back to detect the signal through the bottom mMPC. A comparison of the average spike rates of multiple active channels over two minutes showed no significant difference. This result demonstrated that the top mMPC does not seriously affect the neural activity of the hHO. This result can be found in Supplementary Fig. 12a. We also added the potential effects in the discussion.

Supplementary Fig. 12 | a, The comparison of spike rates recording in the bottom mMPC before and

after cutting the top mMPC. **b**, The change of amplitude of glutamine-induced spikes.

The revised text reads as follows on (Results section, Page 15 lines 443-446):

“[Concerned that pressure from the top mMPC could affect the electrical activity of the hHO, we compared the neural signals detected with and without the top mMPC using the same cyb-organoid in both cases. A comparison of the average spike rates of multiple active channels over two minutes showed no significant difference (Supplementary Fig. 12a).]”

(Discussion section, Page 17 lines 498-504):

“[Although the sandwich structure of our cyb-organoid greatly increased the channel number, this configuration might result in a slight pressure on the hHO. In order to make the top mMPC conformal to the hHO, we would reduce the culture medium to only submerge a little past the surface of the hHO. When the medium is reduced a lot, the drop of the liquid level will drive the top mMPC downward, thus squeezing the hHO. To minimize this potential damage, we must change the cyb-organoid medium daily; only one-third is changed gently each time.]”

- 2. (2.1) Data in Fig. 2 demonstrates the stretchability of the MPC meshwork. Nonetheless, given that the organoid system used in this study is grown externally and then inserted into the device, parameters such as bending stiffness and flexural rigidity have higher relevance and are required for quantitative characterization of the non-invasiveness aspect of the device.**

Author response: Thank you for pointing this out. As suggested by the reviewer, we used three mMPCs to do the bending test. The result is reported in Fig. 2g. The calculation was added in the methods section.

The revised text reads as follows on (Results section, Page 6 lines 196-198):

“[The bending test also performed great stretchability and easy deformation of the mMPC, and displacement of 500 μm requires only 0.005 N force (Fig. 2g, an average bending stiffness of 7.89×10^4 MPa).]”

(Methods section, Page 20 lines 624-632):

“[The bending test was performed at Shiyanjia Lab (www.shiyanjia.com), using a three-point bending method in an electronic universal testing machine. The bending stiffness was calculated using the following equation (We assumed the device was a completely thin sheet to replace the mesh shape for calculations and that the cross-section was rectangular):

$$\text{Bending stiffness } K = \frac{3FL}{2bh^2}$$

F is the force applied to the mMPC when the mMPC fractures; L is the distance between the two support points (124 mm); b is the width of the section (4.54 mm); and h is the thickness of the section (0.03 mm).]”

Fig. 2 | Characterization for the mMPC. **a**, (Left) the single 64-electrode mMPC. Scale bar: 5 mm. (Middle) the distribution of 64 electrodes in the mesh structure. Scale bar: 500 μm . (Right) the structure of electrodes and encapsulation layers. Scale bar: 100 μm . **b**, Optical images of the sandwich structure. Scale bar: 20 μm . **c**, SEM images of electrodes. **d**, Conformal attachment of the stretchable mMPC to the surface of a TPU ball. Scale bar: 1.5 mm. **e**, The mMPC under 400% strain. Scale bar: 5 mm. **f**, Two electrodes, circuits of the mMPC before and after 400% strain. Scale bar: 100 μm . **g**, The bending test of 3 mMPCs. **h**, Impedance of the mMPC across frequency before and after depositing PEDOT on electrodes, and under 100% strain.

Regarding the noninvasive characterization, we compared the morphology of the same HHO in a petri dish, keeping in two-layer mMPCs for 4 days, fixation by PFA, and

fluorescence immunostaining. The results showed that the encapsulation of two layers of mMPC did not leave obvious lattice traces on the surface of the hHO and did not result in the deformation of the hHO. Fluorescence immunostaining showed that the neuronal growth on the surface of hHO was not significantly affected and still had abundant axonal structure. This result is added in Supplementary Fig. 10d.

Supplementary Fig. 10 | Long-term change of hHOs in MEA and noninvasive assessment. *a*, (Left) The hHO on a commercial 2D MEA after culturing for 20 days. Scale bar: 500 μm . (Right) zoom-in view of the red box in the left figure showing the cell migration on the 2D MEA. Scale bar: 100 μm . *b*, (Left) bright-field images of hHOs on the normal mMPC after cocultured for 30 days and on the PLL-Laminin-coated mMPC for 20 days. Scale bar: 100 μm . *c*, The mMPC electrodes with and without PEDOT coating after culturing the hHO for 2 weeks. The red arrows point to the electrodes. Scale bar: 50 μm . *d*, **Non-invasive assessment.** The morphology of the same hHO in dish (Left), in the mMPC for 4 days (Middle), after fixation and immunostaining (Right). Scale bar: 100 μm in the staining image and 2 mm in others. *e*, The cross-sectional view showed the attachment boundary between the mMPC and the hHO. Scale bar: 100 μm . (Right) The estimated areas of the hHO and that covered by the mMPC.

The revised text reads as follows on (Results section, Page 13 lines 390-392):

“[Fluorescence immunostaining of the hHO which was removed from the mMPC revealed no obvious tissue or cellular damage on the hHO surface (Supplementary Fig. 10d).]”

(2.2) Furthermore, I am curious whether authors think the stretchability of the current device could be harnessed for minimally invasive growth in place strategies. If so, it may be beneficial to discuss the potential strategies in the discussion section.

Author response: Thank you for this suggestion. We are currently relying more on the softness of the mMPC to allow for non-invasive signal detection. As shown in the bending test results, although it is possible to bend the mMPC down by 10 mm using a very small force (less than 0.1N), we are concerned that if the mMPC is used in a growing brain tissue, such as implanted in the brain of a young mouse so that it stretches with the growth of the brain tissue, the long-term process from the device will damage the brain tissue due to the reaction force after the stretching. Thus, we do not recommend using mMPC for this situation now. However, our next plan is to find a suitable material with plastic deformation properties to replace the TPU and PU we are currently using, and we hope that the next device will be able to co-grow with the brain tissue without damaging the brain tissue. In this way, we will be able to monitor the growth and maturation of brain tissue *in situ* for a long time.

We have added the suggested content to the discussion section on (Discussion section, Page 18 lines 515-522):

“[More importantly, we relied more on the mMPC's softness to allow for non-invasive signal detection, although the mMPC has great stretchability. We are concerned that a stretched device would damage the tissue for a long time. Thus, our next plan is to find a suitable material with plastic deformation properties to replace the TPU and PU we are currently using, and we hope that the next device will be able to co-grow with the brain tissue without damaging the brain tissue. In this way, we will be able to monitor the growth and maturation of brain tissue in situ for a long time.]”

3. Another strength of the manuscript is the advanced number of electrodes (128) integrated into the MPC mesh structure that deserves further emphasis and representation in the paper. The only place where this is currently shown is the text and Fig. 6f that demonstrates recordings from only a handful of electrodes before glutamine treatment and a moderate number afterwards. Nonetheless, authors report around 40% recording from the channels (54 out of 128) in Fig. s8, which looks inconsistent with the data represented in Fig. 6f and weakens the claim for functionality of the electrode array. Is the 40% yield a typical number or the highest number ever recorded? Corresponding statistics should be provided. Also, how do authors interpret / explain the lack of recording from the other 74

channels?

Author response: Thank you for pointing this out. Our main challenge is that our equipment (Plexon system) has a transmission capacity of only 32 channels. Even though our mMPC has 128 channels, it is impossible to acquire signals from all channels simultaneously, and it must be done in 4 separate measurements. That also limited our access to the full surface neural network of the hHO. Thus, most of our plots were showing 32 channels of information.

Moreover, the experiment with glutamine treatment and that experiment with 54 active channels were not done on the same cyb-organoid system, so the two plots do not correspond. In the future, we will strive to upgrade our detection equipment to utilize mesh MPC's ability in organoid signal detection fully. We also added this limitation in the discussion section.

Our previous assays have been for individual hHOs over a hundred days old, of small size, and many of the electrodes have not touched the hHOs. Thus, despite having 128 channels, many of these electrodes have not come into direct contact with the organoid. That is the first reason for the low effective number of electrodes. Another reason is that a small number of electrodes are inherently defective during the fabrication of mMPCs. For example, in the process of filling MPC ink, even though we replenished the ink several times, there may be individual wire breaks; in the process of connecting the MPC circuits to the flexible printed circuits, if the anisotropic conductive adhesive is not tightly adhered locally, it may also cause individual electrodes to fail. Although we check under the microscope at every step, there is no guarantee that all 128 electrodes are intact at the end. These two main reasons led to the fact that the highest number of active CHs we detected before was 54.

In order to increase the number of active channels, we used hHO-fused assemblies. The effective channel numbers are listed in Supplementary Table 1. Among them, we detected the highest number of 85 active channels. This data shows that organoid size significantly affects the effective channel number. Meanwhile, this result suggested that our mMPC can detect neural circuits after the fusion of multiple organoids, which is crucial for the future detection of neural signals in organoid fusion complexes in multiple brain regions.

Supplementary Table 1 | Active channel numbers in different hippocampal cyb-organoids. Our Plexon system detected and recorded data from up to 32 channels simultaneously. Due to this limitation, detecting four times (128 channels) was necessary, and it was impossible to obtain data from 128 channels simultaneously.

Sample #	CH 1-32	CH 33-64	CH 65-96	CH 97-128	Total active channels
1 assembly	17	18	28	22	85
2 assembly	22	11	13	14	60
3 assembly	12	28	25	9	74
4 individual hHO	18	15	8	3	44
5 individual hHO	22	15	7	10	54

The revised text reads as follows on (Results section, Page 15 lines 428-433):

“[In a series of experimental tests, we detected signals from the single hHO or hHOs-fused assemblies, and the number of active channels was listed in Supplementary Table 1. The number of active channels was mainly affected by the hHO size. For signal acquisition of an individual hHO, we detected signals in a maximum of 54 channels, whereas when we used fused assemblies of 2 hHOs, the maximum number of active channels increased to 85.]”

(Discussion section, Pages 17-18 lines 506-515):

“[Compared with mature commercial systems, such as MaxWell Biosystem, HyperCAM Alpha, and Axion system, which offer an integrated platform with high-density electrodes in multiwell, real-time high-resolution scanning, and controlled environment, there is still much room for developing neural activity detection of hHOs by the mMPC. The ability of our Plexon instrument to detect only 32 channels of data at a time limits the mMPC's ability to record the complete neural activity of the hHO. In addition, obtaining long-term neuronal activity from 3D organoids requires a stable and healthy culture environment. However, there are many challenges in upgrading a system, including instrumentation, stimulation modules, temperature regulation, and gas regulation, to name a few. Adding other modules also introduces noise to the signal acquisition. These will be explored in the future.]”

- 4. The data in Fig. 6e suggests higher spike rates from the MPC mesh compared to the 2D MEA. The authors interpret this as an advantage of examining 3D neural tissues compared to the monolayer neuron cultures, which tends to be likely. Nonetheless, how can we be sure that this is not due to the mechanical assault of squeezing the organoid between the meshes or its downstream effect? Perhaps this should be reexamined and demonstrated with a 64-channel mesh where the original organoid shape is retained?**

Author response: Thank you for this suggestion. The hHO was not significantly deformed. We consider the scenario where the two meshes will squeeze the hHO, especially the top mMPC, and we must admit that the pressure is present and unavoidable during the use. However, to minimize the pressure from the top mesh, we have improved the assembly of the cyb-organoid as much as possible. We think the pressure on the top mesh may have two major sources. If the mesh is stretched, it will exert a force on the hHO. Second, suppose the culture medium is gradually reduced during the changing medium or evaporated after being placed in the incubator. In that case, the top mesh will move downward due to a lower liquid level, which may result in compression.

To address these issues, our improvements include:

- (1) When assembling the two-layer mMPCs, we placed a TPU ball between the two layers of mMPCs to form a reserved space such that they will accommodate the spherical shape of the hHO. When we placed the hHO in the middle of the mMPCs, there was sufficient space to accommodate the hHO. Thus, the top mMPC will not be stretched

so that little external force is applied to the hHO. We added this picture in Supplementary Fig. 9a-b.

Supplementary Fig. 9 | Assembling of 128-channel mMPC and biocompatibility of mMPC. **a**, The integrated device coupling the mMPC with culture dishes. (Left): After removing the glass bottom, a confocal 35-mm glass-bottom dish was covered above the circuits of the mMPC, with the mesh part remaining inside the dishes. PDMS simultaneously fixed them in a large 100-mm dish. The end of the flexible printed circuit was left outside the dish to connect to the Plexon instrument. (Right): Assembly of single- and double-layer mMPCs with the 35-mm dish. The TPU ball in the center was to reserve the space the hHO may need. **b**, (Left) The two-layer mMPCs with hHO's culture medium. After remove the TPU ball, there existed some space between these two layers. (Middle and right) The top and the side view after placing one hHO into the two-layer mMPC. Scale bar: 5 mm. When the culture solution was

just submerged over the hHO, the top mMPC covered the surface of the hHO without significant squeezing.
c, Biocompatibility. The mouse hippocampal cells grew in the mMPC for 30 days. Scale bar: 100 μm .

- (2) In order for the top mMPC to fit the hHO better, after placing the hHO, we removed a certain amount of culture medium until the top mMPC slowly moved down to the surface of the hHO, at which time the culture medium in the dish was generally a little bit above the upper surface of the hHO. The top view of the cyb-organoid tended to give the impression that the hHO was being squeezed, so we replaced the top view via the picture with a certain tilt angle. We also added a picture of the organoid when it was in the petri dish for comparison. This change can be found in Fig. 5a. Our mMPC can wrap around the hHO but maintains its 3D morphology. Also, in Supplementary Fig. 9b, we added a side view. Although the petri dish and culture medium affected capturing the side view, we can still see that the top mMPC is above the hHO, wrapping around the hHO as a curved surface instead of directly squeezing the hHO. This approach greatly reduces the pressure that may be caused by the mesh MPC. We also describe this assembly in detail in the method.

Fig. 5 | The hippocampal cyb-organoid platform. **a,** Integration of the 128-channel mMPC and the hHO. (Upper) The hHO was stayed in a low-attachment dish. (Lower) The hHO was enveloped in two-layer mMPC. Scale bar: 1 mm. Notably, there was no significant deformation when the hHO was inserted between two layers of mMPC. **b,** The bottom view captured by a microscope. Scale bar: (Left) 500 μm ,

(right) 100 μm . **c**, Confocal images of mouse hippocampal cells culturing in the mMPC for 30 days. Scale bar: 100 μm . **d**, Confocal images of MAP2⁺ neurons, GFAP⁺ astrocytes and EGFP⁺ cells in the cyb-organoid platform. Scale bar: 100 μm . **e**, The cross-sectional view of the conformal attachment between the mMPC and the hHO. The arrowhead points to the mMPC. Scale bar: 100 μm .

- (3) To minimize the effect of decreasing culture medium during the changing medium, we changed the medium daily, placed fresh culture medium into the dish first, and then removed one-third of the medium. This approach is described in the method.

Thus, we have minimized the possible pressure on the top mMPC, and these images showed that the hHO was not significantly deformed (Supplementary Fig. 9b and Fig. 5a). To confirm whether the top mMPC affected the electrical activity, we compared the neural signals detected with and without the top mMPC using the same cyb-organoid in both cases. First, we detected neural signals through the bottom mMPC in the case with two layers of mMPC. We then carefully cut the top mMPC and gently lifted it off the hHO surface, after which we went back to detecting the signal through the bottom mMPC. There was no difference in the average spike rates of multiple active channels under these two conditions. This result demonstrated that the top mMPC does not seriously affect the neural activity of the hHO. This result can be found in Supplementary Fig. 12a. We also added the potential effects in the discussion.

The revised text reads as follows on (Results section, Page 15 lines 443-446):

“[Concerned that pressure from the top mMPC could affect the electrical activity of the hHO, we compared the neural signals detected with and without the top mMPC using the same cyb-organoid in both cases. A comparison of the average spike rates of multiple active channels over two minutes showed no significant difference (Supplementary Fig. 12a).]”

(Discussion section, Page 17 lines 498-504):

“[Although the sandwich structure of our cyb-organoid greatly increased the channel number, this configuration might result in a slight pressure on the hHO. In order to make the top mMPC conformal to the hHO, we would reduce the culture medium to only submerge a little past the surface of the hHO. When the medium is reduced a lot, the drop of the liquid level will drive the top mMPC downward, thus squeezing the hHO. To minimize this potential damage, we must change the cyb-organoid medium daily; only one-third is changed gently each time.]”

Supplementary Fig. 12 | a, *The comparison of spike rates recording in the bottom mMPC before and after cutting the top mMPC.* **b**, *The change of amplitude of glutamine-induced spikes.*

5. It is argued that the mesh-type 3D MEA is applied to the hHO for the first-time. Was the first demonstration possible because of technological advancement of the mesh-type? Or do other existing 3D MEAs have capability to be applied to hHO or not? Or were the existing devices not just applied to hHO? Clarification on this would be helpful for understanding the state-of-the-art technologies.

Author response: Thank you for this suggestion. We want to emphasize that this is the first mesh 3D MEA applied to the hHO. Although other types of 3D MEA have also been used to detect signals from brain organoids, previous studies focused on cerebral or

cortical organoids because they already have more mature generation protocols. The previously reported 3D MEA can certainly be applied to the hHO, and similarly, our mMPC can be applied to other brain organoids. In this work, we have solved two problems simultaneously, generating hippocampal organoids and preparing 3D MEAs with high channel numbers, and the assembled cyb-organoid have the potential for future in vitro applications to study important hippocampus-related neurological disorders.

To make the comparison more explicit, we have clarified the statement on the application of other existing 3D MEAs in the introduction on (Page 2 lines 79-83):

*“[Existing 3D MEAs specifically formulated for brain organoids were **only** applied to cerebral²⁰⁻²² or cortical organoids²³ and have yet to be employed on hHOs, **as there exist more established protocols for generating the cerebral organoid and the cortical organoid.** Generating hHOs is challenging because more accurate manipulation of signaling factors is required to induce the differentiation process.]”*

As suggested by the reviewer, we have added the content in the discussion section (Page 18 lines 524-526).

*“[The mMPC is the first 3D MEA system for detecting neural signals in hHOs. **However, the mMPC is not limited to the hHO, and other brain organoids and heart organoids can be detected.**]”*

6. The authors mention the liquid metal electrodes may degrade during the 1-month recording period. Will the degraded metal ions be toxic to the organoids and further affect their activity?

Author response: Thank you for pointing this out. The biosafety of MPC electrodes has been characterized in several publications (ref. 28-30). This time, to better capture the changes of both neuronal cells and the mMPC in prolonged co-culture, we dissociated and raised hippocampal cells of suckling mice on the mMPC and continued observing the same location for over a month. The results were added to Fig. 5c and Supplementary Fig. 9c. Neural cells grew well on the mMPC with significant proliferation. Abundant axons grew on the fully encapsulated TPU-MPC-PU structure and the exposed electrode location. This result indicated that our mMPC was completely safe for nerve cells, even though there was a small amount of liquid metal degradation in the electrode portion during prolonged use.

In addition, the degradation of liquid metal on the electrode can cause an increase in resistance or even lead to some electrodes to fail. Thus, it is difficult for us to characterize whether the degraded liquid metal affects the electrical activity of the hHO through electrical activity. However, the fact that neural cells grew healthily on mMPC for a long period can also indicate the biocompatibility of mMPC.

Fig. 5 | The hippocampal cyb-organoid platform. **a**, Integration of the 128-channel mMPC and the hHO. (Upper) The hHO was stayed in a low-attachment dish. (Lower) The hHO was enveloped in two-layer mMPC. Scale bar: 1 mm. Notably, there was no significant deformation when the hHO was inserted between two layers of mMPC. **b**, The bottom view captured by a microscope. Scale bar: (Left) 500 μm , (right) 100 μm . **c**, Confocal images of mouse hippocampal cells culturing in the mMPC for 30 days. Scale bar: 100 μm . **d**, Confocal images of MAP2⁺ neurons, GFAP⁺ astrocytes and EGFP⁺ cells in the cyb-organoid platform. Scale bar: 100 μm . **e**, The cross-sectional view of the conformal attachment between the mMPC and the hHO. The arrowhead points to the mMPC. Scale bar: 100 μm .

Supplementary Fig. 9 | Assembling of 128-channel mMPC and biocompatibility of mMPC. **a**, The integrated device coupling the mMPC with culture dishes. (Left): After removing the glass bottom, a confocal 35-mm glass-bottom dish was covered above the circuits of the mMPC, with the mesh part remaining inside the dishes. PDMS simultaneously fixed them in a large 100-mm dish. The end of the flexible printed circuit was left outside the dish to connect to the Plexon instrument. (Right): Assembly of single- and double-layer mMPCs with the 35-mm dish. The TPU ball in the center was to reserve the space the hHO may need. **b**, (Left) The two-layer mMPCs with hHO's culture medium. After remove the TPU ball, there existed some space between these two layers. (Middle and right) The top and the side view after placing one hHO into the two-layer mMPC. Scale bar: 5 mm. When the culture solution was just submerged over the hHO, the top mMPC covered the surface of the hHO without significant squeezing. **c**, Biocompatibility. The mouse hippocampal cells grew in the mMPC for 30 days. Scale bar: 100 μ m.

The revised text reads as follows on (Results section, Page 13, lines 369-374):

“[Because of the concern about the potential risk of leakage of liquid metal on the electrode locations and possible toxicity, we cultured mouse hippocampal cells on the mMPC to verify its long-term biocompatibility. Hippocampal cells could proliferate and grow stably on the mMPC for more than 30 days (Supplementary Fig. 9c), and a large number of axonal structures were present at electrode spots (Fig. 5c), indicating that the mMPC possessed excellent biocompatibility.]”

**7. How could the TPU solution fill the mesh-shaped microchannel in PDMS slab #2?
By capillary force or pressurization?**

Author response: Thank you for asking this question and letting us know that more than our description of the fabrication process is needed for the reader to understand easily. After we aligned the PDMS slab #2 for the TPU layer and dropped the TPU solution on one PDMS side, the TPU solution flowed into the mesh microchannels due to capillary force without applying pressure. The lower the TPU solution concentration, the faster the inflow. A higher concentration of TPU could also be used, but it needs vacuuming to help the TPU solution flow into the mesh microchannels quickly.

We have added arrows to the fabrication schematic to help readers understand, as shown in Fig.1b (Steps ii and v). We have also described the TPU solutions entering the mesh microchannels in the method.

The revised text reads as follows on (Methods section, Page 19 lines 592-597):

“[PDMS slab #2 (for the TPU substrate layer) was aligned with PDMS slab #1 under a stereo microscope. Subsequently, the TPU solution was carefully dropped on one side of PDMS slab #2, and the TPU solution (3 wt%) flowed into the mesh microchannels inside PDMS slab #2 due to capillary force without applying pressure. The lower the TPU solution concentration, the faster the inflow. A higher concentration of TPU could also be used, but it needs vacuuming to help the TPU solution flow into the mesh microchannels quickly.]”

Fig. 1 | Overview of coupling the mMPC with the hHO and design of the mMPC. a, Schematic diagram of coupling the mMPC with the hHO. b, Fabrication process of the mMPC. Scale bar: (i) 500 μm . (ii-iii) 200 μm . (vii) 200 μm .

Reviewer 3

Wu and co-authors present a device that can detect electrophysiological activity from human hippocampal organoids. The work presents two fundamental aspects: the development of a device that goes beyond the state of the art for recording the electrophysiological activity of a floating system such as the organoids and an improvement in the protocol for the human hippocampal organoid. However, I find that the work is poorly organized and does not highlight what could have been the real strong point: the device.

In particular:

1. (1.1) In paragraph lines 56-72, the authors highlight the state of the art of the devices available and developed over the years to record the electrophysiological activity from cellular preparations in vitro, highlighting their limitations. However, they do not mention the work of Sosia 2020, which presents a flexible device with 256 channels able to record the basal activity and stimulate at the same time. Moreover, it allows recording the third dimension.

Author response: We think the ref you mentioned is the work published in Lab on a Chip by Soscia in 2020. This work integrated electrodes on 90-micron-wide probes and vertically actuated the flexible probes to form a 3D MEA. There are several differences between their work and the existing MEAs applied for brain organoids: i) The Soscia 2020 work involves mostly dissociated neurons in hydrogels, which are different from human brain organoids; ii) The Soscia 2020 work involves an array of 90-micron-wide probes which in our experience might damage the organoid should these probes be directly inserted; iii) even though the probes reported by Soscia 2020 have some mechanical flexibility, we suspect that these probes cannot be easily bent and stretched like ours.

However, we thank you for pointing this out and opening our minds. To make our discussion more inclusive, we have also added a description of this type of MEA to the introduction.

The revised text reads as follows on (Introduction section, Page 2 lines 60-68):

“[The electrical information in neural systems in vitro is widely detected through a planar MEA^{12,13} and calcium imaging¹⁴⁻¹⁶. Electrodes in commercial MEAs distribute over a small surface area at the bottom of a single well. Thus, these devices cannot match hHOs in the suspension culture system. Implantable electrodes designed for animals are also unsuitable for hHOs due to a mismatch in shape and size^{13,17,18}. Single or arrays of needle electrodes may potentially be inserted into brain organoids to detect signals, like vertically-arrayed 3D MEAs¹⁹. Calcium imaging can allow live imaging, which provides information about electrical activity on a larger scale but is dependent on imaging capabilities.]”

(1.2) On the other hand, the proposed device "envelops" the whole organoid but does not acquire information related to signal transmission within the structure. How can the presented device be considered better?

Author response: We thank the reviewer for the feedback. The advantage of this sandwich-wrapped mMPC is that it can noninvasively detect neural signals on the upper and lower surfaces of brain organs. Previous semi-wrapped electrodes with only 3 or 25 channels can only detect signals from the lower surface of the brain organ, which is a much smaller number of channels than our 128-channel mMPC.

As suggested by the reviewer, we detected more neural signals from hHOs cultured for more than one hundred days by our mMPC. During the signal acquisition, we observed the neural signals from two channels that always appeared in pairs. Moreover, these paired signals always occurred one after the other, and the time interval was stable on millisecond time scales. These data suggested the existence of signal transmission between the neurons of those two channels. Our hHOs, more than one hundred days old, have grown functional synapses between neurons, and our mMPC was able to detect this synchrony. This data we have added is in Fig. 6f.

In addition, we found oscillatory network activity, i.e., coordinated bursts of spikes across electrodes, in the hippocampal cyb-organoid. These indicated that the mesh MPC could characterize the functional network on the surface of the hHO with excitatory and inhibitory neurons. We add this data in Fig. 6g.

However, we must admit that the current detection of hHO signaling by mesh MPC still needs to be improved. The main limitation is that the instrumentation we used (Plexon system) has a transmission capacity of only 32 channels. Even though our mesh MPC has 128 channels, acquiring signals from all channels simultaneously is impossible, thus we have to carry out the measurements in 4 separate experiments. That also limited our access to understanding the complete surface neural network of the hHO. However, the fact that 32 channels could detect neural activities has demonstrated the potential of the mMPC. In the future, we will strive to upgrade our detection equipment to utilize mMPC's ability in organoid signal detection fully. This limitation has been added in the discussion section.

Fig. 6 | Electrical activity recording. **a**, (Upper) Raster plots of neural spikes, and (lower) the continuous signal in a channel of the mMPC. **b**, Continuous recording for a spike. **c**, The uniform spike waveforms and the average waveform in a channel of the mMPC. **d**, Two different spikes appeared in the same channel (upper) and the sorting results (lower). **e**, The comparison of spike rates detected by the mMPC and the 2D MEA (these channels covered by the hHO). **f**, The synchronous signals between two channels. (Upper) Raster plots of neural spikes in these two channels, and (lower) continuous recording for two spikes. The time interval was less than 1 ms. **g**, The oscillatory network activity of the hHO. (Upper) the population activity histogram and (lower) the raster plot (black) identified bursts of spiking detected on 32 channels and coordinated bursts (red) of activity across channels. **h**, Raster plots before and after

Glutamine supplement. *i*, The impedance of the mMPC before and after integration with the hHO for 1 month.

The revised text reads as follows on (Results section, Page 15 lines 428-441):

“[In a series of experimental tests, we detected signals from the single hHO or hHOs-fused assemblies, and the number of active channels was listed in Supplementary Table 1. The number of active channels was mainly affected by the hHO size. For signal acquisition of an individual hHO, we detected signals in a maximum of 54 channels, whereas when we used fused assemblies of 2 hHOs, the maximum number of active channels increased to 85. Simultaneous signals always appeared in pairs in two channels. We found that the time interval of these paired signals was less than 1 ms. They always appeared one after the other (Fig. 6f). This observation suggested that neurons on the hHO formed functional synapses capable of transmitting signals and were detected by the mMPC. In addition, neural activity was detected on the hHO (Fig. 6g). As potential evidence of neural network maturation, the synchrony was proposed to reflect a balance between ExN and InN, and coordinate neuronal communication in the hHO⁴². These results highlight the potential of the hippocampal cyb-organoid applied to understand the development and disorders in the hippocampus.]”

(Discussion section, Pages 17-18 lines 506-515):

“[Compared with mature commercial systems, such as MaxWell Biosystem, HyperCAM Alpha, and Axion system, which offer an integrated platform with high-density electrodes in multiwell, real-time high-resolution scanning, and controlled environment, there is still much room for developing neural activity detection of hHOs by the mMPC. The ability of our Plexon instrument to detect only 32 channels of data at a time limits the mMPC's ability to record the complete neural activity of the hHO. In addition, obtaining long-term neuronal activity from 3D organoids requires a stable and healthy culture environment. However, there are many challenges in upgrading a system, including instrumentation, stimulation modules, temperature regulation, and gas regulation, to name a few. Adding other modules also introduces noise to the signal acquisition. These will be explored in the future.]”

We have described the advantages of our mMPC in the discussion section on (Page 17 lines 476-487):

“[Our fabrication process of the mMPC is easy-to-operate and low equipment-dependent using low-cost materials. The mMPC offered advantages over current 3D MEAs, including conformal attachment to organoids and multichannel recording, enabling the acquisition of as many nerve signals as possible from brain organoids. The free deformation and 500% stretchability performance far exceeded those offered by current 3D MEAs fabricated by photolithography. The 128-channel configuration of the mMPC exceeded that of existing 3D devices developed for brain organoids. More importantly, the design of the two-layer mMPC to conformally enclose the organoid filled the gap in detecting neural activities from the entire surface of organoids in previous studies. Our mMPC detected neural spike, synchronization, and oscillation activities in the hHO and the hHO assembly. These results indicated that hHOs grow a complex neural network.]”

2. The authors point out that their device is able to record from a larger area than other MEA devices, but the reported percentage of active electrodes is about 50%. How can the authors ensure that their device is still a viable alternative?

Author response: Thank you for pointing this out. Our previous measurements have been for individual hHOs over a hundred days old, of small size, and many of the electrodes have not touched the hHOs. Thus, despite having 128 channels, many of these electrodes have not come into direct contact with the organoid. That is the first reason for the low number of active channels. Another reason is that a small number of electrodes are inherently defective during the fabrication of mMPC. For example, in the process of filling MPC ink, even though we replenished the ink several times, there may be individual wire breaks; in the process of connecting the MPC circuits to the flexible printed circuits, if the anisotropic conductive adhesive is not tightly adhered locally, it may also cause individual electrodes to fail. Although we check under the microscope at every step, there is no guarantee that all 128 electrodes are intact at the end. These two main reasons led to the fact that the highest number of active channels we detected before was 54. In addition, even with access to the electrodes, the neurons may not have electrical activities at that location, a phenomenon we also observed on a commercial 2D MEA, as shown in Supplementary Fig. 9b. Thus, warranting the active channel numbers is beyond the scope in this manuscript, but 128 channels our mMPC offered have much more than the existing 3D MEA designed for brain organoids.

In order to increase the number of active channels, we used hHO-fused assemblies. The effective channel numbers are listed in Supplementary Table 1. Among them, we detected the highest number of 85 active channels. This data shows that organoid size significantly affects the effective channel number. Meanwhile, this result suggests that our mMPC can detect neural circuits after the fusion of multiple organoids, which is crucial for the future detection of neural signals in organoid fusion complexes in multiple brain regions.

Supplementary Table 1 | Active channel numbers in different hippocampal cyb-organoids. Our Plexon system detected and recorded data from up to 32 channels simultaneously. Due to this limitation, detecting four times (128 channels) was necessary, and it was impossible to obtain data from 128 channels simultaneously.

Sample #	CH 1-32	CH 33-64	CH 65-96	CH 97-128	Total active channels
1 assembly	17	18	28	22	85
2 assembly	22	11	13	14	60
3 assembly	12	28	25	9	74
4 individual hHO	18	15	8	3	44
5 individual hHO	22	15	7	10	54

The revised text reads as follows on (Results section, Page 15 lines 428-433):

“[In a series of experimental tests, we detected signals from the single hHO or hHOs-fused assemblies, and the number of active channels was listed in Supplementary Table 1. The number of active channels was mainly affected by the hHO size. For signal acquisition of

an individual hHO, we detected signals in a maximum of 54 channels, whereas when we used fused assemblies of 2 hHOs, the maximum number of active channels increased to 85.]”

- 3. The device is built using the properties of PDMS; however, the literature shows how this material can influence the effects of a chemical compound due to its absorption and adsorption processes (Toapke et al., 2006).**

Author response: Thank you for pointing this out. Although Toapke demonstrated that the absorption of small molecules by PDMS in a microfluidic device can significantly change the solution concentration and potentially alter the experimental results, PDMS is not a component in our mMPC. PDMS is only used in the fabrication process to prepare three slabs with microchannels, working as the molds for MPC ink, TPU, and PU layers. All the PDMS slabs would be removed at the end of the fabrication process, and only three components, TPU, PU, and liquid metal, were present in our mMPC. We mainly rely on the fact that TPU and PU solutions are not even in affinity with PDMS, which helps us to easily transfer the MPC circuits from the PDMS mold to the TPU. The light-transmitting nature of PDMS is also utilized, making the alignment between the three PDMS slabs easy. Thus, the absorption of PDMS for small molecules does not affect fabrication in this work, much less our detection of neural signals.

We have stated the components in the introduction section (Page 3 lines 98-101):

“[In this article, we report a method to fabricate the flexible and stretchable mMPC composed of gallium indium (Galn) alloy and elastic polymers (thermoplastic polyurethane, TPU, and polyurethane, PU), aiming to detect signals of hHOs (Fig.1a).]”

To clarify this point more clearly, we added the statement to text on the fabrication section (Page 4 lines 148-150)

“[The mMPC was obtained after removing all PDMS slabs. In the end, the final device has no PDMS in it, only containing three components: TPU, PU, and liquid metal.]”

- 4. The focus (also considering the title reported and chosen by the authors) should be the device. However, all manufacturing and testing processes have been reported in the Supplementary.**

Author response: We think this is an excellent suggestion. We reorganized the contents of Fig. 1 and Fig. 2. The part about fabrication in the Supplementary was integrated into Fig. 1, while the characterization of the morphology, mechanical properties, and electrical properties of mMPC were concentrated in Fig. 2.

Fig. 1 | Overview of coupling the mMPC with the hHO and design of the mMPC. a, Schematic diagram of coupling the mMPC with the hHO. b, Fabrication process of the mMPC. Scale bar: (i) 500 μm . (ii-iii) 200 μm . (vii) 200 μm .

5. Looking at the images shown of the organoids inside the device, the first appear quite deformed. How can the authors prove that this substantial modification of the geometry of the cell preparation does not alter its functioning?

Author response: Thank you for pointing this out. The hHO was not significantly deformed. We also consider the scenario where the two meshes will squeeze the hHO, especially the top mMPC, and we must admit that the pressure is present and unavoidable during the use. However, to minimize the pressure from the top mesh, we have improved the

assembly of the cyb-organoid as much as possible. We think the pressure on the top mesh may have two major sources. If the mesh is stretched, it will exert a force on the hHO. Second, suppose the culture medium is gradually reduced during the changing medium or evaporated after being placed in the incubator. In that case, the top mesh will move downward due to a lower liquid level, which may result in compression.

To address these issues, our improvements include:

- (1) When assembling the two-layer mMPCs, we placed a TPU ball between the two layers of mMPCs to form a reserved space such that they will accommodate the spherical shape of the hHO. When we placed the hHO in the middle of the mMPCs, there was sufficient space to accommodate the hHO. Thus, the top mMPC will not be stretched so that little external force is applied to the hHO. We added this picture in Supplementary Fig. 9a-b.
- (2) In order for the top mMPC to fit the hHO better, after placing the hHO, we removed a certain amount of culture medium until the top mMPC slowly moved down to the surface of the hHO, at which time the culture medium in the dish was generally a little bit above the upper surface of the hHO. The top view of the cyb-organoid tended to give the impression that the hHO was being squeezed, so we replaced the top view via the picture with a certain tilt angle. We also added a picture of the organoid when it was in the petri dish for comparison. This change can be found in Fig. 5a. Our mMPC can wrap around the hHO but maintains its 3D morphology. Also, in Supplementary Fig. 9b, we added a side view. Although the petri dish and culture medium affected capturing the side view, we can still see that the top mMPC is above the hHO, wrapping around the hHO as a curved surface instead of directly squeezing the hHO. This approach greatly reduces the pressure that may be caused by the mesh MPC. We also describe this assembly in detail in the method.
- (3) To minimize the effect of decreasing culture medium during the changing medium, we changed the medium daily, placed fresh culture medium into the dish first, and then removed one-third of the medium. This approach is described in the method.

Supplementary Fig. 9 | Assembling of 128-channel mMPC and biocompatibility of mMPC. **a**, The integrated device coupling the mMPC with culture dishes. (Left): After removing the glass bottom, a confocal 35-mm glass-bottom dish was covered above the circuits of the mMPC, with the mesh part remaining inside the dishes. PDMS simultaneously fixed them in a large 100-mm dish. The end of the flexible printed circuit was left outside the dish to connect to the Plexon instrument. (Right): Assembly of single- and double-layer mMPCs with the 35-mm dish. The TPU ball in the center was to reserve the space the hHO may need. **b**, (Left) The two-layer mMPCs with hHO's culture medium. After remove the TPU ball, there existed some space between these two layers. (Middle and right) The top and the side view after placing one hHO into the two-layer mMPC. Scale bar: 5 mm. When the culture solution was just submerged over the hHO, the top mMPC covered the surface of the hHO without significant squeezing. **c**, Biocompatibility. The mouse hippocampal cells grew in the mMPC for 30 days. Scale bar: 100 μ m.

Fig. 5 | The hippocampal cyb-organoid platform. **a**, Integration of the 128-channel mMPC and the hHO. (Upper) The hHO was stayed in a low-attachment dish. (Lower) The hHO was enveloped in two-layer mMPC. Scale bar: 1 mm. Notably, there was no significant deformation when the hHO was inserted between two layers of mMPC. **b**, The bottom view captured by a microscope. Scale bar: (Left) 500 μ m, (right) 100 μ m. **c**, Confocal images of mouse hippocampal cells culturing in the mMPC for 30 days. Scale bar: 100 μ m. **d**, Confocal images of MAP2⁺ neurons, GFAP⁺ astrocytes and EGFP⁺ cells in the cyb-organoid platform. Scale bar: 100 μ m. **e**, The cross-sectional view of the conformal attachment between the mMPC and the hHO. The arrowhead points to the mMPC. Scale bar: 100 μ m.

The revised text reads as follows on (Results section, Pages 12-13 lines 358-367):

“[To enable the integration of the mMPC with the hHO and facilitate the connection with the Plexon instrument, we assembled the mMPC and culture dishes (Supplementary Fig. 9a). To prevent the two-layer meshes from severely squeezing on the soft hHO, we placed a TPU ball of ~3 mm when assembling the device to reserve the space the hHO may need (Supplementary Fig. 9a-b). After sterilization, the cultured medium was added to the dish, and the TPU ball was removed from the side; the hHO was inserted into the space between the top and bottom mMPCs. The oblique view and the side view showed that the top mMPC varies with the undulation of the hHO surface without causing significant compression on the hHO (Fig. 5a and Supplementary Fig. 9b), indicating the conformal adhesion of the mMPC to the surface of the hHO.]”

(Methods section, Pages 22-23 lines 719-733):

“[The mMPC was fixed with a 35 mm confocal dish which had its glass bottom removed, in a 100 mm dish, leaving the end of flexible printed circuits outside the 100 mm dish (Supplementary Fig. 9a). When assembling the two-layer mMPC, use a TPU ball to pre-simulate the replacement of the hHO and place it between the two layers. All relevant materials, including tapes, dishes, and devices, were sterilized by overnight UV light illumination. Before coculturing with the hHO, the mMPC was soaked in PBS to promote electrode and medium attachment. After removing the TPU ball from the side between the top and bottom meshes, the hHO was inserted into the two-layer mMPC, replacing PBS with 2.5 mL differentiation medium 2 (supplement with BDNF and GDNF) in the 35 mm dish. When the medium was enough to submerge the mMPC completely, the upper mMPC would float up to create a gap between the two layers. The hHO was then aspirated using a wide bore tip and gently placed in the gap. Finally, a small amount of medium was carefully removed until the upper layer adhered to the hHO. The hHO was maintained with the mMPC system in a 37 °C incubator, and the medium was changed daily, only one-third each time.]”

Thus, we have minimized the possible pressure on the top mMPC, and these images showed that the hHO was not significantly deformed (Supplementary Fig. 9b and Fig. 5a). To confirm whether the top mMPC affected the electrical activity, we compared the neural signals detected with and without the top mMPC using the same cyb-organoid in both cases. First, we detected neural signals through the bottom mMPC in the case with two layers of mMPC. We then carefully cut the top mMPC and gently lifted it off the hHO surface, after which we went back to detecting the signal through the bottom mMPC. A comparison of the average spike rates of multiple active channels over two minutes showed no significant difference. This result demonstrated that the top mMPC does not seriously affect the neural activity of the hHO. This result can be found in Supplementary Fig. 12a. We also added the potential effects in the discussion.

Supplementary Fig. 12 | a, *The comparison of spike rates recording in the bottom mMPC before and after cutting the top mMPC. b*, *The change of amplitude of glutamine-induced spikes.*

The revised text reads as follows on (Results section, Page 15 lines 443-446):

“[Concerned that pressure from the top mMPC could affect the electrical activity of the hHO, we compared the neural signals detected with and without the top mMPC using the same cyb-organoid in both cases. A comparison of the average spike rates of multiple active channels over two minutes showed no significant difference (Supplementary Fig. 12a).]”

(Discussion section, Page 17 lines 498-504):

“[Although the sandwich structure of our cyb-organoid greatly increased the channel number, this configuration might result in a slight pressure on the hHO. In order to make the top mMPC conformal to the hHO, we would reduce the culture medium to only

submerge a little past the surface of the hHO. When the medium is reduced a lot, the drop of the liquid level will drive the top mMPC downward, thus squeezing the hHO. To minimize this potential damage, we must change the cyb-organoid medium daily; only one-third is changed gently each time.]”

6. In general, for all the analyzes reported, it is not clear to me what the number of organoids tested is, especially those coupled with the developed device. What is the success rate?

Author response: Thank you for pointing this out. In a typical batch, we generate hHOs in 6 wells, typically more than 4 of them can successfully produce electrical signals, representing a success rate of ~66.7%. This batch is a significant improvement over previous batches, which typically have success rates of 40-50%. We are improving both the culturing condition and the fabrication methodology to improve the success rate. The active channel number of these mMPCs that successfully detected the signal has been counted in Supplementary Table 1.

Supplementary Table 1 | Active channel numbers in different hippocampal cyb-organoids. Our Plexon system detected and recorded data from up to 32 channels simultaneously. Due to this limitation, detecting four times (128 channels) was necessary, and it was impossible to obtain data from 128 channels simultaneously.

Sample #	CH 1-32	CH 33-64	CH 65-96	CH 97-128	Total active channels
1 assembly	17	18	28	22	85
2 assembly	22	11	13	14	60
3 assembly	12	28	25	9	74
4 individual hHO	18	15	8	3	44
5 individual hHO	22	15	7	10	54

7. In general, it would be worth investigating all the aspects related to improving the proposed device compared to those on the market, especially referring to high-density commercial systems that allow you to record from thousands of electrodes simultaneously (MaxWell and 3Brain, for example).

Author response: We have added the suggested content to the manuscript on (Discussion section, Pages 17-18 lines 506-515).

“Compared with mature commercial systems, such as MaxWell Biosystem, HyperCAM Alpha, and Axion system, which offer an integrated platform with high-density electrodes in multiwell, real-time high-resolution scanning, and controlled environment, there is still much room for developing neural activity detection of hHOs by the mMPC. The ability of our Plexon instrument to detect only 32 channels of data at a time limits the mMPC's ability to record the complete neural activity of the hHO. In addition, obtaining long-term neuronal activity from 3D organoids requires a stable and healthy culture environment. However, there are many challenges in upgrading a system, including instrumentation, stimulation modules, temperature regulation, and gas regulation, to name a few. Adding other modules

also introduces noise to the signal acquisition. These will be explored in the future.]"

Minor revision:

- 8. Pay attention to the abbreviations, always writing them the same, for example, wnt3a or WNT3a?**

Author response: Thank you for pointing this out. These abbreviations have been corrected, and all texts have been harmonized into "Wnt3a".

Reviewer 4

Wu et al. have developed a mesh multi electrodes array that can form a 3D neuro-interface with human hippocampal organoids (hHOs). The electrodes are a liquid metal-polymer conductor (MPC)-based system that are flexible and stretchable which enables the mesh MPC (mMPC) to couple with hHOs in a conformal manner. They also found a culture condition for generating hHOs by incorporating Wnt3a and purmorphamine (the SHH signalling activator) that suppresses the expansion of choroid plexus (ChP) during dorsomedial telencephalon (DMT) induction where the hippocampal primordial develops from. They have identified the cell types that compose of the organoids using single-cell RNA sequencing (scRNAseq) and immunofluorescence imaging. Overall, the mMPC system shows a great potential to measure the electrical activity of brain organoids in 3D, but there are some important points that would be important to be addressed for this manuscript to be published.

1. scRNAseq

(1.1) Wu et al. have performed scRNAseq to identify the cell types of the organoids, but I have a major concern about the cells that were used for sequencing. According to the methods section for scRNAseq, the hHOs were dissected into smaller tissues and replaced on PLL-laminin-coated flask to minimise the information loss and capturing more cells and even grow further in this condition for 3-4 days. To my knowledge, to identify the cell type of the organoids, the cells should be dissociated and collected directly from the organoid as a 3D multicellular tissue for sequencing. As soon as the cells from the organoid attach to the flask and be cultured in 2D, the different microenvironment between the cells, such as the different cell-cell interaction, mechanical cues, nutrient or oxygen delivery that the cell sense between 2D and 3D, the cells might differentiate into different cell types than the original 3D hHOs that the current scRNAseq data might not faithfully represent the actual cell types of the hHOs when it was originally 3D. For this reason, the researchers in the organoid field pool as many organoids as possible to have enough cell numbers for scRNAseq. Thus, scRNAseq of the cells that are directly dissociated and collected from the hHOs would be essential to accurately identify the cell types.

Author response: Thank you for pointing this out and we agree with the reviewer's assessment. Unfortunately, the previous time we prepared samples for single-cell RNA sequencing was during the COVID-19 epidemic. At that time, we needed to send the hHOs elsewhere for dissociation, barcoding, and library construction. Leaving the incubator for a long time resulted in the hHOs being in very poor condition, with low cell viability after dissociation, and thus unable to do the library construction. We had no choice but to dissect hHOs into smaller tissues and seed them in 2D culture dishes for a few days to allow many cells to migrate out of the tissue to increase the number and viability of the cells.

As suggested by the reviewer, we used 10 hHOs raised to 81 days for library construction

immediately after dissociation. It took only about 3 hours from removing the hHOs from the incubator, dissociating them, removing the dead cells, to the final building the library. The cell viability reached 78.02%, and we finally captured the information of 6781 cells. Compared to the previous approach (4 hHOs, 94.8% cell viability, 7511 cells captured), the new approach resulted in more damage of cells and less efficient cell capture.

However, we found that the difference between the two approaches was insignificant in identifying cell types. The cell types of the two samples were similar, as shown in Supplementary Fig. 6. Notably, we did not find a cell cluster that collectively expressed oligodendrocyte markers in the new sample, as shown in Supplementary Fig. 7. The number of oligodendrocytes was smaller in the hHO, and we think that it was because the new dissociation approach would damage the cells even more, resulting in a very small amount of oligodendrocytes being damaged. Since we have been able to use immunofluorescence to identify oligodendrocytes (Supplementary Fig. 4e), the extra OPC&oligodendrocytes in the cell clusters of day-71 hHOs are unlikely to be due to the fact that we had cultured the cells in our old approach. Thus, the missing OPC&oligodendrocytes in day-81 hHOs is more likely because the high level of damage of cells probably destroyed all the oligodendrocytes in the new approach. Live cell enrichment procedures in single-cell RNA sequencing typically require large amounts of input materials, which have a high viability.

Supplementary Fig. 6 | Similar cell types in two samples of hHOs. (Left) UMAP visualization of day-70 hHOs. (Right) UMAP visualization of day-81 hHOs.

Supplementary Fig. 7 | Transcriptomic signature of day-81 hHOs. a, UMAP visualization of scRNA-seq clusters. b, The expression of OPC and oligodendrocyte markers. c, The distribution of day-81 hHO sample separated from the integrated dataset.

Supplementary Fig. 4 | HHOs. **a**, ZBTB20⁺PROX1⁺ granule neurons in the day-85 hHO. Scale bar: 100 μ m. **b**, Neuronal marker TAU and MAP2 were expressed in hHOs. Scale bar: 100 μ m. **c**, The hippocampal markers SULF2 and SEMA5A, were expressed in hHOs. Scale bar: 40 μ m. **d**, The expression of hippocampal markers ZBTB20 and PROX1, and neuron markers NEUN and MAP2 in the cells dissociated from hHOs. Scale bar: 100 μ m. **e**, The expression of astrocyte marker GFAP and oligodendrocyte progenitor cell marker OLIG2 in hHOs. Scale bar: 100 μ m.

Thus, from the point of view of being used to identify cell types, we chose to use the data from the previous sample. We placed the data of the new sample and the comparative data between the two samples in the Supplementary Information.

The revised text reads as follows on (Results section, Page 10 lines 292-296):

“[To further characterize cell identities, we performed droplet-based single-cell RNA-sequencing (scRNA-seq) to analyze the transcriptome of 7,511 cells obtained from 4 organoids at day 70, and 6781 cells from 10 organoids at day 81, using 10x Genomics Chromium. The two samples were dissociated differently, and the specific process can be found in the method section.]”

(Results section, Page 10 lines 318-322):

“[Two samples of hHOs had similar cell types (Supplementary Fig. 6), but the hHO at day 81 was missing the population of cells co-expressing oligodendrocyte markers (Supplementary Fig. 7a-b). The reason might be that the direct dissociation method used in day-81 hHOs produced more severe cellular damage, resulting in the disappearance of the oligodendrocyte cluster whose numbers were small.]”

(Methods section, Pages 22 lines 690-708):

“[(Sample preparation of day-70 hHOs) For minimizing information loss and capturing more cells from hHOs, four hHOs were dissected into smaller tissues and replated on a PLL-laminin-coated flask. Briefly, hHOs were incubated with papain enzyme at 37 °C for 20-30 minutes, and then papain enzyme was gently removed. The hHOs were dissected into fragments using tweezers on the differentiation medium 2 and transferred into PLL-laminin-coated flasks. The fragments were cultured for 3-4 days to allow for cell migration. The single-cell suspension of ~200,000 cells was obtained by dissociating cells via papain enzyme on the flasks, and the resulting single-cell suspension had a viability of approximately 94.8%.

(Sample preparation of day-81 hHOs) Ten hHOs were dissociated using the Pierce Primary Neuron Isolation Kit (Thermo Fisher). Briefly, 2-3 hHOs were placed into microcentrifuge tubes and washed with ice-cold HBSS. Then, HBSS was removed gently, and per 0.2 mL papain enzyme was added for 2-3 hHOs. We incubated them at 37 °C water bath for 20-30 minutes, with manual shaking. Then papain enzyme was gently removed and washed hHOs with HBSS. The differentiation medium 2 was added, and the hHOs were broken up by pipetting up and down 15-20 times. The pre-sorting of live cells was completed using magnetic-activated cell sorting. The single-cell suspension of ~67,800 cells was obtained, and the resulting single-cell suspension had a viability of approximately 78.02%.]”

(1.2) Also, to present the correlation between hHOs and in vivo developing hippocampus, it would be much more informative to analyse the integrated data of the hHOs scRNAseq data and in vivo developing hippocampus scRNAseq data as a reference. I believe, in this way, it would show much more informative than the heat map on Fig. 4e. Besides, it might help to identify the cluster of “Others” which contains a substantial number of the cells.

Author response: Thank you for this suggestion. We have put the eight samples of human

hippocampus from the literature and our two samples together in a pool. Then, we did a cluster analysis of all the cells and extracted the cell clusters belonging to human hippocampus *in vivo* \ day-70 hHOs \ day-81 hHOs, respectively. The results are shown in Fig. 4e and Supplementary Fig. 7c. The hHO has most of the cell types of the human hippocampus and possesses a similar distribution. Microglia and endothelial cells that do not belong to the same lineage of differentiation are concentrated in the cell population of the human hippocampus. Human *in vivo* has a richer distribution of cell populations than the hHO, and contains more types of inhibitory neurons and immature neurons. This suggests that the current differentiation strategy needs more fine-tuning to move closer to the complexity of the human hippocampus.

Fig. 4 | Transcriptomic signature of hHOs. a, UMAP visualization of 8 scRNA-seq clusters. b, Violin

plots of expression levels of markers in 7 clusters. **c**, Feature plots of hippocampal ZBTB20⁺ cells, SOX2⁺HOPX⁺ progenitors, SOX2⁺PAX6⁺ progenitors and PROX1⁺ neurons. **d**, Trajectory tree showing cell lineage relationships of all cells. Arrows show the directions of lineages. The 'others' cluster took up one termination in the upper path of neurons, which might be involved in neurodegeneration. **e**, UMAP visualization of the integrated dataset of the human hippocampus in GW22 and day-70 hHOs. (Left) All samples dataset. (Middle) The human hippocampus. (Right) Day-70 hHOs. The distribution of day-81 hHOs was present in Supplementary Fig. 7c.

Supplementary Fig. 7 | Transcriptomic signature of day-81 hHOs. **a**, UMAP visualization of scRNA-seq clusters. **b**, The expression of OPC and oligodendrocyte markers. **c**, The distribution of day-81 hHO sample separated from the integrated dataset.

This comparison also shows the absence of oligodendrocytes in day-81 hHOs, as shown in Supplementary Fig. 7c. Once again, this demonstrates the shortcomings of the new digestion for identifying cell types.

Unfortunately, the cell population 'others' was mainly concentrated in the hHO samples, and we could not identify this cell type from the new analytical method. However, inspired by the new dataset, we found that we could subdivide neural stem cells and immature neurons within the population of neural progenitor cells, so we singled out immature neurons as a cell cluster. The results have been updated in Fig. 4a and b.

The revised text reads as follows on (Results section, Page 11 lines 334-345):

“We integrated our two samples with a published scRNA-seq dataset of the developing human hippocampus³⁷ and compared cell types and their distributions (Fig. 4e). Both had a similar distribution of cell clusters, but the sample in vivo had some unique ImmN and InN, in addition to separate cell populations of endothelial cells and microglia. HHO's absence of endothelial and microglial cells, differentiated from mesodermal cells, was reasonable. This result also suggested that the in vitro differentiation of the hHO needed

to be more finely regulated and even co-cultured with other cell types to more closely resemble the in vivo hippocampus. Cell clusters of day-81 hHO were missing oligodendrocytes (Supplementary Fig. 7c), consistent with Supplementary Fig. 7a-b. The 'other' group mainly presented in hHO's samples. They might not generate in the baby's hippocampus. Combining the immunofluorescence and scRNA-seq datasets, our results indicated the successful generation of hHOs.]”

2. Advantages of mMPC being a 3D MEA system

(2.1) Throughout the manuscript, the three-dimensionality of mMPC system was emphasized. It is exciting to have developed a system where you can “sandwich” and easily fix the position of the brain organoid and can get the electrical signals of the broad range of the organoid. However, there were no data that has shown its advantages in the aspect of 3D. For example, have you observed/could you capture how electrical signal propagate from one location to the other in 3D? If yes, it would be very powerful to show to strengthen the importance of 3D.

Author response: Thank you for pointing this out. As suggested by the reviewer, we detected more neural signals from hHOs cultured for more than one hundred days by our mMPC. During the signal acquisition, we observed the neural signals from two channels that always appeared in pairs. Moreover, these paired signals always occurred one after the other, and the time interval was stable on millisecond time scales. These data suggested the existence of signal transmission between the neurons of those two channels. Our hHOs, more than one hundred days old, have grown functional synapses between neurons, and our mMPC was able to detect this synchrony. This data we have added is in Fig. 6f.

In addition, we found oscillatory network activity, i.e., coordinated bursts of spikes across electrodes, in the hippocampal cyb-organoid. These indicated that the mesh MPC could characterize the functional network on the surface of the hHO with excitatory and inhibitory neurons. We add this data in Fig. 6g.

Fig. 6 | Electrical activity recording. **a**, (Upper) Raster plots of neural spikes, and (lower) the continuous signal in a channel of the mMPC. **b**, Continuous recording for a spike. **c**, The uniform spike waveforms and the average waveform in a channel of the mMPC. **d**, Two different spikes appeared in the same channel (upper) and the sorting results (lower). **e**, The comparison of spike rates detected by the mMPC and the 2D MEA (these channels covered by the hHO). **f**, The synchronous signals between two channels. (Upper) Raster plots of neural spikes in these two channels, and (lower) continuous recording for two spikes. The time interval was less than 1 ms. **g**, The oscillatory network activity of the hHO. (Upper) the population activity histogram and (lower) the raster plot (black) identified bursts of spiking detected on 32 channels and coordinated bursts (red) of activity across channels. **h**, Raster plots before and after

Glutamine supplement. *i*, The impedance of the mMPC before and after integration with the hHO for 1 month.

However, we must admit that the current detection of hHO signaling by mesh MPC still needs to be improved. The main limitation is that the instrumentation we used (Plexon system) has a transmission capacity of only 32 channels. Even though our mesh MPC has 128 channels, acquiring signals from all channels simultaneously is impossible, thus we have to carry out the measurements in 4 separate experiments. That also limited our access to understanding the complete surface neural network of the hHO. However, the fact that 32 channels could detect neural activities has demonstrated the potential of the mMPC. In the future, we will strive to upgrade our detection equipment to utilize mMPC's ability in organoid signal detection fully. This limitation has been added in the discussion section.

The revised text reads as follows on (Results section, Page 15 lines 428-441):

“[In a series of experimental tests, we detected signals from the single hHO or hHOs-fused assemblies, and the number of active channels was listed in Supplementary Table 1. The number of active channels was mainly affected by the hHO size. For signal acquisition of an individual hHO, we detected signals in a maximum of 54 channels, whereas when we used fused assemblies of 2 hHOs, the maximum number of active channels increased to 85. Simultaneous signals always appeared in pairs in two channels. We found that the time interval of these paired signals was less than 1 ms. They always appeared one after the other (Fig. 6f). This observation suggested that neurons on the hHO formed functional synapses capable of transmitting signals and were detected by the mMPC. In addition, neural activity was detected on the hHO (Fig. 6g). As potential evidence of neural network maturation, the synchrony was proposed to reflect a balance between ExN and InN, and coordinate neuronal communication in the hHO⁴². These results highlight the potential of the hippocampal cyb-organoid applied to understand the development and disorders in the hippocampus.]”

(Discussion section, Pages 17-18 lines 506-515):

“[Compared with mature commercial systems, such as MaxWell Biosystem, HyperCAM Alpha, and Axion system, which offer an integrated platform with high-density electrodes in multiwell, real-time high-resolution scanning, and controlled environment, there is still much room for developing neural activity detection of hHOs by the mMPC. The ability of our Plexon instrument to detect only 32 channels of data at a time limits the mMPC's ability to record the complete neural activity of the hHO. In addition, obtaining long-term neuronal activity from 3D organoids requires a stable and healthy culture environment. However, there are many challenges in upgrading a system, including instrumentation, stimulation modules, temperature regulation, and gas regulation, to name a few. Adding other modules also introduces noise to the signal acquisition. These will be explored in the future.]”

(2.2) Also, when comparing mMPC to 2D MEA, one aspect that was mentioned was the cell migration on 2D MEA. According to the methods section, it is stated that the MEA plate was coated with PLL-laminin which would have led the cell migration. To compare the 2D and 3D system in the aspect of cell migration equally, I think the coating condition should be the same, either both non-coated or both coated to compare more properly.

Author response: Thank you for this suggestion. We placed the hHO in the mMPC coated with PLL-Laminin, cultured it for 20 days, and found that this condition promotes cell migration to the mMPC. The hHO also maintained its 3D structure, but we found that locally, there was cell migration to the mMPC, and the curved edges of the hHO at that location were no longer clear, and the tissue and the mMPC were intermingled. This result is added to the Supplementary Fig. 10b.

Supplementary Fig. 10 | Long-term change of hHOs in MEA and noninvasive assessment. a, (Left) The hHO on a commercial 2D MEA after culturing for 20 days. Scale bar: 500 μ m. (Right) zoom-in view of the red box in the left figure showing the cell migration on the 2D MEA. Scale bar: 100 μ m. **b, (Left)**

bright-field images of hHOs on the normal mMPC after cocultured for 30 days and on the PLL-Laminin-coated mMPC for 20 days. Scale bar: 100 μ m. c, The mMPC electrodes with and without PEDOT coating after culturing the hHO for 2 weeks. The red arrows point to the electrodes. Scale bar: 50 μ m. d, Non-invasive assessment. The morphology of the same hHO in dish (Left), in the mMPC for 4 days (Middle), after fixation and immunostaining (Right). Scale bar: 100 μ m in the staining image and 2 mm in others. e, The cross-sectional view showed the attachment boundary between the mMPC and the hHO. Scale bar: 100 μ m. (Right) The estimated areas of the hHO and that covered by the mMPC.

The revised text reads as follows on (Results section, Page 13 lines 376-384):

“[The mMPC offered distinct advantages over the 2D MEA. Specifically, the two-layer mMPCs supported hHOs directly, while fixing hHOs on the 2D MEA was challenging. Additionally, we observed that cells from the hHO gradually migrated in the 2D MEA, leading to a contribution to the signal from these migrated cells rather than the neural network of hHOs (Supplementary Fig. 10a). In contrast, without coating poly-L-lysine (PLL)-laminin, the mMPC allowed for the clear boundary of the hHO to be maintained for 30 days without no significant migration of cells away from the hHO (Supplementary Fig. 10b). Cell migration also occurred in the PLL-laminin-coated mMPC, leading to fusion locally between the hHO and the mesh structure (Supplementary Fig. 10b).]”

3. In Fig. 6g, 1 month of usage means actual 1 month of culturing the organoid on mMPC?

Author response: Thank you for the question. Yes, we have placed hHOs raised over 100 days in the mMPC for up to 30 days. With enough culture medium and daily medium changes, the hHO can maintain its morphology. However, we do recognize that this way is detrimental to the growth of hHO. Thus, we also emphasized this limitation in the discussion. In the future, if we offer the perfused culture medium and improve the diffusion in this system, we should be able to keep the hHO healthier.

The revised text reads as follows on (Discussion section, Page 17 lines 489-496):

“[There are still some limitations of the current cyb-organoid. The free-floating status of all brain organoids pose great challenges for designing electronics and integrating electronics with organoids. Even if the hHO can maintain its morphology in the mMPC for a long time, this fixation of hHO in the same position must be detrimental to its growth compared to the culture condition in a bioreactor. The medium diffusion in the cyb-organoid system is not as good as the flow medium, even though we keep changing the medium every day. This problem may be improved if a flowing medium system can be provided for the cyb-organoid.]”

4. In Fig. 3c, it would be more informative if you could specify the time point of the development of the organoid as a continuation of Fig. 3a.

Author response: Thank you for this suggestion. We agree it would have been informative to explore this aspect. We have added the dates on the developmental axis of the organoid in Fig.3c.

Fig. 3 | Generation of hHOs. **a**, The overall strategy to generate hHOs. **b**, The typical morphology of hHOs at different ages. Scale bar: 100 μm . **c**, The significant markers in the developmental process of the hHO. **d**, Staining images of day-30 hHOs. Upper: Larger-scale image of the whole hHO. Scale bar: 100 μm . Lower: Zoom-in greyscale view. Scale bar: 40 μm . **e**, The hippocampal PAX6⁺ and HOPX⁺ progenitors in day-60 hHOs. Scale bar: (left) 100 μm and (right, zoom-in greyscale view) 40 μm . **f**, The hippocampal PROX1⁺ cells in day-60 hHO and day-85 hHO. Scale bar: 150 μm . **g**, qPCR for genes expressed in day-117 hHOs, day-71 hHOs versus day-46 hHOs (* $P < 0.05$; ** $P < 0.01$, $n = 3$, unpaired t -test). **h**, qPCR for genes expressed in day-45 hHOs versus day-45 DMT organoids (* $P < 0.05$, $n = 3$, unpaired t -test).

Reviewers' comments:

Reviewer #1 (Remarks to the Author):

The authors responded well to most of the reviewer's requests. Through reading the revised manuscript, I noticed some points that the authors would better consider as follows;

Comments

1. The authors have misunderstanding of a previous report which affect their story. Regarding response 2.2, the authors wrote;

Ref. 26 demonstrated that hippocampal cells could be differentiated in the DMT organoid, but these cells were obtained after the researcher cut off the choroid plexus (ChP) from the DMT organoid, dissociated and cultured the remaining tissue alone. We were puzzled as to why the hHO could not be obtained by directly culturing the DMT organoid for a long period, as in this condition, the ChP was able to release growth factors continuously to regulate hHO development as occurs in vivo development. Thus, we conjectured that the presence of ChP may have hindered hHO development, which may be why it was cut off in ref. 26.

This is misunderstanding of the mentioned paper. Ref 26 did not cut off ChP, rather they maintain ChP, hem, and medial pallium. They just cut half of main body of organoid that did not have ChP side for long-term culture.

The authors made story of mutual antagonism of ChP and hippocampal differentiation in DMT organoid, but in vivo situation, both ChP and hippocampal region develop mutually. The organoid is derived from developmental biology and the expansion of developmental knowledge into stem cell field opened up organoid research field. When researchers create hippocampal region with less ChP region by adding wnt and shh signaling more, then the appropriate interpretation is "modulation of DMT induction method into more hippocampal side resulting the generation of more hippocampal region and less ChP region".

Thus, the authors can rewrite related parts; line111-112, line254-261,

2. There are some strong descriptions.

Line 290-291: The authors did not show the activity of astrocyte and oligodendrocyte, and the sentence "These cells play critical roles~~~" is inappropriate.

Line 297-300: The range of relative mRNA level of wnt3a is one to twice, and range of error bar of day71 is large. Level at day 46 and 117 looks almost same. Thus, it is too strong that authors claimed that wnt3a was synthesized in greater abundance during growth period and was no longer so required during maturation phase.

3. Please make it clear where the high magnification picture correspond to lower magnification view at Figure3 d and e.

4. Line 353-358: The authors mentioned possible association with AD and PD, but is it relate to corresponding gene mutation, or just related to general genes that is disrupted in AD/PD? Because both diseases are late onset, there is not enough data to suggest the relationship with these disorders in current data set. The authors can rewrite the corresponding part.

5. The authors used a term “brain organoid”, but currently this term may be considered as inappropriate, because the researchers have not made whole “brain” like tissue. The recommended term is neural organoid or nervous system organoid as ref below.

Paşca SP, Arlotta P, Bateup HS, Camp JG, Cappello S, Gage FH, Knoblich JA, et al (2022) A nomenclature consensus for nervous system organoids and assembloids. *Nature* 609(7929):907-910.

6. Please add information of antibodies of SULF1 and SEMA3a.

Reviewer #2 (Remarks to the Author):

The reviewer recognizes the extensive efforts the authors have made to improve the manuscript. However, my major concerns are not addressed in this revision.

1- My major concern of the organoid being deformed by the interaction with the mesh device still holds. The authors made extensive explanation of the detailed operations during the insertion of organoid into mMPCs that are intended to minimize the pressure, and I agree that these optimizations in experimental operations make sense. However, direct evidence to confirm that there is no deformation or squeezing in the organoid is still missing, which is very critical for evaluating the scientific achievement of the paper. Although the authors aim to minimize the deformation of the organoid, the images of the organoid shown in the devices e.g., Fig. 5a, show a compressed shape rather than a spherical one. My concern is exacerbated by the control organoid shown in Fig. 5a, which also has an unconventional non-spherical shape that is relatively rare in organoids grown in suspension systems. To directly show and confirm the deformation effect of the mMPC device, the authors must clearly provide statistical results and recorded videos of organoid insertion and its effect on the geometry of the organoid before and after insertion into the mesh devices.

2- The authors use a TPU ball to pre-define the geometry of the space between top and bottom mMPCs to minimize the pressure on the organoid. However, this arises additional concerns to the conformal wrapping of the device onto the organoid surface, which is essential for recording high-quality signal. As shown in Supplementary Fig. 9b, the space between the two layers of mMPCs after the TPU ball seems closer to a cylinder shape rather than spherical, which is contradictory to the idea of maximizing the contact between the mesh and round organoids surface. This geometry makes me think that enhancing such contact may only be achievable through further squeezing of the organoid, which again runs into the deformation issue mentioned above. Similarly, in the authors’ response to previous review comment

3, they mentioned fewer channels can work when a smaller size organoid is used. This also supports the fact that without unavoidable squeezing of the organoid, a good device-organoid contacting is not guaranteed.

3- The authors provide a comparison between neural signals detected with and without the top mMPC to demonstrate minimal effect from the sandwich strategy on organoid activity. However, the methodology tends to lack a proper control-- they should measure neural signal from the bottom mMPC before sandwiching, as the control experiment, rather than after sandwiching and uncovering. Besides, the recoding presented in Supplementary Fig. 12a seems different between measured with sandwiched and bottom mMPC only, where the non-sandwiched organoid shows slightly higher spike rate.

4- Another major concern that still holds is the high number of working channels claimed in the paper. The reviewer believes that the major goal of this paper is to obtain a conformal contact between the organoid and a soft mesh device, which is crucial for electro-recording of organoids with varying sizes and shapes. The authors show results from a number of working channels in assemblies and individual organoids with a larger number of working channels with a larger size of organoids/assemblies. This result makes sense but is contrary to the idea of conformal contacting. Besides, if possible, the paper may be much improved if the authors can get one set of simultaneous recording data with all the 128 channels connected.

5- It seems to me that the authors mistakenly introduce and calculate “flexural strength”, which is the maximum stress that a material can withstand while being bent or flexed before it yields, as “bending stiffness”, which is the ability of a structure to resist deformation when subjected to bending moments. The unit of the former, which authors calculate, is Pa whereas the unit of the latter is N.m². As a result, what we learn from the conducted tests is the high stretchability of the device and not the degree of its bendability, which was the point of my previous comment.

Reviewer #3 (Remarks to the Author):

The authors clarified the doubts identified in the first version of the article and significantly improved the final version of the manuscript.

Reviewer #4 (Remarks to the Author):

Most of the points that were raised were addressed or corrected accordingly in the manuscript. However, there is still a concern about choosing the correct scRNAseq dataset to support the study. The main idea of this study is the development of a flexible multielectrode array that is conformal to 3D hHOs. But the dataset shown in the main figure (Figure 4) is from day-70 hHOs that were dissected into

fragments and cultured attached to PLL-laminin coated flask in 2D for 3-4 days before being harvested for scRNAseq. This mixture of 3D and 2D cultured cells do not represent the 3D hHOs that were used for recording the electrical activity using mMPCs. The transcriptomic analysis of day-81 hHOs as 3D as it is before scRNAseq would be more representative of the cells that were measured for their electrical activity. In the rebuttal, the authors also commented that the difference between day-70 and day-81 hHOs in terms of cell type was insignificant. Then I wonder why the authors still chose day-70 dataset despite of its different conditions to 3D hHOs used for electrical activity measurement.

Reviewer 1

The authors responded well to most of the reviewer's requests. Through reading the revised manuscript, I noticed some points that the authors would better consider as follows;

Comments

1. The authors have misunderstanding of a previous report which affect their story. Regarding response 2.2, the authors wrote;

Ref. 26 demonstrated that hippocampal cells could be differentiated in the DMT organoid, but these cells were obtained after the researcher cut off the choroid plexus (ChP) from the DMT organoid, dissociated and cultured the remaining tissue alone. We were puzzled as to why the hHO could not be obtained by directly culturing the DMT organoid for a long period, as in this condition, the ChP was able to release growth factors continuously to regulate hHO development as occurs in vivo development. Thus, we conjectured that the presence of ChP may have hindered hHO development, which may be why it was cut off in ref. 26.

This is misunderstanding of the mentioned paper. Ref 26 did not cut off ChP, rather they maintain ChP, hem, and medial pallium. They just cut half of main body of organoid that did not have ChP side for long-term culture.

The authors made story of mutual antagonism of ChP and hippocampal differentiation in DMT organoid, but in vivo situation, both ChP and hippocampal region develop mutually. The organoid is derived from developmental biology and the expansion of developmental knowledge into stem cell field opened up organoid research field. When researchers create hippocampal region with less ChP region by adding wnt and shh signaling more, then the appropriate interpretation is "modulation of DMT induction method into more hippocampal side resulting the generation of more hippocampal region and less ChP region".

Thus, the authors can rewrite related parts; line111-112, line254-261.

Author response: Thank you for pointing this out and we think it is an excellent suggestion. As suggested by the reviewer, we rephrased the related descriptions.

The revised text reads as follows on (Introduction section, Page 3 lines 110-113):

"[Furthermore, we modulated the DMT induction method into a more hippocampal side, generating a more hippocampal region and a smaller ChP region in the organoid. Under the Wnt3a and SHH activator induction, we generated hHOs with many PAX6⁺, HOPX⁺ hippocampal progenitors, and PROX1⁺ granule neurons.]"

(Results section, Pages 7-8 lines 236-244):

"[Researchers have proved that some hippocampal neurons, accompanied by the ChP and CH regions, could be derived from DMT organoids (Supplementary Fig. 3b)²⁶. However, in an in vitro environment, the expansion of ChP tissues in DMT organoid tissues might affect the development of hippocampal tissue. Specifically, it has been observed that the TTR⁺

ChP tissue became more abundant while the neural tissue became smaller after long-term culture of the DMT organoids (Supplementary Fig. 3c-3e). This result might be attributed to the enriched expression of BMPs in the ChP, which promoted its proliferation and inhibited differentiation of neural progenitors³²⁻³⁴. To generate hHOs from DMT tissue, alternative growth factors were considered to inhibit the ChP expansion.]”

2. There are some strong descriptions.

Line 290-291: The authors did not show the activity of astrocyte and oligodendrocyte, and the sentence “These cells play critical roles~~~”is inappropriate.

Author response: Thank you for pointing this out. We have rephrased the relevant sentence and cited some literature related to glial cells and oligodendrocyte function.

The revised text reads as follows on (Results section, Page 8 lines 268-271):

*“[Additionally, GFAP⁺ astrocytes and OLIG2⁺ oligodendrocyte progenitor cells were present (Supplementary Fig. 4e). **These cells have been reported to play roles in maintaining the growth environment of neurons and support the signal transmission between nerves⁴¹⁻⁴³.**]”*

Line 297-300: The range of relative mRNA level of wnt3a is one to twice, and range of error bar of day71 is large. Level at day 46 and 117 looks almost same. Thus, it is too strong that authors claimed that wnt3a was synthesized in greater abundance during growth period and was no longer so required during maturation phase.

Author response: Thank you for pointing this out. We have rephrased the sentence as follows on (Results section, Page 8 lines 273-279).

*“[Compared with day-45 DMT organoids, we found a significant increase in PROX1 mRNA expression and a significant decrease in TTR mRNA expression in the day-45 hHO (Fig. 3h). Along with the hHO growing, the ZBTB20 and PROX1 mRNA expression gradually increased (Fig. 3g). Immunostaining also showed increased PROX1 expression in the more mature hHO (Fig. 3f). **Wnt3a mRNA expression did not, however, consistently increase (Fig. 3g).**]”*

3. Please make it clear where the high magnification picture correspond to lower magnification view at Figure3 d and e.

Author response: Thank you for pointing this out. We have added the magnified locations labeled with white boxes in Fig. 3.

Fig. 3 | Generation of hHOs. a, The overall strategy to generate hHOs. **b**, The typical morphology of hHOs at different ages. Scale bar: 100 μ m. **c**, The significant markers in the developmental process of the hHO. **d**, Staining images of day-30 hHOs. Upper: Lager-scale image of the whole hHO. Scale bar:

100 μm . Lower: Zoom-in greyscale view of the white box in the upper figures. Scale bar: 40 μm . **e**, The hippocampal PAX6⁺ and HOPX⁺ progenitors in day-60 hHOs. Scale bar: (Left) 100 μm and (Right, zoom-in greyscale view of the white box in the left figures) 40 μm . **f**, The hippocampal PROX1⁺ cells in day-60 hHO and day-85 hHO. Scale bar: 150 μm . **g**, qPCR for genes expressed in day-117 hHOs, day-71 hHOs versus day-46 hHOs (* $P < 0.05$; ** $P < 0.01$, $n = 3$, unpaired t -test). **h**, qPCR for genes expressed in day-45 hHOs versus day-45 DMT organoids (* $P < 0.05$, $n = 3$, unpaired t -test).

4. Line 353-358: The authors mentioned possible association with AD and PD, but is it relate to corresponding gene mutation, or just related to general genes that is disrupted in AD/PD? Because both diseases are late onset, there is not enough data to suggest the relationship with these disorders in current data set. The authors can rewrite the corresponding part.

Author response: Thank you for pointing this out. We checked the highly expressed genes in the 'other' cluster, and some were related to the general genes in the AD/PD pathways. But more experiments will be required to confirm causal relationship. Thanks to this suggestion, we think our previous description was too strong. We have rephrased the relevant sentence.

The revised text reads as follows on (Results section, Pages 10-11 lines 326-343):

“We integrated our two samples with a published scRNA-seq dataset of the developing human hippocampus³⁷ and compared cell types and their distributions (Fig. 4d and Supplementary Fig. 8c). Both had a similar distribution of cell clusters, but the sample in vivo had some unique ImmN and InN, in addition to separate cell populations of endothelial cells and microglia. The absence of endothelial and microglial cells, differentiated from mesodermal cells, was reasonable. This result also suggested that the in vitro differentiation of the hHO needed to be more finely regulated and even co-cultured with other cell types to more closely resemble the in vivo hippocampus. Cell clusters of day-81 hHOs were missing oligodendrocytes, consistent with Fig. 4a-b. We identified a subgroup of cells as 'others' that we could not define based on their differentially expressed genes. The 'other' group mainly presented in hHOs samples, close to the neuron clusters. The single-cell trajectory analysis also indicated that the identified cells in the 'others' subgroup were located at a separate termination of neuron branches that began from the NPCs (Supplementary Fig. 8d). They did not occur in the fetal hippocampus. The imperfect in vitro culture might lead to other neuronal cells in the hHO that did not belong to the hippocampus or were associated with neurological disorders¹¹. But more experiments will be required to confirm causal relationship. Combining the immunofluorescence and scRNA-seq datasets, our results indicated the successful generation of hHOs.”

5. The authors used a term “brain organoid”, but currently this term may be considered as inappropriate, because the researchers have not made whole “brain” like tissue. The recommended term is neural organoid or nervous system organoid as ref below.

Paşca SP, Arlotta P, Bateup HS, Camp JG, Cappello S, Gage FH, Knoblich JA, et al (2022) A nomenclature consensus for nervous system organoids and assembloids.

Nature 609 (7929):907-910.

Author response: Thank you for this suggestion. As described in the ref., 'Neural organoids should be named as broadly as possible unless strong evidence for recapitulating a specific domain is provided (for example, spinal, hindbrain or arcuate nucleus).' we unified all occurrences of 'brain organoids' in the manuscript and used 'neural organoids' instead.

6. Please add information of antibodies of SULF2 and SEMA3a.

Author response: Thank you for pointing this out. We have added the information in Supplementary Information.

Antibodies

LEF1	Abcam	ab137872
SOX2	Abcam	Ab92494
PAX6	Proteintech	12323-1-AP
PAX6	Abcam	ab78545
ZBTB20	Proteintech	23987-1-AP
Olig2	Abcam	ab109186
FOXP1	Abcam	Ab196868
FOXP1	Genetex	GTX134018
HOPX	Proteintech	11419-1-AP
HOPX	Thermo Fisher	PA590538
Prox1	R&D Systems	AF2727
Prox1	Abcam	ab199359
MAP2	Invitrogen	MA512826
Nestin	Abcam	ab6320
NeuN	Genetex	GTX132974-S
GFAP	Genetex	GTX108711
β -Tubulin 3	Abcam	Ab27074
TAU	Abcam	ab92676
TTR	Proteintech	11891-1-AP
SULF2	Abcam	ab232835
SEMA5A	EpigenTek	A64642-020
Goat-Rabbit 488	Abcam	ab150077
Goat-Rabbit 647	Abcam	ab150083
Goat-Mouse 488	Genetex	GTX213111-04
Goat-Mouse 647	Abcam	ab150115
Goat-Rabbit 594	Abcam	ab150080
Phalloidin 555	Abcam	ab176756

Reviewer 2:

The reviewer recognizes the extensive efforts the authors have made to improve the

manuscript. However, my major concerns are not addressed in this revision.

1- My major concern of the organoid being deformed by the interaction with the mesh device still holds. The authors made extensive explanation of the detailed operations during the insertion of organoid into mMPCs that are intended to minimize the pressure, and I agree that these optimizations in experimental operations make sense. However, direct evidence to confirm that there is no deformation or squeezing in the organoid is still missing, which is very critical for evaluating the scientific achievement of the paper. Although the authors aim to minimize the deformation of the organoid, the images of the organoid shown in the devices e.g., Fig. 5a, show a compressed shape rather than a spherical one. My concern is exacerbated by the control organoid shown in Fig. 5a, which also has an unconventional non-spherical shape that is relatively rare in organoids grown in suspension systems. To directly show and confirm the deformation effect of the mMPC device, the authors must clearly provide statistical results and recorded videos of organoid insertion and its effect on the geometry of the organoid before and after insertion into the mesh devices.

Author response: Thank you for pointing this out. Not all human hippocampal organoids (hHOs) were perfectly spherical after long-term culture. In order to reflect the conformal attachment of mMPC on the surface of hHO, we purposely used this hHO with some undulations in the side view to show in Fig. 5a, but in fact, it is also round when viewed from the top, similar as in Fig. 5b.

This squeezing did not seriously affect the morphology of the hHO, nor did it cause significant damage to the tissues. We thank the reviewer for pointing out the lack of direct evidence we have shown. Following the reviewer's suggestion, we recorded the complete insertion of the hHO into the two-layer mMPC in Video 3 and the change of hHOs before and after the top mMPC attached to them in Video 4. After the hHO was inserted into the two-layer mMPC, the hHO within mMPC was intentionally shaken to indicate that the hHO was still free but confined in this space. When a part of the culture medium was removed, the top mMPC gradually attached to the surface of hHO, and the hHO was fixed between the two layers. At this time, it was impossible to shake hHO, indicating that the two layers of mMPC had been adhered to the surface of hHO. The morphology change of the side and bottom edges was presented in Supplementary Fig. 10a-c, showing only about 4% deformation. The micro-CT scanning image also showed the similar height of the hHO in the free-floating state and under the sandwiching of mMPCs, with a difference of no more than 1% (Supplementary Fig. 10d).

Supplementary Fig. 10 | The deformation of hHOs sandwiched between two mMPCs. a, The side view of hHOs before and after sandwiched between two mMPCs. Scale bar: 1 mm. **b**, The bottom view of hHO and hHO assembly before and after sandwiched between two mMPCs. Scale bar: 100 μm . **c**, The side view of Fluo-8-loaded hHOs before and after sandwiched between two mMPCs. Scale bar: 500 μm . The complete process of Figure a-c was recorded in Supplementary Video 4. When the hHO was inserted into the space between the top and bottom mMPCs, the hHO was still free-floating but confined in this space. We could shake it. After slowly removing the medium, the top mMPC was attached to the hHO surface. The hHO was fixed at that location and we could not shake it. **d**, Micro-CT Images of hHOs sandwiched between two mMPCs. (Left) The hHO tissue could not be distinguished from the medium due to the lack of vascular structures. The mMPC could be captured due to its higher density. (Middle) The cross-section in the red line of the left figure. (Right) The suspended hHO at the bottom mMPC. Scale bar: 1 mm.

The mMPC naturally formed a structure with a slightly concave center after being peeled from the PDMS slabs (Supplementary Fig. 9a). After cutting off the redundant TPU/PU materials around the mMPC, it also naturally presented a concave structure. When assembling the mMPC, we fixed the distance between the two sides so that the mMPC appeared in a "bowl" shape, which is favorable for wrapping the hHO. We are very sorry that we retained some of the original TPU material in the previous presentation, similar as the Supplementary Fig. 9b, which is not conducive to showing this structure and to showing the side view of the sandwich structure. In this recorded video, we removed all the redundant TPU/PU, and the side shot showed this concave structure, as shown in Video 3 and Supplementary Fig. 9a.

Supplementary Fig. 9 | Assembling of 128-channel mMPC and biocompatibility of mMPC. a, The integrated device coupling the mMPC with culture dishes. (Left): The mesh naturally formed a structure with a slightly concave center after being peeled from the PDMS slabs. The enlarged image showed that

after cutting off the redundant TPU/PU materials around the mMPC, the center also naturally presented a concave structure. (Upper right) When assembling the bottom mMPC, we fixed the distance between the two sides so that the bottom mMPC appeared in a "bowl" shape. (Lower right): Assembly of double-layer mMPCs with the 35-mm dish. The TPU ball in the center was to reserve the space the hHO may need. **b**, (Left) The two-layer mMPCs with culture medium. After remove the TPU ball, there existed some space between these two layers. (Middle and right) The top and the side view after placing one hHO assembly into the two-layer mMPC. Scale bar: 5 mm. When the culture solution was just submerged over the hHO assembly, the top mMPC covered the surface of the hHO assembly without significant squeezing. **c**, Biocompatibility. The mouse hippocampal cells grew in the mMPC for 30 days. Scale bar: 100 μ m.

The revised text reads as follows on (Results section, Page 12 lines 355-372):

"[To enable the integration of the mMPC with the hHO and facilitate the connection with the Plexon instrument, we assembled the mMPC and culture dishes. To prevent the two-layer meshes from severely squeezing on the soft hHO, the mMPC was designed as a 'bowl' to hold the hHO when we assembled this device (Supplementary Fig. 9a). We placed a TPU ball of \sim 3 mm when assembling the device to reserve the space the hHO may need (Supplementary Fig. 9a-b). After sterilization, the cultured medium was added to the dish, and the TPU ball was removed from the side; the hHO was inserted into the space between the top and bottom mMPCs. The oblique view and the side view showed that the top mMPC varies with the undulation of the hHO surface without causing significant compression on the hHO and the hHO assembly (Fig. 5a and Supplementary Fig. 9b), indicating the conformal attachment of the mMPC to the surface of the hHO. More details about the process can be found in Video 3. Comparisons before and after attachment of the top mMPC showed that the top mMPC could slightly squeeze the hHO, resulting in an area enlargement of less than 5% in the bottom view and a reduction of less than 5% in the side view (Video 4 and Supplementary Fig. 10a-c). Micro-computed tomography (micro-CT) images showed the hHO changes in height within 1% compared to its suspended state (Supplementary Fig. 10d and Video 5). This squeeze did not severely affect the morphology of the hHO or cause damage to the hHO.]"

2- The authors use a TPU ball to pre-define the geometry of the space between top and bottom mMPCs to minimize the pressure on the organoid. However, this arises additional concerns to the conformal wrapping of the device onto the organoid surface, which is essential for recording high-quality signal. As shown in Supplementary Fig. 9b, the space between the two layers of mMPCs after the TPU ball seems closer to a cylinder shape rather than spherical, which is contradictory to the idea of maximizing the contact between the mesh and round organoids surface. This geometry makes me think that enhancing such contact may only be achievable through further squeezing of the organoid, which again runs into the deformation issue mentioned above. Similarly, in the authors' response to previous review comment 3, they mentioned fewer channels can work when a smaller size organoid is used. This also supports the fact that without unavoidable squeezing of the organoid, a good device-organoid contacting is not guaranteed.

Author response: Thank you for pointing this out. After removing the TPU ball, the mMPC

floated in the culture medium, and the top mMPC appeared to be cylindrical, as shown in Supplementary Fig.9b. The width of the mMPC is 4.75 mm, while the diameter of our hHO was generally ~2 mm. Therefore, for a single hHO, after the center part of the top mMPC is attached to the hHO, its edge can be close to the bottom mMPC, which was not cylindrical at this time, as shown in Fig. 5a. The micro-CT scanning also showed the complete space after the top mMPC attached to the hHO in Supplementary Video 5.

To better show whether the top mMPC squeezes the larger hHO, we inserted an hippocampal assembly into the mMPCs in Supplementary Fig. 9b. Therefore, a long ellipsoidal assembly can be seen from the middle figure. Thus, the center portion of the mMPC attached to the hHO surface, but the edge was propped up by the large size of the assembly, forming the cylindrical shape in the right figure. This result indicated that the mMPC did not squeeze the larger neural organoid to severe deformation. If the neural organoid was large, more electrodes in the mMPC center could access the neural organoid, leading to more active channels rather than being squeezed. However, the two layers did not completely wrap the hHO surface 100%. There would always be a distance between the two layers, regardless of the size of the hHOs we used. We have added this limitation in the discussion section.

Supplementary Fig. 9 | Assembling of 128-channel mMPC and biocompatibility of mMPC. **a**, The integrated device coupling the mMPC with culture dishes. (Left): The mesh naturally formed a structure with a slightly concave center after being peeled from the PDMS slabs. The enlarged image showed that after cutting off the redundant TPU/PU materials around the mMPC, the center also naturally presented a concave structure. (Upper right) When assembling the bottom mMPC, we fixed the distance between the two sides so that the bottom mMPC appeared in a "bowl" shape. (Lower right): Assembly of double-layer mMPCs with the 35-mm dish. The TPU ball in the center was to reserve the space the hHO may need. **b**, (Left) The two-layer mMPCs with culture medium. After remove the TPU ball, there existed some space between these two layers. (Middle and right) The top and the side view after placing one hHO assembly into the two-layer mMPC. Scale bar: 5 mm. When the culture solution was just submerged over the hHO assembly, the top mMPC covered the surface of the hHO assembly without significant squeezing. **c**, Biocompatibility. The mouse hippocampal cells grew in the mMPC for 30 days. Scale bar: 100 μ m.

The revised text reads as follows on (Discussion section, Page 17 lines 504-514):

“[Although the sandwich structure of our cyb-organoid greatly increased the channel number, this configuration might result in a slight pressure on the hHO. In order to make the top mMPC attach to the hHO, we would reduce the culture medium to only submerge a little past the surface of the hHO. The comparison of the signal recordings with and without the top mMPC illustrated that this squeezing improved the contact between the hHO and the electrodes. At the same time, however, the signals detected in this case were not exactly the spontaneous activity of the hHO in a free state but rather the neural activity under a slight squeeze. This limitation may cause some bias in our understanding of the neural activity. In addition, the mMPC still needed to cover the surface of neural organoids completely. There was a certain distance between the two layers, which still puts us far from understanding the complete neural network on the surface of the hHO.]”

3- The authors provide a comparison between neural signals detected with and without the top mMPC to demonstrate minimal effect from the sandwich strategy on organoid activity. However, the methodology tends to lack a proper control– they should measure neural signal from the bottom mMPC before sandwiching, as the control experiment, rather than after sandwiching and uncovering. Besides, the recoding presented in Supplementary Fig. 12a seems different between measured with sandwiched and bottom mMPC only, where the non-sandwiched organoid shows slightly higher spike rate.

Author response: Thank you for this suggestion. We added the comparison of the neural signals before and after sandwiching in Supplementary Fig. 14b. We found that only a few active channels were in the absence of the top mMPC. The reason should be that the gravity of hHO alone deposited it to the center of the bottom mMPC, which did not guarantee the contact between hHO and electrodes. When we put the top mMPC back, more channels recorded electrical signals. This result suggested that squeezing from the top mMPC improved the contact of the mMPC electrodes with the hHO. We illustrated this change in both results section and discussion section. We also rephrased the sentence with more precise descriptions for the difference between sandwiching and cutting the top mMPC.

Supplementary Fig. 14 | a. The comparison of spike rates recorded by the bottom mMPC before and after cutting the top mMPC. Scale bar: 5 mm. **b,** The comparison of spike rates recorded by the bottom mMPC before and after attachment of the top mMPC. Scale bar: 5 mm.

The revised text reads as follows on (Results section, Pages 14-15 lines 454-467):

“[Concerned that pressure from the top mMPC could affect the electrical activity of the hHO, we compared the neural signals **before and after cutting the top mMPC**. A comparison of the average spike rates of multiple active channels over two minutes showed no significant difference, **only a few channels showing slightly higher spike rate (Supplementary Fig. 14a)**. However, as in the absence of top mMPC, we placed the hHO at the bottom mMPC directly and found that only a few electrodes were able to detect the signal when the hHO was sunk onto the bottom mMPC by its gravity alone (Supplementary Fig. 14b). When the top mMPC was used simultaneously, more electrodes at bottom mMPC detected the signal. Pressure from the top mMPC can improve the contact between the hHO and the mMPC, allowing more channels to work. Following one month of integration, we measured the electrode impedance after removing the hHO from the mMPC. At 40 kHz, impedance increased from 95.83 kΩ to 196.03 kΩ compared with the initial stage (Supplementary Fig. 13b). This increase in impedance might have resulted from the degradation of some Galn alloy from the electrodes, leading to a reduction in the conductivity.]”

(Discussion section, Page 17 lines 504-514):

“[Although the sandwich structure of our cyb-organoid greatly increased the channel

number, this configuration might result in a slight pressure on the hHO. In order to make the top mMPC attach to the hHO, we would reduce the culture medium to only submerge a little past the surface of the hHO. The comparison of the signal recordings with and without the top mMPC illustrated that this squeezing improved the contact between the hHO and the electrodes. At the same time, however, it was obvious that the signals detected in this case were not exactly the spontaneous activity of the hHO in a free state but rather the neural activity under a slight squeeze. This limitation may cause some bias in our understanding of the neural activity. In addition, the mMPC still needed to cover the surface of neural organoids completely. There was a certain distance between the two layers, which still puts us far from understanding the complete neural network on the surface of the hHO.]”

4- Another major concern that still holds is the high number of working channels claimed in the paper. The reviewer believes that the major goal of this paper is to obtain a conformal contact between the organoid and a soft mesh device, which is crucial for electro-recording of organoids with varying sizes and shapes. The authors show results from a number of working channels in assemblies and individual organoids with a larger number of working channels with a larger size of organoids/assemblies. This result makes sense but is contrary to the idea of conformal contacting. Besides, if possible, the paper may be much improved if the authors can get one set of simultaneous recording data with all the 128 channels connected.

Author response: Thank you for pointing this out. We thought more electrodes worked when the large hHO was put into the mMPC because more electrodes could reach the hHO. This did not contradict the conformality of the mMPC. Nevertheless, unfortunately, we have to recognize that mMPC, although it was soft, did not completely wrap around the surface of the hHO. Regardless of the size of the hHO, there was inevitably a certain gap between the two layers of mMPC. This limitation has been added to the discussion section.

The revised text reads as follows on (Discussion section, Page 17 lines 504-514):

“[Although the sandwich structure of our cyb-organoid greatly increased the channel number, this configuration might result in a slight pressure on the hHO. In order to make the top mMPC attach to the hHO, we would reduce the culture medium to only submerge a little past the surface of the hHO. The comparison of the signal recordings with and without the top mMPC illustrated that this squeezing improved the contact between the hHO and the electrodes. At the same time, however, the signals detected in this case were not exactly the spontaneous activity of the hHO in a free state but rather the neural activity under a slight squeeze. This limitation may cause some bias in our understanding of the neural activity. In addition, the mMPC still needed to cover the surface of neural organoids completely. There was a certain distance between the two layers, which still puts us far from understanding the complete neural network on the surface of the hHO.]”

In addition, the simultaneous detection of 128 channels of signals is still very difficult for us. We believe this research is important in providing a 3D multi-channel device that is

more suitable for neural organoid signal detection. This novel fabrication process, with great plasticity, enables other researchers that are in a position to do so to prepare electrodes suitable for their systems. Thanks to the reviewer's suggestion, we integrated the four 32-channel data detected separately for visualization, as shown in Fig. 6i and Video 6. This demonstration better showed the potential of mMPC in detecting signals from 3D models.

Fig. 6 | Electrical activity recording. *a*, (Upper) Raster plots of neural spikes, and (lower) the continuous signal in a channel of the mMPC. *b*, Continuous recording for a spike. *c*, The uniform spike waveforms and the average waveform in a channel of the mMPC. *d*, Two different spikes appeared in the same

channel (upper) and the sorting results (lower). **e**, The comparison of spike rates detected by the mMPC and the 2D MEA (these channels covered by the hHO). **f**, The synchronous signals between two channels. (Upper) Raster plots of neural spikes in these two channels, and (lower) continuous recording for two spikes. The time interval was less than 1 ms. **g**, The oscillatory network activity of the hHO. (Upper) the population activity histogram and (lower) the raster plot (black) identified bursts of spiking detected on 32 channels and coordinated bursts (red) of activity across channels. **h**, Raster plots before and after Glutamine supplement. **i**, 3D visualization plot showing average spike numbers throughout 128 channels.

The revised text reads as follows on (Results section, Page 14 lines 432-452):

*“[In a series of experimental tests, we detected signals from the single hHO or hHO-fused assemblies, and the number of active channels was listed in Supplementary Table 1. The number of active channels was mainly affected by the hHO size. For signal acquisition of an individual hHO, we detected signals in a maximum of 54 channels, whereas when we used fused assemblies of 2 hHOs, the maximum number of active channels increased to 85. Simultaneous signals always appeared in pairs in two channels. We found that the time interval of these paired signals was less than 1 ms. They always appeared one after the other (Fig. 6f). This observation suggested that neurons on the hHO formed functional synapses capable of transmitting signals and were detected by the mMPC. In addition, neural activity was detected on the hHO (Fig. 6g). As potential evidence of neural network maturation, the synchrony was proposed to reflect a balance between ExN and InN, and coordinate neuronal communication in the hHO⁴⁵. We further confirmed the capability of mMPCs by examining the signal of hHOs response to 10 mM glutamine, a precursor of the neurotransmitter glutamate. The spike raster before and after the supplement of glutamine showed a significant glutamine-induced increase in spikes (Fig. 6h). Furthermore, we found no significant difference in spike amplitude (only a slight increase in the average spike waveform, +6.23 μ V, Supplementary Fig. 13a). **Electrophysiological activity was distributed throughout the 128 channels and was able to be visually represented in 3D plots based on the location of the electrodes (Fig. 6i and Video 6). These visualizations have the potential for future spatial and temporal mapping of electrical activities across neural organoids.]”***

5- It seems to me that the authors mistakenly introduce and calculate “flexural strength”, which is the maximum stress that a material can withstand while being bent or flexed before it yields, as “bending stiffness”, which is the ability of a structure to resist deformation when subjected to bending moments. The unit of the former, which authors calculate, is Pa whereas the unit of the latter is N.m². As a result, what we learn from the conducted tests is the high stretchability of the device and not the degree of its bendability, which was the point of my previous comment.

Author response: Thank you for pointing this out. We have calculated the bending stiffness using the equation updated in the Method section, and we also added this information in the result section.

The revised text reads as follows on (Results section, Page 6 lines 198-200):

“[The bending test also performed great stretchability and easy deformation of the mMPC,

and displacement of 500 μm requires only 0.005 N force (Fig. 2g, an average bending stiffness of $1.50 \times 10^{-3} \text{ N}\cdot\text{m}^2$).]”

(Method section, Page 20 lines 634-643)

[The bending test was performed at Shiyanjia Lab (www.shiyanjia.com), using a three-point bending method in an electronic universal testing machine. The bending stiffness was calculated using the following equation (We assumed the device was a completely thin sheet to replace the mesh shape for calculations and that the cross-section was rectangular):

$$\text{Bending stiffness: B.S.} = EI = \frac{FL^3}{4bh^3\delta} \cdot \frac{bh^3}{12}$$

F is the average force applied to the mMPC when the mMPC fractures (1.73 N); E is the modulus of elasticity, and I is the area moment of inertia; L is the distance between the two support points (124 mm); b is the width of the section (4.54 mm); and h is the thickness of the section (0.03 mm); δ is the deflection, we used the average maximum deflection (45.93 mm). The bending stiffness is calculated as $1.50 \times 10^{-3} \text{ N}\cdot\text{m}^2$.]

Reviewer 3:

The authors clarified the doubts identified in the first version of the article and significantly improved the final version of the manuscript.

Reviewer 4

Most of the points that were raised were addressed or corrected accordingly in the manuscript. However, there is still a concern about choosing the correct scRNAseq dataset to support the study. The main idea of this study is the development of a flexible multielectrode array that is conformal to 3D hHOs. But the dataset shown in the main figure (Figure 4) is from day-70 hHOs that were dissected into fragments and cultured attached to PLL-laminin coated flask in 2D for 3-4 days before being harvested for scRNAseq. This mixture of 3D and 2D cultured cells do not represent the 3D hHOs that were used for recording the electrical activity using mMPCs. The transcriptomic analysis of day-81 hHOs as 3D as it is before scRNAseq would be more representative of the cells that were measured for their electrical activity. In the rebuttal, the authors also commented that the difference between day-70 and day-81 hHOs in terms of cell type was insignificant. Then I wonder why the authors still chose day-70 dataset despite of its different conditions to 3D hHOs used for electrical activity measurement.

Author response: Thank you for pointing this out. We had previously considered only the cell type issue and thought that the dataset from day-70 hHOs would demonstrate the presence of oligodendrocytes in this aggregate, and, therefore, we chose this dataset. However, the point you raise is very important. Indeed, the dataset from day-81 hHOs is

more representative of the objects of the signal detection in the later results. Therefore, we have updated the day-81 hHOs dataset in Figure 4 and supplementary figures. The day-70 hHOs dataset, and the oligodendrocyte information was showed in the Supplementary Information.

Fig. 4 | Transcriptomic signature of hHOs. a, UMAP visualization of 7 scRNA-seq clusters from day-81 hHOs. The cluster of oligodendrocytes and NPCs was presented in day-70 hHOs dataset in Supplementary Fig. 8a-c. **b**, Violin plots of expression levels of markers in 7 clusters. **c**, Feature plots of hippocampal ZBTB20⁺ cells, SOX2⁺HOPX⁺ progenitors, SOX2⁺PAX6⁺ progenitors and PROX1⁺ neurons. **d**, UMAP visualization of the integrated dataset, the human hippocampus in GW22 and day-81 hHOs. (Left) All samples dataset. (Middle) The human hippocampus. (Right) Day-81 hHOs. The distribution of day-70 hHOs was present in Supplementary Fig. 8c.

REVIEWERS' COMMENTS

Reviewer #1 (Remarks to the Author):

The authors responded well to the reviewer's requests in rebuttal, but I still noticed some points that the authors would better consider as follows;

Comments

1. The authors modulate DMT organoid induction to more hippocampal side with less ChP region, but they made story that ChP tissues in DMT organoid tissues might affect the development of hippocampal tissue (lines 242-249), which is inconsistent with developmental biology. BMPs produced from ChP is important for expression of Lhx2 which is known as a hippocampal switching factor. The size of human ChP during developmental period is large and it is reasonable that the size of ChP in organoid is large as well.

The authors did not mention about BMP, rather mention about Wnt and Shh, but BMP role is also important (lines 90-96).

The authors can rewrite these developmental aspect correctly.

2. There are some speculations in result section which is inappropriate.

Line 315-318: "The reason might be that ~~~" is speculative description and inappropriate in result section.

Line 363-370: The explanation of 'others' also looks like discussion. It is good to describe in discussion section, but I also think the category 'others' is difficult to catch up for readers. It looks the author tried to change AD/PD related discussion in previous version, but 'others' looks broad concept and is difficult to be replaced instead of disease related description.

3. Line 334-335: The author suggested the presence of mature PROX1+ granule cells in the hHOs, but it is difficult to prove the cells are "mature" only by transcriptome analysis.

4. Line 413: I noticed Supplementary Fig 9c needs quantification.

Reviewer #2 (Remarks to the Author):

The authors have made extensive efforts and the revised manuscript is much improved. Especially, additional data is included in the manuscript to confirm neglectable deformation/compression to the organoids by applying mMPCs. This has very much released my previous concern.

However, these additional data also raise a new question to be addressed before this paper can be accepted. The key feature of the reported mMPCs is their ability to completely detect signals from entire

surface of organoids through conformal wrapping, enabled by their large number of electrodes, stretchability, and flexibility. However, in the tremendous sideview and micro-CT images of the attached organoid-mMPC combination from the newly included data (e.g., Supplementary figure 10 a and d and Supplementary figure 9b), the mMPCs only contact a relatively small portion of the organoid surface. It is indicated in the newly added figures and text that surface tension, by removing part of the bio-media, enables the contact between mMPC and organoid surface without compression of the organoid. Such surface tension induced adhesion does not significantly tense the meshes of device, but mainly bend it instead, as shown in the sideview images. As a result, these bended mMPCs mathematically can not follow the entire spherical surface, which is in line with the recording results in Fig. 6i that many electrodes located around the edge seem not active. The authors also mentioned in the response that the two layers do not completely wrap the hHO surface 100% and there would always be a distance between them. By considering all these together, it seems the claim of “with a wrapping coverage of over 75% of the organoid surface, which is larger than all current 3D MEAs” in the introduction and “the design of the 2-layer mMPC to conformally enclose the organoid fills the gap in detecting neural activities from the entire surface of organoids in previous studies”, and the “conformal” concepts throughout the manuscript may be misleading the readers or at least not fully supported. The authors should modify these sections to further tone down.

Besides, to remain the device-organoid adhesion through surface tension, the bio-media needs to be kept at relatively low level. As shown in Supplementary figure 10d, in some cases the organoid is even only partially merged in the bio media. Will the quality of organoids be affected by keeping in this condition for long term?

Reviewer #4 (Remarks to the Author):

In this revised manuscript, the authors have clarified and addressed all the raised points and improved the manuscript.

Response to reviewers for the manuscript (NCOMMS-23-19904B-Z)

Please see below, in blue, for a point-by-point response to the reviewers' comments and concerns. Their **comments and suggestions** are in **bold black** fonts. All page numbers refer to the revised manuscript file without tracked changes. The *updated contents* related to the corresponding comments are highlighted in *red italic* fonts. The *original texts* from the manuscript or any cited literature we have referred to are highlighted in *black italic* fonts.

Reviewer 1

The authors responded well to the reviewer's requests in rebuttal, but I still noticed some points that the authors would better consider as follows;

Comments

1. The authors modulate DMT organoid induction to more hippocampal side with less ChP region, but they made story that ChP tissues in DMT organoid tissues might affect the development of hippocampal tissue (lines 242-249), which is inconsistent with developmental biology. BMPs produced from ChP is important for expression of Lhx2 which is known as a hippocampal switching factor. The size of human ChP during developmental period is large and it is reasonable that the size of ChP in organoid is large as well.

The authors did not mention about BMP, rather mention about Wnt and Shh, but BMP role is also important (lines90-96).

The authors can rewrite these developmental aspect correctly.

Author response: Thank you for pointing this out. The effects of BMPs on the developing telencephalon and hippocampus are dynamic. There is not enough space to fully describe the interaction of BMPs and other signals to pattern telencephalon. I have rephrased these sentences and added the description for BMP role.

The revised text reads as follows on (Introduction section, Page 2 lines 83-95):

“[Human hippocampal primordium develops from dorsomedial telencephalon (DMT), which is regulated by BMPs, Wnts, and SHH^{24,25}. *Feedback pathways among BMPs and transcription factors, such as LHX2 and FGFs, pattern the telencephalon and induce and organize hippocampal fields^{26,27}. Previous in vivo^{24,25} and in vitro models^{7,28} proved the importance of Wnt3a and SHH signals in hippocampal development.* Researchers have identified the feasible timing to add BMP4 and CHIR (a chemical activator of Wnts) in the medium to generate DMT organoids. These DMT organoids gave rise to cortical hem (CH), choroid plexus (ChP), and hippocampal primordium²⁹. The CH and ChP release Wnts and BMPs to regulate hippocampus growth²⁴. In particular, several works to grow hHOs in vitro have opted for long-term use of the Wnt pathway (Wnt3a, 14 days⁷/ Wnt3a, 20 days⁵/ CHIR, 90 days⁶) to induce hippocampal fate. However, the relationship between the Wnt&SHH pathways, ChP in DMT organoids, and hippocampal differentiation has yet to be demonstrated.]”

(Results section, Page 8 lines 237-245):

“[Researchers have proved that some hippocampal neurons, accompanied by the ChP and CH regions, could be derived from DMT organoids (Supplementary Fig. 3a-b)²⁹. However, in an in vitro environment, the expansion of ChP tissues in DMT organoid tissues might affect the development of hippocampal tissue. Specifically, it has been observed that the TTR⁺ ChP tissue became more abundant while the neural tissue became smaller after long-term culture of the DMT organoids (Supplementary Fig. 3c-e). This result might be attributed to the enriched expression of BMPs in the ChP, which induced ChP epithelial fate and promoted its proliferation³⁴⁻³⁶. To generate hHOs from DMT tissue, alternative growth factors were considered to inhibit the ChP expansion.]”

2. There are some speculations in result section which is inappropriate.

Line 315-318: “The reason might be that ~~~” is speculative description and inappropriate in result section.

Line 363-370: The explanation of ‘others’ also looks like discussion. It is good to describe in discussion section, but I also think the category ‘others’ is difficult to catch up for readers. It looks the author tried to change AD/PD related discussion in previous version, but ‘others’ looks broad concept and is difficult to be replaced instead of disease related description.

Author response: Thank you for this suggestion. We have deleted the statement of our supposal in Line 315-318. The revised text reads as follows on (Results section, Page 10 lines 306-310):

“[Two samples of hHOs had similar cell types (Supplementary Fig. 5), but the hHO at day 81 was missing the population of cells co-expressing oligodendrocyte markers (Fig. 4a and Supplementary Fig. 8a-b). ~~The reason might be that the direct dissociation method used in day-81 hHOs produced more severe cellular damage, resulting in the disappearance of the oligodendrocyte cluster whose numbers were small.~~ These identified groups closely resemble the cell types found in the human developmental hippocampus up to 22 gestational weeks (GW22)³⁹.]”

Indeed, the speculation linking the "other" cluster to neurodegenerative diseases is subjective and would not be inappropriate in the Results section. WE deleted the description. The revised text reads as follows on (Results section, Page 11 lines 339-346):

“[The single-cell trajectory analysis also indicated that the identified cells in the ‘others’ subgroup were located at a separate termination of neuron branches that began from the NPCs (Supplementary Fig. 8d). They did not occur in the fetal hippocampus. The imperfect in vitro culture might lead to other neuronal cells in the hHO that did not belong to the hippocampus ~~or were associated with neurological disorders¹¹~~. But more experiments will be required to confirm causal relationship. Combining the immunofluorescence and scRNA-seq datasets, our results indicated the successful generation of hHOs.]”

3. Line 334-335: The author suggested the presence of mature PROX1+ granule cells in the hHOs, but it is difficult to prove the cells are “mature” only by transcriptome analysis.

Author response: Thank you for pointing this out. We mainly relied on the expression of mature neuron markers in Supplementary Fig.4d to figure out the maturation of PROX1⁺ granule cells. It's right that it's inappropriate to state the maturation only by transcriptome analysis. We have deleted the "mature" description.

The revised text reads as follows on (Results section, Page 10 lines 322-327):

*"[The ZBTB20⁺SOX2⁺ cells were clustered as immature cells (progenitors) and were further divided into HOPX⁺ and PAX6⁺ progenitor subgroups (Fig. 4c). Notably, some ZBTB20⁺PROX1⁺ cells were located in the cluster of excitatory neurons, suggesting the presence of **mature** PROX1⁺ granule cells in the hHOs (Fig. 4c). These results were consistent with the earlier immunofluorescence staining results in Fig. 3.]"*

4. Line 413: I noticed Supplementary Fig 9c needs quantification.

Author response: Thank you for this suggestion. Supplementary Fig. 9c shows the results of the biocompatibility test. We seeded the cells extracted from the mouse hippocampus on the mMPC and cultured them for over a month. We photographed the same location at different times. This result showed no significant change in the mMPC, and the number of cells increased. However, because the culture time was too long, the growing cells formed a cell sheet rather than monolayer cells. In this case, it was not feasible to quantify the number of cells, so we did not quantify the number of cells from this condition. We added the quantification data in Supplementary Figure 9d, comparing cell density between samples of cultured cells with and without the mMPC for 10 days.

The revised text reads as follows on (Results section, Page 12 lines 378-384):

*"[Because of the concern about the potential risk of leakage of liquid metal on the electrode locations and possible toxicity, we cultured cells from suckling mice hippocampus on the mMPC to verify its long-term biocompatibility. **After placing the mMPC in a petri dish, cells could proliferate and grow stably on the mMPC for more than 30 days (Supplementary Fig. 9c). The cell density was not significantly different from that of the control group without the mMPC (Supplementary Fig. 9d). A large number of axonal structures were present at electrode spots (Fig. 5c), indicating that the mMPC possessed excellent biocompatibility.**]"*

Supplementary Fig. 9 | Assembling of 128-channel mMPC and biocompatibility of mMPC. **a**, The integrated device coupling the mMPC with culture dishes. (Left): The mesh naturally formed a structure with a slightly concave center after being peeled from the PDMS slabs, which helped to place the hHO in it. The enlarged image also showed that a concave structure was present naturally after cutting off the redundant TPU/PU materials around the mMPC. (Upper right) When assembling the bottom mMPC, we fixed the distance between the two sides so that the bottom mMPC appeared in a "bowl" shape. (Lower Right): Assembly of two-layer mMPCs with the 35-mm dish. The TPU ball in the center was to reserve the space the hHO may need. **b**, (Left) The two-layer mMPCs with culture medium. After removing the TPU ball, some space existed between these two layers. (Middle and right) The top and the side view after placing one hHO assembly into the two-layer mMPC. Scale bar: 5 mm. When the culture solution was just submerged over the hHO assembly, the top mMPC covered the surface of the hHO assembly without significant squeezing. **c**, Biocompatibility. The mouse hippocampal cells grew in the mMPC for 30 days. Scale bar: 100 μ m. **d**, Comparison of cell density between cultured neural cells with and without mMPC for 10 days ($n = 5$, 5 independent measurements using 5 samples).

Reviewer 2:

The authors have made extensive efforts and the revised manuscript is much improved. Especially, additional data is included in the manuscript to confirm neglectable deformation/compression to the organoids by applying mMPCs. This has very much released my previous concern.

However, these additional data also raise a new question to be addressed before this paper can be accepted. The key feature of the reported mMPCs is their ability to completely detect signals from entire surface of organoids through conformal wrapping, enabled by their large number of electrodes, stretchability, and flexibility. However, in the tremendous sideview and micro-CT images of the attached organoid-mMPC combination from the newly included data (e.g., Supplementary figure 10 a and d and Supplementary figure 9b), the mMPCs only contact a relatively small portion of the organoid surface. It is indicated in the newly added figures and text that surface tension, by removing part of the bio-media, enables the contact between mMPC and organoid surface without compression of the organoid. Such surface tension induced adhesion does not significantly tense the meshes of device, but mainly bend it instead, as shown in the sideview images. As a result, these bended mMPCs mathematically can not follow the entire spherical surface, which is in line with the recording results in Fig. 6i that many electrodes located around the edge seem not active. The authors also mentioned in the response that the two layers do not completely wrap the hHO surface 100% and there would always be a distance between them. By considering all these together, it seems the claim of “with a wrapping coverage of over 75% of the organoid surface, which is larger than all current 3D MEAs” in the introduction and “the design of the 2-layer mMPC to conformally enclose the organoid fills the gap in detecting neural activities from the entire surface of organoids in previous studies”, and the “conformal” concepts throughout the manuscript may be misleading the readers or at least not fully supported. The authors should modify these sections to further tone down.

Author response: Thank you for this suggestion. Considering that the two-layer mMPC wrapping organoids of different sizes are different (Supplementary 9b and 10), we believe that using the estimated data for a single case to represent all is inappropriate. The estimated data in Supplementary Fig. 11c was removed and only used to show a certain distance always existed between two layers of mMPC. This result has been described in the discussion section. We also deleted our previous description of "conformal" concepts in this manuscript.

Supplementary Fig. 11 | Long-term change of hHOs in MEA and noninvasive assessment. **a**, (Left) The hHO on a commercial 2D MEA after culturing for 20 days. Scale bar: 500 μm . (Right) zoom-in view of the red box in the left figure showing the cell migration on the 2D MEA. Scale bar: 100 μm . **b**, (Left) bright-field images of hHOs on the normal mMPC after cocultured for 30 days and on the PLL-Laminin-coated mMPC for 20 days. Scale bar: 100 μm . **c**, The mMPC electrodes with and without PEDOT coating after culturing the hHO for 2 weeks. The red arrows point to the electrodes. Scale bar: 50 μm . **d**, Non-invasive assessment. The morphology of the same hHO in dish (Left); in the mMPC for 4 days (Middle); after fixation and immunostaining (Right). Scale bar: 100 μm in the staining image and 2 mm in others. **e**, The cross-sectional view showed the attachment boundary between the mMPC and the hHO. Scale bar: 100 μm . There was a certain distance between the two layers.

The revised text reads as follows on (Discussion section, Page 17 lines 488-496):

“[Our protocol to generate hHOs revealed that the *Wnt3a* and *SHH* signals promoted hippocampal fate while inhibiting the ChP expansion *in vitro*. These hHOs contained a significant number of hippocampal progenitors and DG granule neurons, with a high correlation of transcriptomic signature to the human *in vivo* hippocampus. **Our fabrication process of the mMPC is easy-to-operate and low equipment-dependent using low-cost**

materials. The mMPC offered multichannel recording, enabling the acquisition of as many nerve signals as possible from neural organoids. Our mMPC detected neural spike, synchronization, and oscillation activities in the hHO and the hHO assembly. These results indicated that hHOs grow a complex neural network.]”

(Discussion section, Page 17 lines 507-517)

“[Although the sandwich structure of our cyb-organoid greatly increased the channel number, this configuration might result in a slight pressure on the hHO. In order to make the top mMPC attach to the hHO, we would reduce the culture medium to only submerge a little past the surface of the hHO. The comparison of the signal recordings with and without the top mMPC illustrated that this squeezing improved the contact between the hHO and the electrodes. At the same time, however, it was obvious that the signals detected in this case were not exactly the spontaneous activity of the hHO in a free state but rather the neural activity under a slight squeeze. This limitation may cause some bias in our understanding of neural activity. In addition, this configuration leaves a distance between the two layers of mMPC and does not cover the complete surface of the organoid. In future work, we will focus on the problem by changing device geometry and using softer materials.]”

Besides, to remain the device-organoid adhesion through surface tension, the bio-media needs to be kept at relatively low level. As shown in Supplementary figure 10d, in some cases the organoid d is even only partially merged in the bio media. Will the quality of organoids be affected by keeping in this condition for long term?

Author response: Thank you for pointing this out. Indeed, keeping neural organoids in this condition long-term is not feasible because the organoids will die easily. If enough culture media is added, organoids can be cultivated for a long time, even if confined in the space between the two layers of mMPC. Therefore, we chose to remove a certain amount of culture solution only when the signal needs to be detected. This is also one of the reasons we recorded signals for a short period. We have added this to the discussion and methods section.

The revised text reads as follows on (Discussion section, Page 17 lines 498-505):

“[There are still some limitations of the current cyb-organoid. The free-floating status of all neural organoids poses great challenges for designing electronics and integrating electronics with organoids. Even if the hHOs can maintain its morphology in the mMPC for a long time, this fixation of hHO in the same position must be detrimental to its growth compared to the culture condition in a bioreactor. The decreasing medium used when recording signals also affects organoid growth. The medium diffusion in the system is not as good as the flow medium, but this problem may be improved if a flowing medium system can be provided for the cyb-organoid.]”

(Methods section, Page 22 lines 713-728)

“[The mMPC was fixed in a 100 mm dish with a 35 mm confocal dish in which the glass bottom was removed, leaving the end of flexible printed circuits outside the 100 mm dish. When assembling the two-layer mMPC, a TPU ball was used to pre-simulate the

replacement of the hHO and place it between the two layers. All relevant materials, including tapes, dishes, and devices, were sterilized by overnight UV light illumination. Before coculturing with the hHO, the mMPC was soaked in PBS to promote electrode and medium attachment. After removing the TPU ball from the side between the top and bottom meshes, the hHO was inserted into the two-layer mMPC, replacing PBS with 2.5 mL differentiation medium 2 (supplement with BDNF and GDNF) in the 35 mm dish. When the medium was enough to submerge the mMPC completely, the upper mMPC would float up to create a gap between the two layers. The hHO was then aspirated using a wide bore tip and gently placed in the gap. *In daily cultures, to maintain the hHO in a healthier state, a sufficient amount of medium would be provided so that the top mMPC would not squeeze the hHO. With enough medium, the hHO could be maintained with the mMPC system long-term in a 37 °C incubator. When detecting neural signal, a small amount of medium was carefully removed until the upper layer adhered to the hHO.]”*